# A General Framework for Sample-Efficient Function Approximation in Reinforcement Learning

**Zixiang Chen**[‡,*], **Chris Junchi Li**[◊,*], **Angela Yuan**[‡,*], **Quanquan Gu**[‡], **Michael I. Jordan**[◊,†]

[‡] Department of Computer Science, University of California, Los Angles
{chenzx19, hzyuan, qgu}@cs.ucla.edu
[◊] Department of Electrical Engineering and Computer Sciences, University of California, Berkeley
junchili@berkeley.edu, jordan@cs.berkeley.edu
[†] Department of Statistics, University of California, Berkeley

## Abstract

With the increasing need for handling large state and action spaces, general function approximation has become a key technique in reinforcement learning (RL). In this paper, we propose a general framework that unifies model-based and model-free RL, and an Admissible Bellman Characterization (ABC) class that subsumes nearly all Markov Decision Process (MDP) models in the literature for tractable RL. We propose a novel estimation function with decomposable structural properties for optimization-based exploration and the functional eluder dimension as a complexity measure of the ABC class. Under our framework, a new sample-efficient algorithm namely OPtimization-based ExploRation with Approximation (OPERA) is proposed, achieving regret bounds that match or improve over the best-known results for a variety of MDP models. In particular, for MDPs with low Witness rank, under a slightly stronger assumption, OPERA improves the state-of-the-art sample complexity results by a factor of $dH$. Our framework provides a generic interface to design and analyze new RL models and algorithms.

## 1 Introduction

Reinforcement learning (RL) is a decision-making process that seeks to maximize the expected reward when an agent interacts with the environment (Sutton & Barto, 2018). Over the past decade, RL has gained increasing attention due to its successes in a wide range of domains, including Atari games (Mnih et al., 2013), Go game (Silver et al., 2016), autonomous driving (Yurtsever et al., 2020), Robotics (Kober et al., 2013), etc. Existing RL algorithms can be categorized into value-based algorithms such as Q-learning (Watkins, 1989) and policy-based algorithms such as policy gradient (Sutton et al., 1999). They can also be categorized as a model-free approach where one directly models the value function classes, or alternatively, a model-based approach where one needs to estimate the transition probability.

Due to the intractably large state and action spaces that are used to model the real-world complex environment, function approximation in RL has become prominent in both algorithm design and theoretical analysis. It is a pressing challenge to design sample-efficient RL algorithms with general function approximations. In the special case where the underlying Markov Decision Processes (MDPs) enjoy certain linear structures, several lines of works have achieved polynomial sample complexity and/or $\sqrt{T}$ regret guarantees under either model-free or model-based RL settings. For linear MDPs where the transition probability and the reward function admit linear structure, Yang & Wang (2019) developed a variant of $Q$-learning when granted access to a generative model, Jin et al. (2020) proposed an LSVI-UCB algorithm with a $\widetilde{\mathcal{O}}(\sqrt{d^3 H^3 T})$ regret bound and Zanette et al. (2020a) further extended the MDP model and improved the regret to $\widetilde{\mathcal{O}}(dH\sqrt{T})$. Another line of work considers linear mixture MDPs Yang & Wang (2020); Modi et al. (2020); Jia et al. (2020); Zhou et al. (2021a), where the transition probability can be represented by a mixture of base models. In Zhou et al. (2021a), an $\widetilde{\mathcal{O}}(dH\sqrt{T})$ minimax optimal regret was achieved with weighted linear

---

*Equal contribution.

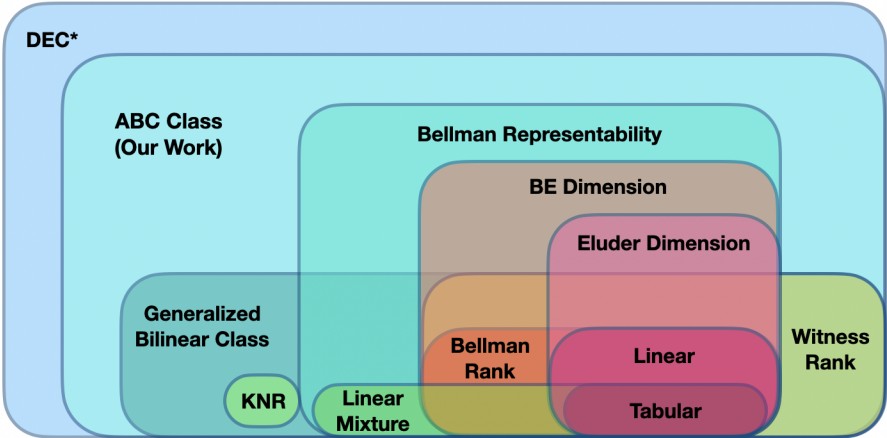

Figure 1: **Venn-Diagram Visualization of Prevailing Sample-Efficient RL Classes.** As by far the richest concept, the DEC framework is both a necessary and sufficient condition for sample-efficient interactive learning. BE dimension is a rich class that subsumes both low Bellman rank and low eluder dimension and addresses almost all model-free RL classes. The generalized Bilinear Class captures model-based RL settings including KNRs, linear mixture MDPs and low Witness rank MDPs, yet precludes some eluder-dimension based models. Bellman Representability is another unified framework that subsumes the vanilla bilinear classes but fails to capture KNRs and low Witness rank MDPs. Our ABC class encloses both generalized Bilinear Class and Bellman Representability and subsumes almost all known solvable MDP cases, with the exception of the $Q^*$ state-action aggregation and deterministic linear $Q^*$ MDP models, which neither Bilinear Class nor our ABC class captures.

regression and a Bernstein-type bonus. Other structural MDP models include the block MDPs (Du et al., 2019) and FLAMBE (Agarwal et al., 2020b) [1], to mention a few.

In a more general setting, however, there is still a gap between the plethora of MDP models and sample-efficient RL algorithms that can learn the MDP model with function approximation. The question remains open as to what constitutes minimal structural assumptions that admit sample-efficient reinforcement learning. To answer this question, there are several lines of work along this direction. Russo & Van Roy (2013); Osband & Van Roy (2014) proposed an structural condition named eluder dimension, and Wang et al. (2020) extended the LSVI-UCB for general linear function classes with small eluder dimension. Another line of works proposed low-rank structural conditions, including Bellman rank (Jiang et al., 2017; Dong et al., 2020) and Witness rank (Sun et al., 2019). Recently, Jin et al. (2021) proposed a complexity called Bellman eluder (BE) dimension, which unifies low Bellman rank and low eluder dimension. Concurrently, Du et al. (2021) proposed Bilinear Classes, which can be applied to a variety of loss estimators beyond vanilla Bellman error. Very recently, Foster et al. (2021) proposed Decision-Estimation Coefficient (DEC), which is a necessary and sufficient condition for sample-efficient interactive learning. To apply DEC to RL, they proposed a RL class named Bellman Representability, which can be viewed as a generalization of the Bilinear Class. Nevertheless, Sun et al. (2019) is limited to model-based RL, and Jin et al. (2021) is restricted to model-free RL. The only frameworks that can unify both model-based and model-free RL are Du et al. (2021) and Foster et al. (2021), but their sample complexity results when restricted to special MDP instances do not always match the best-known results. Viewing the above gap, we aim to answer the following question:

*Is there a unified framework that includes all model-free and model-based RL classes while maintaining sharp sample efficiency?*

In this paper, we tackle this challenging question and give a *nearly* affirmative answer to it. We summarize our contributions as follows:

- We propose a general framework called Admissible Bellman Characterization (ABC) that covers a wide set of structural assumptions in both model-free and model-based RL, such as linear

---

[1]In this paper, we use FLAMBE to refer to both the algorithm and the low-rank MDP with unknown feature mappings.

MDPs, FLAMBE, linear mixture MDPs, kernelized nonlinear regulator (Kakade et al., 2020), etc. Furthermore, our framework encompasses comparative structural frameworks such as the low Bellman eluder dimension and low Witness rank.

- Under our ABC framework, we design a novel algorithm, OPtimization-based ExploRation with Approximation (OPERA), based on maximizing the value function while constrained in a small confidence region around the model minimizing the estimation function.

- We apply our framework to several specific examples that are known to be not sample-efficient with value-based algorithms. For the kernelized nonlinear regulator (KNR), our framework is the first general framework to derive a $\sqrt{T}$ regret-bound result. For the witness rank, our framework yields a sharper sample complexity with a mild additional assumption compared to prior works.

We visualize and compare prevailing sample-efficient RL frameworks and ours in Figure 1. We can see that both the general Bilinear Class and our ABC frameworks capture most existing MDP classes, including the low Witness rank and the KNR models.

**Notation.** For a state-action sequence $s_1, a_1, \ldots, s_H$ in our given context, we use $\mathcal{J}_h := \sigma(s_1, a_1, \ldots, s_h)$ to denote the $\sigma$-algebra generated by trajectories up to step $h \in [H]$. Let $\pi_f$ denote the policy of following the max-$Q$ strategy induced by hypothesis $f$. When $f = f^i$ we write $\pi_{f^i}$ as $\pi^i$ for notational simplicity. We write $s_h \sim \pi$ to indicate the state-action sequence are generated by step $h \in [H]$ by following policy $\pi(\cdot \mid s)$ and transition probabilities $\mathbb{P}(\cdot \mid s, a)$ of the underlying MDP model $M$. We also write $a_h \sim \pi$ to mean $a_h \sim \pi(\cdot \mid s_h)$ for the $h$-th step. Let $\|\cdot\|_2$ denote the $\ell_2$-norm and $\|\cdot\|_\infty$ the $\ell_\infty$-norm of a given vector. Other notations will be explained at their first appearances.

## 2 PRELIMINARIES

We consider a finite-horizon, episodic Markov Decision Process (MDP) defined by the tuple $M = (\mathcal{S}, \mathcal{A}, \mathbb{P}, r, H)$, where $\mathcal{S}$ is the space of feasible states, $\mathcal{A}$ is the action space. $H$ is the horizon in each episode defined by the number of action steps in one episode, and $\mathbb{P} := \{\mathbb{P}_h\}_{h \in [H]}$ is defined for every $h \in [H]$ as the transition probability from the current state-action pair $(s_h, a_h) \in \mathcal{S} \times \mathcal{A}$ to the next state $s_{h+1} \in \mathcal{S}$. We use $r_h(s, a) \geq 0$ to denote the reward received at step $h \in [H]$ when taking action $a$ at state $s$ and assume throughout this paper that for any possible trajectories, $\sum_{h=1}^{H} r_h(s_h, a_h) \in [0, 1]$.

A deterministic policy $\pi$ is a sequence of functions $\{\pi_h : \mathcal{S} \mapsto \mathcal{A}\}_{h \in [H]}$, where each $\pi_h$ specifies a strategy at step $h$. Given a policy $\pi$, the action-value function is defined to be the expected cumulative rewards where the expectation is taken over the trajectory distribution generated by $\{(\mathbb{P}_h(\cdot \mid s_h, a_h), \pi_h(\cdot \mid s_h))\}_{h \in [H]}$ as

$$Q_h^\pi(s, a) := \mathbb{E}_\pi \left[ \sum_{h'=h}^{H} r_{h'}(s_{h'}, a_{h'}) \,\middle|\, s_h = s, a_h = a \right].$$

Similarly, we define the state-value function for policy $\pi$ as the expected cumulative rewards as

$$V_h^\pi(s) := \mathbb{E}_\pi \left[ \sum_{h'=h}^{H} r_{h'}(s_{h'}, a_{h'}) \,\middle|\, s_h = s \right].$$

We use $\pi^*$ to denote the optimal policy that satisfies $V_h^{\pi^*}(s) = \max_\pi V_h^\pi(s)$ for all $s \in \mathcal{S}$ (Puterman, 2014). For simplicity, we abbreviate $V_h^{\pi^*}$ as $V_h^*$ and $Q_h^{\pi^*}$ as $Q_h^*$. Moreover, for a sequence of value functions $\{Q_h\}_{h \in [H]}$, the Bellman operator at step $h$ is defined as:

$$(\mathcal{T}_h Q_{h+1})(s, a) = r_h(s, a) + \mathbb{E}_{s' \sim \mathbb{P}_h(\cdot \mid s, a)} \max_{a' \in \mathcal{A}} Q_{h+1}(s', a').$$

We also call $Q_h - (\mathcal{T}_h Q_{h+1})$ the Bellman error (or Bellman residual). The goal of an RL algorithm is to find an $\epsilon$-optimal policy such that $V_1^\pi(s_1) - V_1^*(s_1) \leq \epsilon$. For an RL algorithm that updates the policy $\pi^t$ for $T$ iterations, the cumulative regret is defined as

$$\text{Regret}(T) := \sum_{t=1}^{T} \left[ V_1^*(s_1) - V_1^{\pi^t}(s_1) \right],$$

**Hypothesis Classes.** Following Du et al. (2021), we define the hypothesis class for both model-free and model-based RL. Generally speaking, a hypothesis class is a set of functions that are used to estimate the value functions (for model-free RL) or the transitional probability and reward (for model-based RL). Specifically, a hypothesis class $\mathcal{F}$ on a finite-horizon MDP is the Cartesian product of $H$ hypothesis classes $\mathcal{F} := \mathcal{F}_1 \times \ldots \times \mathcal{F}_H$ in which each hypothesis $f = \{f_h\}_{h \in [H]} \in \mathcal{F}$ can be identified by a pair of value functions $\{Q_f, V_f\} = \{Q_{h,f}, V_{h,f}\}_{h \in [H]}$. Based on the value function pair, it is natural to introduce the greedy policy $\pi_{h,f}(s) = \arg\max_{a \in \mathcal{A}} Q_{h,f}(s, a)$ at each step $h \in [H]$, and the corresponding $\pi_f(s)$ as the sequence of time-dependent policies $\{\pi_{h,f}\}_{h=0}^{H-1}$. An example of a model-free hypothesis class is defined by a sequence of action-value function $\{Q_{h,f}\}_{h \in [H]}$. The corresponding state-value function is given by:

$$V_{h,f}(s) = \mathbb{E}_{a \sim \pi_{h,f}} [Q_{h,f}(s, a)].$$

In another example that falls under the model-based RL setting, where for each hypothesis $f \in \mathcal{F}$ we have the knowledge of the transition matrix $\mathbb{P}_f$ and the reward function $r_f$. We define the value function $Q_{h,f}$ corresponding to hypothesis $f$ as the optimal value function following $M_f := (\mathbb{P}_f, r_f)$:

$$Q_{h,f}(s, a) = Q^*_{h,M_f}(s, a) \qquad \text{and} \quad V_{h,f}(s) = V^*_{h,M_f}(s).$$

We also need the following realizability assumption that requires the true model $M_{f^*}$ (model-based RL) or the optimal value function $f^*$ (model-free RL) to belong to the hypothesis class $\mathcal{F}$.

**Assumption 1** (Realizability). For an MDP model $M$ and a hypothesis class $\mathcal{F}$, we say that the hypothesis class $\mathcal{F}$ is *realizable with respect to $M$* if there exists a $f^* \in \mathcal{F}$ such that for any $h \in [H]$, $Q^*_h(s, a) = Q_{h,f^*}(s, a)$. We call such $f^*$ an *optimal hypothesis*.

This assumption has also been made in the Bilinear Classes (Du et al., 2021) and low Bellman eluder dimension frameworks (Jin et al., 2021). We also define the $\epsilon$-*covering number* of $\mathcal{F}$ under a well-defined metric $\rho$ of a hypothesis class $\mathcal{F}$:[2]

**Definition 2** ($\epsilon$-covering Number of Hypothesis Class). For any $\epsilon > 0$ and a hypothesis class $\mathcal{F}$, we use $N_{\mathcal{F}}(\epsilon)$ to denote the $\epsilon$-covering number, which is the smallest possible cardinality of (an $\epsilon$-cover) $\mathcal{F}_\epsilon$ such that for any $f \in \mathcal{F}$ there exists a $f' \in \mathcal{F}_\epsilon$ such that $\rho(f, f') \leq \epsilon$.

**Functional Eluder Dimension.** We proceed to introduce our new complexity measure, *functional eluder dimension*, which generalizes the concept of *eluder dimension* firstly proposed in bandit literature (Russo & Van Roy, 2013; 2014). It has since become a widely used complexity measure for function approximations in RL (Wang et al., 2020; Ayoub et al., 2020; Jin et al., 2021; Foster et al., 2021). Here we revisit its definition:

**Definition 3** (Eluder Dimension). For a given space $\mathcal{X}$ and a class $\mathcal{F}$ of functions defined on $\mathcal{X}$, the *eluder dimension* $\dim_{\mathcal{E}}(\mathcal{F}, \epsilon)$ is the length of the existing longest sequence $x_1, \ldots, x_n \in \mathcal{X}$ satisfying for some $\epsilon' \geq \epsilon$ and any $2 \leq t \leq n$, there exist $f_1, f_2 \in \mathcal{F}$ such that $\sqrt{\sum_{i=1}^{t-1} (f_1(x_i) - f_2(x_i))^2} \leq \epsilon'$ while $|f_1(x_t) - f_2(x_t)| > \epsilon'$.

The eluder dimension is usually applied to the state-action space $\mathcal{X} = \mathcal{S} \times \mathcal{A}$ and the corresponding value function class $\mathcal{F} : \mathcal{S} \times \mathcal{A} \to \mathbb{R}$ (Jin et al., 2021; Wang et al., 2020). We extend the concept of eluder dimension as a complexity measure of the hypothesis class, namely, the *functional eluder dimension*, which is formally defined as follows.

**Definition 4** (Functional Eluder Dimension). For a given hypothesis class $\mathcal{F}$ and a function $G$ defined on $\mathcal{F} \times \mathcal{F}$, the *functional eluder dimension (FE dimension)* $\dim_{\text{FE}}(\mathcal{F}, G, \epsilon)$ is the length of the existing longest sequence $f_1, \ldots, f_n \in \mathcal{F}$ satisfying for some $\epsilon' \geq \epsilon$ and any $2 \leq t \leq n$, there exists $g \in \mathcal{F}$ such that $\sqrt{\sum_{i=1}^{t-1} (G(g, f_i))^2} \leq \epsilon'$ while $|G(g, f_t)| > \epsilon'$. Function $G$ is dubbed as the *coupling function*.

The notion of functional eluder dimension introduced in Definition 4 is generalizable in a straightforward fashion to a sequence $G := \{G_h\}_{h \in [H]}$ of coupling functions: we simply set $\dim_{\text{FE}}(\mathcal{F}, G, \epsilon) = \max_{h \in [H]} \dim_{\text{FE}}(\mathcal{F}, G_h, \epsilon)$ to denote the FE dimension of $\{G_h\}_{h \in [H]}$. The Bellman eluder (BE)

---

[2]For example for model-free cases where $f, g$ are value functions, $\rho(f, g) = \max_{h \in [H]} \|f_h - g_h\|_\infty$. For model-based RL where $f, g$ are transition probabilities, we adopt $\rho(\mathbb{P}, \mathbb{Q}) = \max_{h \in [H]} \int (\sqrt{d\mathbb{P}_h} - \sqrt{d\mathbb{Q}_h})^2$ which is the maximal (squared) Hellinger distance between two probability distribution sequences.

dimension recently proposed by (Jin et al., 2021) is in fact a special case of FE dimension with a specific choice of coupling function sequence.[3] As will be shown later, our framework based on FE dimension with respect to the corresponding coupling function captures many specific MDP instances such as the kernelized nonlinear regulator (KNR) (Kakade et al., 2020) and the generalized linear Bellman complete model (Wang et al., 2019), which are not captured by the framework of low BE dimension. As we will see in later sections, introducing the concept of FE dimension allows the coverage of a strictly wider range of MDP models and hypothesis classes.

# 3 ADMISSIBLE BELLMAN CHARACTERIZATION FRAMEWORK

In this section, we introduce the framework of admissible Bellman characterization.

## 3.1 ADMISSIBLE BELLMAN CHARACTERIZATION

Given an MDP $M$, a sequence of states and actions $s_1, a_1, \ldots, s_H$, two hypothesis classes $\mathcal{F}$ and $\mathcal{G}$ satisfying the realizability assumption (Assumption 1),[4] and a *discriminator function class* $\mathcal{V} = \{v(s, a, s') : \mathcal{S} \times \mathcal{A} \times \mathcal{S} \to \mathbb{R}\}$, the *estimation function* $\ell = \{\ell_{h,f'}\}_{h \in [H], f' \in \mathcal{F}}$ is an $\mathbb{R}^{d_s}$-valued function defined on the set consisting of $o_h := (s_h, a_h, s_{h+1}) \in \mathcal{S} \times \mathcal{A} \times \mathcal{S}$, $f \in \mathcal{F}$, $g \in \mathcal{G}$ and $v \in \mathcal{V}$ and serves as a surrogate loss function of the Bellman error. Note that our estimation function is a vector-valued function, and is more general than the scalar-valued estimation function (or discrepancy function) used in Foster et al. (2021); Du et al. (2021). The *discriminator* $v$ originates from the function class the Integral Probability Metrics (IPM) (Müller, 1997) is taken with respect to (as a metric between two distributions), and is also used in the definition of Witness rank (Sun et al., 2019). We use a coupling function $G_{h,f^*}(f, g)$ defined on $\mathcal{F} \times \mathcal{F}$ to characterize the interaction between two hypotheses $f, g \in \mathcal{F}$. The subscript $f^*$ is an indicator of the *true model* and is by default unchanged throughout the context. When the two hypotheses coincide, our characterization of the coupling function reduces to the Bellman error.

**Definition 5** (Admissible Bellman Characterization). Given an MDP $M$, two hypothesis classes $\mathcal{F}, \mathcal{G}$ satisfying the realizability assumption (Assumption 1) and $\mathcal{F} \subset \mathcal{G}$, an estimation function $\ell_{h,f'} : (\mathcal{S} \times \mathcal{A} \times \mathcal{S}) \times \mathcal{F} \times \mathcal{G} \times \mathcal{V} \to \mathbb{R}^{d_s}$, an operation policy $\pi_{\mathrm{op}}$ and a constant $\kappa \in (0, 1]$, we say that $G$ is an *admissible Bellman characterization* of $(M, \mathcal{F}, \mathcal{G}, \ell)$ if the following conditions hold:

(i) **(Dominating Average Estimation Function)** For any $f, g \in \mathcal{F}$

$$\max_{v \in \mathcal{V}} \mathbb{E}_{s_h \sim \pi_g, a_h \sim \pi_{\mathrm{op}}} || \mathbb{E}_{s_{h+1}} [\ell_{h,g}(o_h, f_{h+1}, f_h, v) \mid s_h, a_h] ||^2 \geq (G_{h,f^*}(f, g))^2.$$

(ii) **(Bellman Dominance)** For any $(h, f) \in [H] \times \mathcal{F}$,

$$\kappa \cdot \left| \mathbb{E}_{s_h, a_h \sim \pi_f} [Q_{h,f}(s_h, a_h) - r(s_h, a_h) - V_{h+1,f}(s_{h+1})] \right| \leq |G_{h,f^*}(f, f)|.$$

We further say $(M, \mathcal{F}, \mathcal{G}, \ell, G)$ is an *ABC class* if $G$ is an admissible Bellman characterization of $(M, \mathcal{F}, \mathcal{G}, \ell)$.

In Definition 5, one can choose either $\pi_{\mathrm{op}} = \pi_g$ or $\pi_{\mathrm{op}} = \pi_f$. We refer readers to §D for further explanations on $\pi_{\mathrm{op}}$. The ABC class is quite general and de facto covers many existing MDP models; see §3.2 for more details.

**Comparison with Existing MDP Classes.** Here we compare our ABC class with three recently proposed MDP structural classes: Bilinear Classes (Du et al., 2021), low Bellman eluder dimension (Jin et al., 2021), and Bellman Representability (Foster et al., 2021).

- *Bilinear Classes.* Compared to the structural framework of Bilinear Class in Du et al. (2021, Definition 4.3), Definition 5 of Admissible Bellman Characterization does not require a bilinear structure and recovers the Bilinear Class when we set $G_{h,f^*}(f, g) = \langle W_h(g) - W_h(f^*), X_h(f) \rangle$. Our ABC class is strictly broader than the Bilinear Class since the latter does not capture low eluder dimension models, and our ABC class does. In addition, the ABC class admits an estimation function that is *vector-valued*, and the corresponding algorithm achieves a $\sqrt{T}$-regret for KNR case while the BiLin-UCB algorithm for Bilinear Classes (Du et al., 2021) does not.

---

[3]Indeed, when the coupling function is chosen as the expected Bellman error $G_h(g, f) := \mathbb{E}_{\pi_{h,f}}(Q_{h,g} - \mathcal{T}_h Q_{g,h+1})$ where $\mathcal{T}_h$ denotes the Bellman operator, we recover the definition of BE dimension (Jin et al., 2021), i.e. $\dim_{\mathrm{FE}}(\mathcal{F}, G, \epsilon) = \dim_{\mathrm{BE}}(\mathcal{F}, G, \epsilon)$.

[4]We assume $\mathcal{F} \subseteq \mathcal{G}$ throughout this paper and in the general case where $\mathcal{F} \not\subseteq \mathcal{G}$, we overload $\mathcal{G} := \mathcal{F} \cup \mathcal{G}$.

- *Low Bellman Eluder Dimension.* Definition 5 subsumes the MDP class of low BE dimension when $\ell_{h,f'}(o_h, f_{h+1}, g_h, v) := Q_{h,g}(s_h, a_h) - r_h - V_{h+1,f}(s_{h+1})$. Moreover, our definition unifies the $V$-type and $Q$-type problems under the same framework by the notion of $\pi_{\mathrm{op}}$. We will provide a more detailed discussion on this in §3.2. Our extension from the concept of the Bellman error to estimation function (i.e. the surrogate of the Bellman error) enables us to accommodate model-based RL for linear mixture MDPs, KNR model, and low Witness rank.

- *Bellman Representability.* Foster et al. (2021) proposed DEC framework which is another MDP class that unifies both the Bilinear Class and the low BE dimension. Indeed, our ABC framework introduced in Definition 5 shares similar spirits with the Bellman Representability Definition F.1 in Foster et al. (2021). Nevertheless, our framework and theirs bifurcate from the base point: our work studies an optimization-based exploration instead of the posterior sampling-based exploration in Foster et al. (2021). Structurally different from their DEC framework, our ABC requires estimation functions to be vector-valued, introduces the discriminator function $v$, and imposes the weaker Bellman dominance property (i) in Definition 5 than the corresponding one as in Foster et al. (2021, Eq. (166)). In total, this allows broader choices of coupling function $G$ as well as our ABC class (with low FE dimension) to include as special instances both low Witness rank and KNR models, which are not captured in Foster et al. (2021).

**Decomposable Estimation Function.** Now we introduce the concept of *decomposable estimation function*, which generalizes the Bellman error in earlier literature and plays a pivotal role in our algorithm design and analysis.

**Definition 6** (Decomposable Estimation Function). A *decomposable estimation function* $\ell : (\mathcal{S} \times \mathcal{A} \times \mathcal{S}) \times \mathcal{F} \times \mathcal{G} \times \mathcal{V} \to \mathbb{R}^{d_s}$ is a function with bounded $\ell_2$-norm such that the following two conditions hold:

(i) **(Decomposability)** There exists an operator that maps between two hypothesis classes $\mathcal{T}(\cdot) : \mathcal{F} \to \mathcal{G}^5$ such that for any $f \in \mathcal{F}$, $(h, f', g, v) \in [H] \times \mathcal{F} \times \mathcal{G} \times \mathcal{V}$ and all possible $o_h$

$$\ell_{h,f'}(o_h, f_{h+1}, g_h, v) - \mathbb{E}_{s_{h+1}}\left[\ell_{h,f'}(o_h, f_{h+1}, g_h, v) \mid s_h, a_h\right] = \ell_{h,f'}(o_h, f_{h+1}, \mathcal{T}(f)_h, v).$$

Moreover, if $f = f^*$, then $\mathcal{T}(f) = f^*$ holds.

(ii) **(Global Discriminator Optimality)** For any $f \in \mathcal{F}$ there exists a global maximum $v_h^*(f) \in \mathcal{V}$ such that for any $(h, f', g, v) \in [H] \times \mathcal{F} \times \mathcal{G} \times \mathcal{V}$ and all possible $o_h$

$$||\mathbb{E}_{s_{h+1}}\left[\ell_{h,f'}(o_h, f_{h+1}, f_h, v_h^*(f)) \mid s_h, a_h\right]|| \geq ||\mathbb{E}_{s_{h+1}}\left[\ell_{h,f'}(o_h, f_{h+1}, f_h, v) \mid s_h, a_h\right]||.$$

Compared with the discrepancy function or estimation function used in prior work (Du et al., 2021; Foster et al., 2021), our estimation function (EF) admits the unique properties listed as follows:

(a) Our EF enjoys a decomposable property inherited from the Bellman error — intuitively speaking, the decomposability can be seen as a property shared by all functions in the form of the difference of a $\mathcal{J}_h$-measurable function and a $\mathcal{J}_{h+1}$-measurable function;

(b) Our EF involves a discriminator class and assumes the global optimality of the discriminator on all $(s_h, a_h)$ pairs;

(c) Our EF is a vector-valued function which is more general than a scalar-valued estimation function (or the discrepancy function).

We remark that when $f = g$, $\mathbb{E}_{s_{h+1}}\left[\ell_{h,f'}(o_h, f_{h+1}, f_h, v) \mid s_h, a_h\right]$ measures the discrepancy in optimality between $f$ and $f^*$. In particular, when $f = f^*$, $\mathbb{E}_{s_{h+1}}\left[\ell_{h,f'}(o_h, f_{h+1}^*, f_h^*, v) \mid s_h, a_h\right] = 0$. Consider a special case when $\ell_{h,f'}(o_h, f_{h+1}, g, v) := Q_{h,g}(s_h, a_h) - r(s_h, a_h) - V_{h+1,f}(s_{h+1})$. Then the decomposability (i) in Definition 6 reduces to

$$[Q_{h,g}(s_h, a_h) - r(s_h, a_h) - V_{h+1,f}(s_{h+1})] - [Q_{h,g}(s_h, a_h) - (\mathcal{T}_h V_{h+1})(s_h, a_h)]$$
$$= (\mathcal{T}_h V_{h+1})(s_h, a_h) - r(s_h, a_h) - V_{h+1,f}(s_{h+1}).$$

In addition, we make the following Lipschitz continuity assumption on the estimation function.

---

[5]The decomposability item (i) in Definition 6 directly implies that a Generalized Completeness condition similar to Assumption 14 of Jin et al. (2021) holds.

**Assumption 7** (Lipschitz Estimation Function). There exists a $L > 0$ such that for any $(h, f', f, g, v) \in [H] \times \mathcal{F} \times \mathcal{F} \times \mathcal{G} \times \mathcal{V}, (\widetilde{f}, \widetilde{g}, \widetilde{v}, \widetilde{f}') \in \mathcal{F} \times \mathcal{G} \times \mathcal{V} \times \mathcal{F}$ and all possible $o_h$,

$$\left\| \ell_{h,f'}(\cdot, f, g, v) - \ell_{h,f'}(\cdot, \widetilde{f}, g, v) \right\|_\infty \le L\rho(f, \widetilde{f}), \qquad \left\| \ell_{h,f'}(\cdot, f, g, v) - \ell_{h,f'}(\cdot, f, \widetilde{g}, v) \right\|_\infty \le L\rho(g, \widetilde{g}),$$

$$\left\| \ell_{h,f'}(\cdot, f, g, v) - \ell_{h,f'}(\cdot, f, g, \widetilde{v}) \right\|_\infty \le L \left\| v - \widetilde{v} \right\|_\infty, \quad \left\| \ell_{h,f'}(\cdot, f, g, v) - \ell_{h,\widetilde{f}'}(\cdot, f, g, v) \right\|_\infty \le L\rho(f', \widetilde{f}').$$

Note that we have omitted the subscript $h$ of hypotheses in Assumption 7 for notational simplicity. We further define the induced estimation function class as $\mathcal{L} = \{\ell_{h,f'}(\cdot, f, g, v) : (h, f', f, g, v) \in [H] \times \mathcal{F} \times \mathcal{F} \times \mathcal{G} \times \mathcal{V}\}$. We can show that under Assumption 7, the covering number of the induced estimation function class $\mathcal{L}$ can be upper bounded as $N_\mathcal{L}(\epsilon) \le N_\mathcal{F}^2(\frac{\epsilon}{4L}) N_\mathcal{G}(\frac{\epsilon}{4L}) N_\mathcal{V}(\frac{\epsilon}{4L})$, where $N_\mathcal{F}(\epsilon), N_\mathcal{G}(\epsilon), N_\mathcal{V}(\epsilon)$ are the $\epsilon$-covering number of $\mathcal{F}, \mathcal{G}$ and $\mathcal{V}$, respectively. Later in our theoretical analysis in §4, our regret upper bound will depend on the growth rate of the covering number or the *metric entropy*, $\log N_\mathcal{L}(\epsilon)$.

### 3.2 MDP INSTANCES IN THE ABC CLASS

In this subsection, we present a number of MDP instances that belong to ABC class with low FE dimension. As we have mentioned before, for all special cases with $\ell_{h,f'}(o_h, f_{h+1}, g_h, v) := Q_{h,g}(s_h, a_h) - r_h - V_{h+1,f}(s_{h+1})$, both conditions in Definition 5 are satisfied automatically with $G_{h,f^*}(f, g) = \mathbb{E}_{s_h \sim \pi_g, a_h \sim \pi_{op}}[Q_{h,f}(s_h, a_h) - r_h - V_{h+1,f}(s_{h+1})]$. The FE dimension under this setting recovers the the BE dimension. Thus, all model-free RL models with low BE dimension (Jin et al., 2021) belong to our ABC class with low FE dimension. In the rest of this subsection, our focus shifts to the model-based RLs that belong to the ABC class: linear mixture MDPs, low Witness rank, and kernelized nonlinear regulator.

**Linear Mixture MDPs.** We start with a model-based RL with a linear structure called the *linear mixture MDP* (Modi et al., 2020; Ayoub et al., 2020; Zhou et al., 2021b). For known transition and reward feature mappings $\phi(s, a, s') : \mathcal{S} \times \mathcal{A} \times \mathcal{S} \to \mathcal{H}, \psi(s, a) : \mathcal{S} \times \mathcal{A} \to \mathcal{H}$ taking values in a Hilbert space $\mathcal{H}$ and an unknown $\theta^* \in \mathcal{H}$, a linear mixture MDP assumes that for any $(s, a, s') \in \mathcal{S} \times \mathcal{A} \times \mathcal{S}$ and $h \in [H]$, the transition probability $\mathbb{P}_h(s' \mid s, a)$ and the reward function $r(s, a)$ are linearly parameterized as

$$\mathbb{P}_h(s' \mid s, a) = \langle \theta_h^*, \phi(s, a, s') \rangle, \qquad r(s, a) = \langle \theta_h^*, \psi(s, a) \rangle.$$

We provide the following proposition, which shows that linear mixture MDPs belong to the ABC class with low FE dimension.

**Proposition 8** (Linear Mixture MDP $\subset$ ABC with Low FE Dimension). The linear mixture MDP model belongs to the ABC class with estimation function

$$\ell_{h,f'}(o_h, f_{h+1}, g_h, v) = \theta_{h,g}^\top \left[ \psi(s_h, a_h) + \sum_{s'} \phi(s_h, a_h, s') V_{h+1,f'}(s') \right] - r_h - V_{h+1,f'}(s_{h+1}), \quad (3.1)$$

and coupling function $G_{h,f^*}(f, g) = \left\langle \theta_{h,g} - \theta_h^*, \mathbb{E}_{s_h, a_h \sim \pi_f} \left[ \psi(s_h, a_h) + \sum_{s'} \phi(s_h, a_h, s') V_{h+1,f}(s') \right] \right\rangle$. Moreover, it has a low FE dimension.

**Low Witness Rank.** We defer the formal definition of witness rank to §E.2 and provide the following proposition showing that low Witness rank models belongs to our ABC class with low FE dimension.

**Proposition 9** (Low Witness Rank $\subset$ ABC with Low FE Dimension). The low Witness rank model belongs to the ABC class with estimation function

$$\ell_{h,f'}(o_h, f_{h+1}, g_h, v) = \mathbb{E}_{\widetilde{s} \sim g_h} v(s_h, a_h, \widetilde{s}) - v(s_h, a_h, s_{h+1}), \quad (3.2)$$

and coupling function $G_{h,f^*}(f, g) = \langle W_h(g), X_h(f) \rangle$. Moreover, it has a low FE dimension.

**Kernelized Nonlinear Regulator.** The *kernelized nonlinear regulator (KNR)* proposed recently by Mania et al. (2020); Kakade et al. (2020) models a nonlinear control dynamics on an RKHS $\mathcal{H}$ of finite or countably infinite dimensions. Under the KNR setting, given current $s_h, a_h$ at step $h \in [H]$ and a known feature mapping $\phi : \mathcal{S} \times \mathcal{A} \to \mathcal{H}$, the subsequent state obeys a Gaussian distribution with mean vector $U_h^* \phi(s_h, a_h)$ and homoskedastic covariance $\sigma^2 I$, where $\left\{ U_h^* \in \mathbb{R}^{d_s} \times \mathcal{H} \right\}_{h \in [H]}$ are true model parameters and $d_s$ is the dimension of the state space. Mathematically, we have for each $h = 1, \ldots, H$,

$$s_{h+1} = U_h^* \phi(s_h, a_h) + \epsilon_{h+1}, \quad \text{where } \epsilon_{h+1} \stackrel{\text{i.i.d.}}{\sim} \mathcal{N}(0, \sigma^2 I). \quad (3.3)$$

Furthermore, we assume bounded reward $r \in [0, 1]$ and uniformly bounded feature map $||\phi(s, a)|| \le B$. The following proposition shows that KNR belongs to the ABC class with low FE dimension.

---

**Algorithm 1** OPtimization-based ExploRation with Approximation (OPERA)

---

1: **Initialize**: $\mathcal{D}_h = \varnothing$ for $h = 1, \ldots, H$
2: **for** iteration $t = 1, 2, \ldots, T$ **do**
3:      Set $\pi^t := \pi_{f^t}$ where $f^t$ is taken as $\mathrm{argmax}_{f \in \mathcal{F}} Q_{1,f}(s_1, \pi_f(s_1))$ subject to

$$\max_{v \in \mathcal{V}} \left\{ \sum_{i=1}^{t-1} ||\ell_{h,f^i}(o_h^i, f_{h+1}, f_h, v)||^2 - \inf_{g_h \in \mathcal{G}_h} \sum_{i=1}^{t-1} ||\ell_{h,f^i}(o_h^i, f_{h+1}, g_h, v)||^2 \right\} \le \beta \quad \text{for all } h \in [H]$$

(4.1)

4:      For any $h \in [H]$, collect tuple $(r_h, s_h, a_h, s_{h+1})$ by rolling in $s_h \sim \pi^t$ and executing $a_h \sim \pi_{\text{est}}$

5:      Augment $\mathcal{D}_h = \mathcal{D}_h \cup \left\{ (r_h, s_h, a_h, s_{h+1}) \right\}$
6: **end for**
7: **Output**: $\pi_{\text{out}}$ uniformly sampled from $\{\pi^t\}_{t=1}^T$

---

**Proposition 10** (KNR $\subset$ ABC with Low FE Dimension). KNR belongs to the ABC class with estimation function

$$\ell_{h,f'}(o_h, f_{h+1}, g_h, v) = U_{h,g}\phi(s_h, a_h) - s_{h+1},$$

(3.4)

and coupling function $G_{h,f^*}(f, g) := \sqrt{\mathbb{E}_{s_h, a_h \sim \pi_g} ||(U_{h,f} - U_h^*)\phi(s_h, a_h)||^2}$. Moreover, it has a low FE dimension.

Although the dimension of the RKHS $\mathcal{H}$ can be infinite, our complexity analysis depends solely on its effective dimension $d_\phi$.

## 4    Algorithm and Main Results

In this section, we present an RL algorithm for the ABC class. Then we present the regret bound of this algorithm, along with its implications to several MDP instances in the ABC class.

### 4.1    OPERA Algorithm

We first present the *OPtimization-based ExploRation with Approximation (OPERA)* algorithm in Algorithm 1, which finds an $\epsilon$-optimal policy in polynomial time. Following earlier algorithmic art in the same vein e.g., GOLF (Jin et al., 2021), the core optimization step of OPERA is optimization-based exploration under the constraint of an identified confidence region; we additionally introduce an estimation policy $\pi_{\text{est}}$ sharing the similar spirit as in Du et al. (2021). Due to space limit, we focus on the $Q$-type analysis here and defer the $V$-type results to §D in the appendix.[6]

Pertinent to the constrained optimization subproblem in Eq. (4.1) of our Algorithm 1, we adopt the confidence region based on a general DEF, extending the Bellman-error-based confidence region used in Jin et al. (2021). As a result of such an extension, our algorithm can deal with more complex models such as low Witness rank and KNR. Similar to existing literature on RL theory with general function approximation, our algorithm is in general computationally inefficient. Yet OPERA is oracle efficient given the oracle for solving the optimization problem in Line 3 of Algorithm 1. We will discuss its computational issues in detail in §E.1, §E.2 and §E.3.

### 4.2    Regret Bounds

We are ready to present the main theoretical results of our ABC class with low FE dimension:

**Theorem 11** (Regret Bound of OPERA). For an MDP $M$, hypothesis classes $\mathcal{F}, \mathcal{G}$, a Decomposable Estimation Function $\ell$ satisfying Assumption 7, an admissible Bellman characterization $G$, suppose $(M, \mathcal{F}, \mathcal{G}, \ell, G)$ is an ABC class with low functional eluder dimension. For any fixed $\delta \in (0, 1)$, we choose $\beta = \mathcal{O}\left(\log(THN_{\mathcal{L}}(1/T)/\delta)\right)$ in Algorithm 1. Then for the on-policy case when $\pi_{\text{op}} = \pi_{\text{est}} = \pi^t$, with probability at least $1 - \delta$, the regret is upper bounded by

$$\text{Regret}(T) = \mathcal{O}\left( \frac{H}{\kappa} \sqrt{T \cdot \dim_{\text{FE}}\left(\mathcal{F}, G, \sqrt{1/T}\right) \cdot \beta} \right).$$

---

[6]Here and throughout our paper we considers $\pi_{\text{est}} = \pi^t$ for $Q$-type models. For $V$-type models, we instead consider $\pi_{\text{est}} = U(\mathcal{A})$ to be the uniform distribution over the action space. Such a representation of estimation policy allows us to unify the $Q$-type and $V$-type models in a single analysis.

We defer the proof of Theorem 11, together with a corollary for sample complexity analysis, to §C in the appendix. We observe that the regret bound of the OPERA algorithm is dependent on both the functional eluder dimension $\dim_{\mathrm{FE}}$ and the covering number of the induced DEF class $N_{\mathcal{L}}(\sqrt{1/T})$. In the special case when DEF is chosen as the Bellman error, the relation $\dim_{\mathrm{FE}}(\mathcal{F}, G, \sqrt{1/T}) = \dim_{\mathrm{BE}}(\mathcal{F}, \Pi, \sqrt{1/T})$ holds with $\Pi$ being the function class induced by $\{\pi_f, f \in \mathcal{F}\}$, and our Theorem 11 reduces to the regret bound in Jin et al. (2021) (Theorem 15). We will provide a detailed comparison between our framework and other related frameworks in §A when applied to different MDP models in the appendix.

### 4.3 IMPLICATION FOR SPECIFIC MDP INSTANCES

Here we focus on comparing our results applied to model-based RLs that are hardly analyzable in the model-free framework in §3.2. We demonstrate how OPERA can find near-optimal policies and achieve a state-of-the-art sample complexity under our new framework. Regret-bound analyses of linear mixture MDPs and several other MDP models can be found in §B in the appendix.

We highlight that Algorithm 1 not only provides a simple optimization-based scheme, recovers previous near-optimal algorithms in literature (Algorithms 2 and 4 in §E) when applied to specific MDP instances, but also reduces to a novel Algorithm 3 for low witness rank MDPs with improved sample complexity.

**Low Witness Rank.** We first provide a sample complexity result for the low Witness rank model structure. Let $|\mathcal{M}|$ and $|\mathcal{V}|$ be the cardinality of the model class[7] $\mathcal{M}$ and discriminator class $\mathcal{V}$, respectively, and $W_\kappa$ be the witness rank (Definition 28) of the model. We have the following sample complexity result for low Witness rank models.

**Corollary 12** (Finite Witness Rank). For an MDP model $M$ with finite witness rank structure and any fixed $\delta \in (0, 1)$, we choose $\beta = \mathcal{O}\left(\log(TH|\mathcal{M}||\mathcal{V}|/\delta)\right)$ in Algorithm 1. With probability at least $1 - \delta$, Algorithm 1 outputs an $\epsilon$-optimal policy $\pi_{\mathrm{out}}$ within $T = \widetilde{\mathcal{O}}\left(H^2|\mathcal{A}|W_\kappa\beta/(\kappa^2\epsilon^2)\right)$ trajectories.

Proof of Corollary 12 is delayed to §E.4.[8] Compared with previous best-known sample complexity result of $\widetilde{O}\left(H^3 W_\kappa^2 |\mathcal{A}| \log(T|\mathcal{M}||\mathcal{V}|/\delta)/(\kappa^2\epsilon^2)\right)$ due to Sun et al. (2019), our sample complexity is superior by a factor of $dH$ up to a polylogarithmic prefactor in model parameters.

**Kernel Nonlinear Regulator.** Now we turn to the implication of Theorem 11 for learning *KNR models*. We have the following regret bound result for KNR.

**Corollary 13** (KNR). For the KNR model in Eq. (3.3) and any fixed $\delta \in (0, 1)$, we choose $\beta = \mathcal{O}\left(\sigma^2 d_\phi d_s \log^2(TH/\delta)\right)$ in Algorithm 1. With probability at least $1 - \delta$, the regret is upper bounded by $\widetilde{\mathcal{O}}\left(H^2\sqrt{d_\phi T\beta}/\sigma\right)$.

We remark that neither the low BE dimension nor the Bellman Representability classes admit the KNR model with a sharp regret bound. Among earlier attempts, Du et al. (2021, §6) proposed to use a generalized version of Bilinear Classes to capture models including KNR, Generalized Linear Bellman Complete, and finite Witness rank. Nevertheless, their characterization requires imposing monotone transformations on the statistic and yields a suboptimal $\mathcal{O}(T^{3/4})$ regret bound. Our ABC class with low FE dimension is free of monotone operators, albeit that the coupling function for the KNR model is not of a bilinear form.

## 5 CONCLUSION AND FUTURE WORK

In this paper, we proposed a unified framework that subsumes *nearly* all Markov Decision Process (MDP) models in existing literature from model-based and model-free RLs. For the complexity analysis, we propose a new type of estimation function with the decomposable property for optimization-based exploration and use the functional eluder dimension with respect to an admissible Bellman characterization function as the complexity measure of our model class. In addition, we proposed a new sample-efficient algorithm, OPERA, which matches or improves the state-of-the-art sample complexity (or regret) results.

On the other hand, we notice that some MDP instances are not covered by our framework such as the $Q^*$ state-action aggregation, and the deterministic linear $Q^*$ models where only $Q^*$ has a linear structure. We leave it as a future work to include these MDP models.

---

[7]Hypothesis class reduces to model class (Sun et al., 2019) when restricted to model-based setting.

[8]The definition of witness rank adopts a $V$-type representation and hence we can only derive the sample complexity of our algorithm. For detailed discussion on the $V$-type cases, we refer readers to §D in the appendix.

ACKNOWLEDGEMENT

We thank the anonymous reviewers for their helpful comments. ZC, AY and QG are supported in part by the National Science Foundation CAREER Award 1906169 and IIS-2008981. MIJ is supported in part by the Mathematical Data Science program of the Office of Naval Research under grant number N00014-18-1-2764 and by the Vannevar Bush Faculty Fellowship program under grant number N00014-21-1-2941 and NSF grant IIS-1901252.

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

APPENDIX

The appendix is organized as follows. §A discusses the related work, providing comparisons with previous frameworks based on both coverage and sharpness of sample complexity. §B compares our regret bound and sample complexity on specific examples and discusses several additional examples including reactive POMDPs, FLAMBE, LQR, and the generalized linear Bellman complete model. §C proves the main results (Theorem 11 and Corollary 26 on sample complexity of OPERA). §D explains the $V$-type setting and the corresponding results. §E discusses the OPERA algorithm when being applied to special examples (linear mixture MDPs, low Witness rank MDPs, KNRs). §F details the delayed proofs of technical lemmas. §G details the proofs relevant to FE dimension.

## A    RELATED WORK

**Tabuler RL.**    Tabular RL considers MDPs with finite state space $\mathcal{S}$ and action space $\mathcal{A}$. This setting has been extensively studied (Auer et al., 2008; Dann & Brunskill, 2015; Brafman & Tennenholtz, 2002; Agrawal & Jia, 2017; Azar et al., 2017; Zanette & Brunskill, 2019; Zhang et al., 2020) and the minimax-optimal regret bound is proved to be $\widetilde{O}(\sqrt{H^2|\mathcal{S}||\mathcal{A}|T})$ (Jin et al., 2018; Domingues et al., 2021). The minimax optimal bounds suggests that the tabular RL is information-theoretically hard for large $|\mathcal{S}|$ and $|\mathcal{A}|$. Therefore, in order to deal with high-dimensional state-action space arose in many real-world applications, more advanced structural assumptions that enable function approximation are in demand.

**Complexity Measures for Statistical Learning.**    In classic statistical learning, a variety of complexity measures have been proposed to upper bound the sample complexity required for achieving a certain accuracy, including VC Dimension (Vapnik, 1999), covering number (Pollard, 2012), Rademacher Complexity (Bartlett & Mendelson, 2002), sequential Rademacher complexity (Rakhlin et al., 2010) and Littlestone dimension (Littlestone, 1988). However, for reinforcement learning, it is a major challenge to find such general complexity measures that can be used to analyze the sample complexity under a general framework.

**RL with Linear Function Approximation.**    A line of work studied the MDPs that can be represented as a linear function of some given feature mapping. Under certain completeness conditions, the proposed algorithms can enjoy sample complexity/regret scaling with the dimension of the feature mapping rather than $|\mathcal{S}|$ and $|\mathcal{A}|$. One such class of MDPs is linear MDPs (Jin et al., 2020; Wang et al., 2019; Neu & Pike-Burke, 2020), where the transition probability function and reward function are linear in some feature mapping over state-action pairs. Zanette et al. (2020a;b) studied MDPs under a weaker assumption called low inherent Bellman error, where the value functions are nearly linear w.r.t. the feature mapping. Another class of MDPs is linear mixture MDPs (Modi et al., 2020; Jia et al., 2020; Ayoub et al., 2020; Zhou et al., 2021b; Cai et al., 2020), where the transition probability kernel is a linear mixture of a number of basis kernels. The above paper assumed that feature vectors are known in the MDPs with linear approximation while Agarwal et al. (2020b) studied a harder setting where both the feature and parameters are unknown in the linear model.

**RL with General Function Approximation.**    Beyond the linear setting, a recent line of research attempted to unify existing sample-efficient approaches with general function approximation. Osband & Van Roy (2014) proposed an structural condition named eluder dimension. Wang et al. (2020) further proposed an efficient algorithm LSVI-UCB for general linear function classes with small eluder dimension. Another line of works proposed low-rank structural conditions, including Bellman rank (Jiang et al., 2017; Dong et al., 2020) and Witness rank (Sun et al., 2019). Yang et al. (2020) studied the MDPs with a structure where the action-value function can be represented by a kernel function or an over-parameterized neural network. Recently, Jin et al. (2021) proposed a complexity called Bellman eluder (BE) dimension. The RL problems with low BE dimension subsume the problems with low Bellman rank and low eluder dimension. Simultaneously Du et al. (2021) proposed Bilinear Classes, which can be applied to a variety of loss estimators beyond vanilla Bellman error, but with possibly worse sample complexity. Very recently, Foster et al. (2021) proposed Decision-Estimation Coefficient (DEC), which is a necessary and sufficient condition for sample-efficient interactive learning. To apply DEC to reinforcement learning, Foster et al. (2021) further proposed a RL class named Bellman Representability, which can be viewed as a generalization of the Bilinear Class.
In Table 1, we compare our ABC framework with other structural RL frameworks in terms of the model coverage and sample complexity.

| | Bilinear Class | Low BE Dimension | DEC and Bellman Representability | ABC Class (with Low FE Dimension) |
|---|---|---|---|---|
| Linear MDPs (Yang & Wang, 2019; Jin et al., 2020) | $d^3 H^4 / \epsilon^2$ | $d^2 H^2 / \epsilon^2$ | $d^3 H^3 / \epsilon^2$ | $d^2 H^2 / \epsilon^2$ |
| Linear Mixture MDPs (Modi et al., 2020) | $d^3 H^4 / \epsilon^2$ | ✘ | $d^3 H^3 / \epsilon^2$ | $d^2 H^2 / \epsilon^2$ |
| Bellman Rank (Jiang et al., 2017) | $d^2 H^5 |\mathcal{A}| / \epsilon^2$ | $d H^2 |\mathcal{A}| / \epsilon^2$ | $d^2 H^3 |\mathcal{A}| / \epsilon^2$ | $d H^2 |\mathcal{A}| / \epsilon^2$ |
| Eluder Dimension (Wang et al., 2020) | ✘ | $\dim_{\mathrm{E}} H^2 / \epsilon^2$ | $\dim_{\mathrm{E}}^2 H^3 / \epsilon^2$ | $\dim_{\mathrm{E}} H^2 / \epsilon^2$ |
| Witness Rank (Sun et al., 2019) | — | ✘ | — | $W_\kappa H^2 |\mathcal{A}| / \epsilon^2$ |
| Low Occupancy Complexity (Du et al., 2021) | $d^3 H^4 / \epsilon^2$ | $d^2 H^2 / \epsilon^2$ | $d^3 H^3 / \epsilon^2$ | $d^2 H^2 / \epsilon^2$ |
| Kernelized Nonlinear Regulator (Kakade et al., 2020) | — | ✘ | — | $d_\phi^2 d_s H^4 / \epsilon^2$ |
| Linear $Q^*/V^*$ (Du et al., 2021) | $d^3 H^4 / \epsilon^2$ | $d^2 H^2 / \epsilon^2$ | $d^3 H^3 / \epsilon^2$ | $d^2 H^2 / \epsilon^2$ |

Table 1: Comparison of sample complexity for different MDP models under different RL frameworks. "—" indicates that the original work of framework does not provide an explicit sample complexity result for that model (although can be computed in principle), "✘" indicates the model is not included in the framework for complexity analysis. For models with the linear structure on a $d$-dimensional space, we present the sample complexity in terms of $d$. For models with their own complexity measures, we use $W_\kappa$ to denote the witness rank, $\dim_{\mathrm{E}}$ the eluder dimension, $d_\phi$ the dimension of $\mathcal{H}$ in KNR and $d_s$ the dimension number of the state space of KNR. The dependency on $\rho$-covering number is deliberately ignored for Bellman rank, eluder dimension, and the witness rank.

## B ADDITIONAL EXAMPLES

In this section, we compare our work with other results in the literature in terms of regret bounds/sample complexity. First of all, as we mentioned earlier in §3 when taking DEF as $\ell_{h,f'}(o_h, f_{h+1}, g_h, v) = Q_{h,g}(s_h, a_h) - r_h - V_{h+1,f}(s_{h+1})$ the ABC function reduces to the average Bellman error, and our ABC framework recovers the low Bellman eluder dimension framework for all cases compatible with such an estimation function. On several model-free structures, our regret bound is equivalent to that of the GOLF algorithm (Jin et al., 2021). For example for linear MDPs, OPERA exhibits a $\widetilde{\mathcal{O}}(dH\sqrt{T})$ regret bound that matches the state-of-the-art result on linear function approximation provided in Zanette et al. (2020a). For low eluder dimension models, the dependency on the eluder dimension $d$ in our regret analysis is $\widetilde{\mathcal{O}}(\sqrt{d})$ while the dependency in Wang et al. (2020) is $\widetilde{\mathcal{O}}(d)$. Also, for models with low Bellman rank $d$, our sample complexity scales linearly in $d$ as in Jin et al. (2021) while complexity in Jiang et al. (2017) scales quadratically.

For model-based RL settings with linear structure that are not within the low BE dimension framework such as the linear mixture MDPs, our OPERA algorithm obtains a $d_{\mathrm{FE}} H \sqrt{T}$ regret bound and $d_{\mathrm{FE}}^2 H^2 / \epsilon^2$ sample complexity result. In comparison, Jia et al. (2020); Modi et al. (2020) proposed an UCRL-VTR algorithm on linear mixture MDPs with a $dH\sqrt{T}$ regret bound, and Zhou et al. (2021a) improves this result by $\sqrt{H}$ via a Bernstein-type bonus for exploration. The Bilinear Classes (Du et al., 2021) is a general framework that covers linear mixture MDPs as a special case. The sample complexity of the BiLin-UCB algorithm when constrained to linear mixture models is $d^3 H^4 / \epsilon^2$, which is $dH^2$ worse than that of OPERA in this work. In terms of lower bound, when specialized to linear mixture MDPs, our result of $\widetilde{\mathcal{O}}(dH\sqrt{T})$ matches the lower bound provided in Zhou et al. (2021a) up to a factor of $H^{1/2}$.

In the rest of this section, we compare on six additional examples: the linear $Q^*/V^*$ model (Du et al., 2021), the low occupancy complexity model (Du et al., 2021), kernel reactive POMDPs, FLAMBE/Feature Selection, Linear Quadratic Regulator, and finally Generalized Linear Bellman Complete. Moreover, we added a discussion on the $Q^*$ state-action aggregation model.

## B.1 Linear $Q^*/V^*$

The linear $Q^*/V^*$ model was proposed in Du et al. (2021). In addition to the linear structure of the optimal action-value function $Q^*$, we further assume linear structure of the optimal state-value function $V^*$. We formally define the linear $Q^*/V^*$ model as follows:

**Definition 14** (Linear $Q^*/V^*$, Definition 4.5 in Du et al. 2021). A linear $Q^*/V^*$ model satisfies for two Hilbert spaces $\mathcal{H}_1, \mathcal{H}_2$ and two given feature mappings $\phi(s, a) : \mathcal{S} \times \mathcal{A} \to \mathcal{H}_1, \psi(s') : \mathcal{S} \to \mathcal{H}_2$, there exist $w_h^* \in \mathcal{H}_1, \theta_h^* \in \mathcal{H}_2$ such that

$$Q_h^*(s, a) = \langle w_h^*, \phi(s, a) \rangle \qquad \text{and} \quad V_h^*(s') = \langle \theta_h^*, \psi(s') \rangle$$

for any $h \in [H]$ and $(s, a, s') \in \mathcal{S} \times \mathcal{A} \times \mathcal{S}$.

Suppose that $\mathcal{H}_1$ and $\mathcal{H}_2$ has dimension number $d_1$ and $d_2$, separately, Du et al. (2021) shows that linear $Q^*/V^*$ model belongs to the Bilinear Class with dimension $d = d_1 + d_2$ and BiLin-UCB algorithm achieves an $\widetilde{\mathcal{O}}(\frac{d^3 H^4}{\epsilon^2})$ sample complexity. On the other hand the sample complexity of OPERA is of $\widetilde{\mathcal{O}}(\frac{d^2 H^2}{\epsilon^2})$.

## B.2 Low Occupancy Complexity

The low occupancy complexity model assumes linearity on the state-action distribution and has been proposed in Du et al. (2021). We recap its definition formally as follows:

**Definition 15** (Low Occupancy Complexity, Definition 4.7 in Du et al. 2021). A low occupancy complexity model is an MDP $M$ satisfying for some hypothesis class $\mathcal{F}$, a Hilbert space $\mathcal{H}$ and feature mappings $\phi_h(\cdot, \cdot) : \mathcal{S} \times \mathcal{A} \to \mathcal{H}, \forall h \in [H]$ that there exists a function on hypothesis classes $\beta_h : \mathcal{F} \to \mathcal{H}$ such that

$$d^{\pi_f}(s_h, a_h) = \langle \beta_h(f), \phi_h(s_h, a_h) \rangle, \qquad \forall f \in \mathcal{F}, \quad \forall (s_h, a_h) \in \mathcal{S} \times \mathcal{A}.$$

Du et al. (2021) proved that the low occupancy complexity model belongs to the Bilinear Classes and has a sample complexity of $d^3 H^4 / \epsilon^2$ under the BiLin-UCB algorithm. In the meantime, the low occupancy complexity model admits an improved sample complexity of $d^2 H^2 / \epsilon^2$ under the OPERA algorithm.

## B.3 Kernel Reactive POMDPs

The Reactive POMDP (Krishnamurthy et al., 2016) is a partially observable MDP (POMDP) model that can be described by the tuple $(\mathcal{S}, \mathcal{A}, \mathcal{O}, \mathbb{T}, \mathbb{O}, r, H)$, where $\mathcal{S}$ and $\mathcal{A}$ are the state and action spaces respectively, $\mathcal{O}$ is the observation space, $\mathbb{T}$ is the transition matrix that maps each $(s, a) \in \mathcal{S} \times \mathcal{A}$ to a probability measure on $\mathcal{S}$ and determines the dynamics of the next state as $s_{h+1} \sim \mathbb{T}(\cdot \mid s_h, a_h)$, $\mathbb{O}$ is the emission measure that determines the observation $o_h \sim \mathbb{O}(\cdot \mid s_h)$ given current state $s_h$. The reactiveness of a POMDP refers to the property that the optimal value function $Q^*$ depends only on the current observation and action. In other words, for all $h$, there exists a $f_h^* : \mathcal{O} \times \mathcal{A} \to [0, 1]$ such that for any given trajectory $\tau_h = [o_1, a_1, \ldots, o_h]$ and $a_h$, we have

$$Q^*(\tau_h, a_h) = f_h^*(o_h, a_h).$$

Given the definition of a reactive POMDP, we define the kernel reactive POMDP (Jin et al., 2021) as follows:

**Definition 16** (Kernel Reactive POMDP). A kernel reactive POMDP is a reactive POMDP that satisfies for each $h \in [H]$ and a given seperable Hilbert space $\mathcal{H}$, there exist feature mappings $\phi_h : \mathcal{S} \times \mathcal{A} \to \mathcal{H}$ and $\psi_h : \mathcal{S} \to \mathcal{H}$ such that the transition matrix $\mathbb{T}_h(s' \mid s, a) = \langle \phi_h(s, a), \psi_h(s') \rangle_{\mathcal{H}}$ and $\psi$ is bounded in the sense that for any $V(\cdot) : \mathcal{S} \to [0, 1], \left\| \sum_{s' \in \mathcal{S}} V(s') \psi(s') \right\|_{\mathcal{H}} \leq 1$.

In Jin et al. (2021), the authors showed that the kernel reactive POMDP with vanilla estimation function $\ell_h(o_h, f_{h+1}, g_h, v) = Q_{h,g}(s_h, a_h) - r_h - V_{h+1,f}(s_{h+1})$ has $V$-type BE dimension bounded by the effective dimension. According to Proposition 34, the kernel reactive POMDP model also has low FE dimension bounded by the effective dimension.

## B.4 FLAMBE/Feature Selection

For FLAMBE/feature selection model firstly introduced in Agarwal et al. (2020b), similarity is shared with the linear MDP setting but the main difference lies in that the feature mappings are *unknown*. We formally define the feature selection model as follows:

**Definition 17** (Feature Selection). A low rank feature selection model is an MDP $M$ that satisfies for any $h \in [H]$ and a given Hilbert space $\mathcal{H}$, there exist unknown feature mappings $\mu_h^* : \mathcal{S} \to \mathcal{H}$ and $\phi^* : \mathcal{S} \times \mathcal{A} \to \mathcal{H}$ such that the transition probability satisfies:

$$\mathbb{P}_h(s' \mid s, a) = \mu_h^*(s')^\top \phi^*(s, a), \quad \forall (s, a, s') \in \mathcal{S} \times \mathcal{A} \times \mathcal{S}.$$

We consider the feature selection model with DEF $\ell_h(o_h, f_{h+1}, g_h, v) := Q_{h,g}(s_h, a_h) - r_h - V_{h+1,f}(s_{h+1})$. In Du et al. (2021) they have proved in Lemma A.1 that

$$\mathbb{E}_{s_h \sim \pi_g, a_h \sim \pi_f} \left[ Q_{h,f}(s_h, a_h) - r_h - V_{h+1,f}(s_{h+1}) \right] = \langle W_h(f), X_h(g) \rangle, \tag{B.1}$$

where

$$W_h(f) := \int_{s \in \mathcal{S}} \mu_h^*(s) \left( V_{h,f}(s) - r(s, \pi_f(s)) - \mathbb{E}_{s' \sim \mathbb{P}_h(\cdot | s, \pi_f(s))} \left[ V_{h+1,f}(s') \right] \right) ds,$$
$$X_h(g) := \mathbb{E}_{s_{h-1}, a_{h-1} \sim \pi_g} \left[ \phi^*(s_{h-1}, a_{h-1}) \right].$$

We note that Eq. (B.1) ensures condition (i) and (ii) in Definition 5 at the same time and the ABC of the feature selection setting has a bilinear structure that enables us to apply Proposition 35 to conclude low FE dimension.

## B.5 LINEAR QUADRATIC REGULATOR

In a linear quadratic regulator (LQR) model (Bradtke, 1992; Anderson & Moore, 2007; Dean et al., 2020), we consider the $d$ dimensional state space $\mathcal{S} \subseteq \mathbb{R}^d$ and $K$ dimensinal action space $\mathcal{A} \subseteq \mathbb{R}^K$. The transition dynamics of an LQR model can be written in matrix form so that the induced value function is quadratic (Jiang et al., 2017). We formally define the LQR model as follows:

**Definition 18** (Linear Quadratic Regulator). A linear quadratic regulator model is an MDP $M$ such that there exist unknown matrix $A \in \mathbb{R}^{d \times d}$, $B \in \mathbb{R}^{d \times K}$ and $Q \in \mathbb{R}^{d \times d}$ satisfying for $\forall h \in [H]$ and zero-centered random variables $\epsilon_h, \tau_h$ with $\mathbb{E}[\epsilon_h \epsilon_h^\top] = \Sigma$ and $\mathbb{E}[\tau_h^2] = \sigma^2$ that

$$s_{h+1} = As_h + Ba_h + \epsilon_h,$$
$$r_h = s_h^\top Q s_h + a_h^\top a_h + \tau_h.$$

The LQR model has been analyzed in Du et al. (2021) and proved to belong to the Bilinear Classes. Du et al. (2021) used the hypothesis class defined as

$$\mathcal{F}_h = \left\{ (C_h, \Lambda_h, O_h) : C_h \in \mathbb{R}^{K \times d}, \Lambda_h \in \mathbb{R}^{d \times d}, O_h \in \mathbb{R} \right\}_{h \in [H]}.$$

For each hypothesis in the class $f \in \mathcal{F}$, the corresponding policy and value function are

$$\pi_f(s_h) = C_{h,f} s_h, \quad V_{h,f}(s_h) = s_h^\top \Lambda_{h,f} s_h + O_{h,f}.$$

Under the above setting, we use the DEF for LQR $\ell_h(o_h, f_{h+1}, g_h, v) := Q_{h,g}(s_h, a_h) - r_h - V_{h+1,f}(s_{h+1})$ and Lemma A.4 in Du et al. (2021) showed that

$$\mathbb{E}_{s_h, a_h \sim \pi_g} \left[ Q_{h,f}(s_h, a_h) - r_h - V_{h+1,f}(s_{h+1}) \right] = \langle W_h(f), X_h(g) \rangle, \tag{B.2}$$

where

$$W_h(f) = \left[ \text{vec}(\Lambda_{h,f} - Q - C_{h,f}^\top C_{h,f} - (A + BC_{h,f})^\top \Lambda_{h+1,f}(A + BC_{h,f})), \right.$$
$$\left. O_{h,f} - O_{h+1,f} - \text{trace}(\Lambda_{h+1,f} \Sigma) \right],$$
$$X_h(f) = \left[ \text{vec}(\mathbb{E}_{s_h \sim \pi_f}[s_h s_h^\top]), 1 \right].$$

We note that Eq. (B.2) ensures condition (i) and (ii) in Definition 5 simultaneously and the ABC of the LQR model setting admits a bilinear structure that enables us to apply Proposition 35 and conclude low FE dimension.

## B.6 GENERALIZED LINEAR BELLMAN COMPLETE

Next we introduce the generalized linear Bellman complete model, showing that our ABC class with low FE dimension captures this model even without the monotone operator $\sqrt{x}$ used in Du et al. (2021).

**Definition 19** (Generalized Linear Bellman Complete). A generalized linear Bellman complete model consists of an inverse link function $\sigma : \mathbb{R} \to \mathbb{R}^+$ and a hypothesis class $\mathcal{F} := \left\{ \mathcal{F}_h = \sigma(\theta_h^\top \phi(s,a)) : \theta_h \in \mathcal{H}, ||\theta_h|| \leq R \right\}_{h \in [H]}$ such that for any $f \in \mathcal{F}$ and $\forall h \in [H]$ the Bellman completeness condition holds:

$$r(s,a) + \mathbb{E}_{s' \in \mathbb{P}_h} \max_{a' \in \mathcal{A}} \sigma(\theta_{h+1,f}^\top \phi(s',a')) \in \mathcal{H}_h.$$

By the choice of the hypothesis class $\mathcal{F}$, we know that there exists a mapping $\mathcal{T}_h : \mathcal{H} \to \mathcal{H}$ such that

$$\sigma\left(\mathcal{T}_h(\theta_{h+1,f})^\top \phi(s,a)\right) = r(s,a) + \mathbb{E}_{s' \in \mathbb{P}_h} \max_{a' \in \mathcal{A}} \sigma(\theta_{h+1,f}^\top \phi(s',a')). \tag{B.3}$$

We note that in Du et al. (2021) they choose a discrepancy function dependent on a discriminator function $v$. In this work, we choose a different estimation function that allows much simpler calculation and sharper sample complexity result. We let

$$\ell_h(o_h, f_{h+1}, g_h, v) := \sigma(\theta_{h,g}^\top \phi(s_h, a_h)) - r_h - \max_{a'} \theta_{h+1,f}^\top \phi(s_{h+1}, a').$$

By Eq. (B.3), it is easy to check that the above DEF satisfies the decomposable condition. Assuming $a \leq \sigma'(x) \leq b$, Lemma 6.2 in Du et al. (2021) has already shown the Bellman dominance property that

$$\left| \mathbb{E}_{s_h, a_h \sim \pi_f} \left[ Q_{h,f}(s_h, a_h) - r_h - V_{h+1,f}(s_{h+1}) \right] \right|$$
$$\leq b\sqrt{\left\langle \text{vec}\left((\theta_{h,f} - \mathcal{T}_h(\theta_{h+1,f})(\theta_{h,f} - \mathcal{T}_h(\theta_{h+1,f})^\top)\right), \text{vec}\left(\mathbb{E}_{s_h, a_h \sim \pi_f} \phi(s_h, a_h)\phi(s_h, a_h)^\top\right)\right\rangle}$$
$$= b\sqrt{\langle W_h(f), X_h(f) \rangle}.$$

Next, we illustrate that the Dominating Average EF condition holds in our framework. We have

$$\mathbb{E}_{s_h \sim \pi_g, a_h \sim \pi_{op}} || \mathbb{E}_{s_{h+1}} \left[ \ell_{h,g}(o_h, f_{h+1}, f_h, v) \mid s_h, a_h \right] ||^2$$
$$= \mathbb{E}_{s_h, a_h \sim \pi_g} || \sigma(\theta_{h,f}^\top \phi(s_h, a_h)) - \sigma(\mathcal{T}_h(\theta_{h+1,f})^\top \phi(s,a)) ||^2$$
$$\geq a\mathbb{E}_{s_h, a_h \sim \pi_g} \left( (\theta_{h,f} - \mathcal{T}_h(\theta_{h+1,f}))^\top \phi(s_h, a_h) \right)^2 \geq a \langle W_h(f), X_h(g) \rangle,$$

where

$$W_h(f) := \text{vec}\left((\theta_{h,f} - \mathcal{T}_h(\theta_{h+1,f})(\theta_{h,f} - \mathcal{T}_h(\theta_{h+1,f})^\top)\right),$$
$$X_h(f) := \text{vec}\left(\mathbb{E}_{s_h, a_h \sim \pi_f} \phi(s_h, a_h)\phi(s_h, a_h)^\top\right).$$

Analogous to the KNR case and the proof of Lemma 30, the aforementioned model with ABC function $\sqrt{\langle W_h(f), X_h(f) \rangle}$ has low FE dimension.

## B.7 $Q^*$ STATE-ACTION AGGREGATION

Finally, we consider the $Q^*$ state-action aggregation model (Dong et al., 2019), which cannot be covered by the bilinear classes (Du et al., 2021). We illustrate that our ABC framework covers this model with a nonlinear coupling function.

**Definition 20** ($Q^*$ state-action aggregation). We call an MDP $\mathcal{M}$ a $Q^*$ state-action model if there exists a $\xi(s,a) : \mathcal{S} \times \mathcal{A} \to \mathcal{B}$ such that for any state-action pairs $(s,a), (s',a') \in \mathcal{S} \times \mathcal{A}$, if $\xi(s,a) = \xi(s',a')$, then $Q^*(s,a) = Q^*(s',a')$. The dimension $d$ of the $Q^*$ state-action aggregation model is defined as the cardinality of $\mathcal{B}$, i.e., $d = |\mathcal{B}|$.

For each element $b \in \mathcal{B}$, we note that for all $(s,a) \in \mathcal{S} \times \mathcal{A}$ satisfying $\xi(s,a) = b$, they share the same value of $Q(s,a)$. For notational simplicity, we use $Q(b)$ to denote this common value for all $\xi(s,a) = b$. Moreover, we let $w_h^*$ be a $d$-dimensional vector of all aggregated values with $(w_h^*)_b = Q(b)$ and $Q^*(s,a)$ can be expressed as a linear function on the aggregated values:

$$Q_h^*(s,a) = \langle w_h^*, \psi(s,a) \rangle$$

where $\psi(s,a) : \mathcal{S} \times \mathcal{A} \to \{0,1\}^d$ is a one-hot vector satisfying

$$(\psi(s,a))_b = 1 \text{ when } \xi(s,a) = b \text{ and } (\psi(s,a))_b = 0 \text{ otherwise.}$$

For this case, we use the hypothesis class defined as

$$\mathcal{F}_h = \left\{ w_h \in \mathbb{R}^d \right\}_{h \in [H]},$$

where we take

$$Q_{h,g}(s,a) = \langle w_{h,g}, \psi(s,a) \rangle, \quad V_{h,g}(s') = \max_{a' \in \mathcal{A}} \langle w_{h,g}, \psi(s',a') \rangle.$$

We define the DEF as the Bellman residual and conclude that

$$
\begin{aligned}
\ell_h(o_h, g_{h+1}, g_h, v) &= Q_{h,g}(s_h, a_h) - r_h - V_{h,g}(s_{h+1}) \\
&= Q_{h,g}(s_h, a_h) - V_{h,g}(s_{h+1}) - [Q_h^*(s_h, a_h) - V_h^*(s_{h+1})] \\
&= \langle w_{h,g} - w_h^*, \psi(s,a) \rangle - \left[ \max_{a' \in \mathcal{A}} \langle w_{h,g}, \psi(s_{h+1}, a') \rangle - \max_{a' \in \mathcal{A}} \langle w_h^*, \psi(s_{h+1}, a') \rangle \right].
\end{aligned}
$$
(B.4)

Taking expectation over the distribution on $o_h$ given by $s_h, a_h \sim \pi_f$ and $s_{h+1} \sim \mathbb{P}_h$, we have

$$
\begin{aligned}
&\mathbb{E}_{s_h, a_h \sim \pi_f} [\ell_h(o_h, g_{h+1}, g_h, v)] \\
&= \left\langle w_{h,g} - w_h^*, \mathbb{E}_{s_h, a_h \sim \pi_f} \psi(s_h, a_h) \right\rangle \\
&\quad - \mathbb{E}_{s_h, a_h \sim \pi_f, s_{h+1} \sim \mathbb{P}_h} \left[ \max_{a' \in \mathcal{A}} \langle w_{h,g}, \psi(s_{h+1}, a') \rangle - \max_{a' \in \mathcal{A}} \langle w_h^*, \psi(s_{h+1}, a') \rangle \right].
\end{aligned}
$$
(B.5)

If we define $G(g,f)$ by the right hand side of (B.5), it is obvious that $G_{h,f^*}(\cdot, \cdot)$ with $\ell$ defined in (B.4) serves as the coupling function and the estimation function of an ABC class, respectively. In the mean time, Equation (B.5) cannot be expressed as a inner product of some $W_h(f), X_h(g)$ and thus cannot be covered by Du et al. (2021). Nevertheless, in our ABC framework, it is unclear if the FE dimension of $\mathcal{F}$ with respect to $G_{h,f^*}(\cdot, \cdot)$ can be bounded in a nontrivial way (i.e., $\ll |\mathcal{S}| \cdot |\mathcal{A}|$).

## C  PROOF OF MAIN RESULTS

In this section, we provide proofs of our main result Theorem 11 and a sample complexity corollary of the OPERA algorithm. Originated from proof techniques widely used in confidence bound based RL algorithms Russo & Van Roy (2013) our proof steps generalizes that of the GOLF algorithm Jin et al. (2021) but admits general DEF and ABCs. We prove our main result as follows:

### C.1  PROOF OF THEOREM 11

*Proof of Theorem 11.* We recall that the objective of an RL problem is to find an $\epsilon$-optimal policy satisfying $V_1^*(s_1) - V_1^{\pi^t}(s_1) \leq \epsilon$. Moreover, the regret of an RL problem is defined as $\sum_{t=1}^T V_1^*(s_1) - V_1^{\pi^t}(s_1)$, where $\pi^t$ is the output policy of an algorithm at time $t$.

**Step 1: Feasibility of $f^*$.**  First of all, we show that the optimal hypothesis $f^*$ lies within the confidence region defined by Eq. (4.1) with high probability:

**Lemma 21** (Feasibility of $f^*$). In Algorithm 1, given $\rho > 0$ and $\delta > 0$ we choose $\beta = c(\log(THN_\mathcal{L}(\rho)/\delta) + T\rho)$ for some large enough constant $c$. Then with probability at least $1 - \delta$, $f^*$ satisfies for any $t \in [T]$:

$$\max_{v \in \mathcal{V}} \left\{ \sum_{i=1}^{t-1} \|\ell_{h,f_h^i}(o_h^i, f_{h+1}^*, f_h^*, v)\|^2 - \inf_{g_h \in \mathcal{G}_h} \sum_{i=1}^{t-1} \|\ell_{h,f_h^i}(o_h^i, f_{h+1}^*, g_h, v)\|^2 \right\} \leq \mathcal{O}(\beta).$$

Lemma 21 shows that at each round of updates the optimal hypothesis $f^*$ stays in the confidence region depicted by Eq. (4.1) with radius $\mathcal{O}(\beta)$. We delay the proof of Lemma 21 to §F.2. Lemma 21 together with the optimization procedure Line 3 of Algorithm 1 implies an upper bound of $V_1^*(s_1) - V_1^{\pi^t}(s_1)$ with probability at least $1 - \delta$ as follows:

$$V_1^*(s_1) - V_1^{\pi^t}(s_1) \leq V_{1,f^t}(s_1) - V_1^{\pi^t}(s_1).$$
(C.1)

**Step 2: Policy Loss Decomposition.**  The second step is to upper bound the regret by the summation of Bellman errors. We apply the policy loss decomposition lemma in Jiang et al. (2017).

**Lemma 22** (Lemma 1 in Jiang et al. 2017). $\forall f \in \mathcal{H}$,

$$V_{1,f^t}(s_1) - V_1^{\pi^t}(s_1) = \sum_{h=1}^{H} \mathbb{E}_{s_h, a_h \sim \pi^t} \left[ Q_{h,f^t}(s_h, a_h) - r_h - V_{h+1,f^t}(s_{h+1}) \right].$$

Combining Lemma 22 with Eq. (C.1) we have the following:

$$V_1^*(s_1) - V_1^{\pi^t}(s_1) \le V_{1,f^t}(s_1) - V_1^{\pi^t}(s_1) = \sum_{h=1}^{H} \mathbb{E}_{s_h, a_h \sim \pi^t} \left[ Q_{h,f^t}(s_h, a_h) - r_h - V_{h+1,f^t}(s_{h+1}) \right].$$

(C.2)

**Step 3: Small ABC Value in the Confidence Region.** The third step is devoted to controlling the cumulative square of Admissible Bellman Characterization function. Recalling that the ABC function is upper bounded by the average DEF, where each feasible DEF stays in the confidence region that satisfies Eq. (4.1), we arrive at the following Lemma 23:

**Lemma 23.** In Algorithm 1, given $\rho > 0$ and $\delta > 0$ we choose $\beta = c(\log(TH\mathcal{N}_{\mathcal{L}}(\rho)/\delta) + T\rho)$ for some large enough constant $c$. Then with probability at least $1 - \delta$, for all $(t, h) \in [T] \times [H]$, we have

$$\sum_{i=1}^{t-1} \left( G_{h,f^*}(f^t, f^i) \right)^2 \le \mathcal{O}(\beta).$$

(C.3)

The proof of Lemma 23 makes use of Freedman's inequality (the precise version as in Agarwal et al. (2014)) and we delay the proof to §F.1.

**Step 4: Bounding the Cumulative Bellman Error by Functional Eluder Dimension.** In the fourth step, we aim to traslate the upper bound of the cumulative squared ABC at $(f^t, f^i)$ in Eq. (C.3) to an upper bound of the cumulative ABC at $(f^t, f^t)$. The following Lemma 24 is adapted from Lemma 41 in Jin et al. (2021) and Lemma 2 in Russo & Van Roy (2013). Lemma 24 controls the sum of ABC functions by properties of the functional eluder dimension.

**Lemma 24.** For a hypothesis class $\mathcal{F}$ and a given coupling function $G(\cdot, \cdot) : \mathcal{F} \times \mathcal{F} \to \mathcal{R}$ with bounded image space $|G(\cdot, \cdot)| \le C$. For any pair of sequences $\{f_t\}_{t \in [T]}, \{g_t\}_{t \in [T]} \subseteq \mathcal{F}$ satisfying for all $t \in [T]$, $\sum_{i=1}^{t-1} (G(f_t, g_i))^2 \le \beta$, the following inequality holds for all $t \in [T]$ and $\omega > 0$:

$$\sum_{i=1}^{t} |G(f_i, g_i)| \le \mathcal{O}\left( \sqrt{\dim_{\text{FE}}(\mathcal{F}, G, \omega)\beta t} + C \cdot \min\{t, \dim_{\text{FE}}(\mathcal{F}, G, \omega)\} + t\omega \right).$$

The proof of Lemma 24 is in §F.3.

**Step 5: Combining Everything.** In the final step, we combine the regret bound decomposition argument, the cumulative ABC bound, and the Bellman dominance property together to derive our final regret guarantee.

For any $h \in [H]$, we take $G(\cdot, \cdot) = G_{h,f^*}(\cdot, \cdot)$, $g_i = f^i$, $f_t = f^t$ and $\omega = \sqrt{\frac{1}{T}}$ in Lemma 24. By Eq. (C.3) in Lemma 23, we have for any $h \in [H]$ and $t \in [T]$,

$$\sum_{i=1}^{t} |G_{h,f^*}(f^i, f^i))| \le \mathcal{O}\left( \sqrt{\dim_{\text{FE}}(\mathcal{F}, G_{h,f^*}, \sqrt{1/T})\beta t} + C \cdot \min\{t, \dim_{\text{FE}}(\mathcal{F}, G_{h,f^*}, \sqrt{1/T})\} + \sqrt{t} \right)$$

$$\le \mathcal{O}\left( \sqrt{\dim_{\text{FE}}(\mathcal{F}, G_{h,f^*}, \sqrt{1/T})\beta t} \right).$$

We recall our choice of $\beta = c(\log(TH\mathcal{N}_{\mathcal{L}}(\rho)/\delta) + T\rho)$. Taking $\rho = \frac{1}{T}$, we have

$$\sum_{i=1}^{t} |G_{h,f^*}(f^i, f^i))| \le \mathcal{O}\left( \sqrt{\dim_{\text{FE}}\left(\mathcal{F}, G_{h,f^*}, \sqrt{1/T}\right) \log(TH\mathcal{N}_{\mathcal{L}}(1/T)/\delta) \cdot t} \right)$$

$$\le \mathcal{O}\left( \sqrt{\dim_{\text{FE}}\left(\mathcal{F}, G, \sqrt{1/T}\right) \log(TH\mathcal{N}_{\mathcal{L}}(1/T)/\delta) \cdot t} \right).$$

Combining this with property (ii) in Definition 5 and decomposition (C.2), we conclude our main result that with probability at least $1 - \delta$,

$$
\sum_{t=1}^{T} V_1^*(s_1) - V_1^{\pi^t}(s_1) \leq \frac{1}{\kappa} \sum_{t=1}^{T} \sum_{h=1}^{H} |G_{h,f^*}(f^t, f^t)|
$$

$$
\leq \mathcal{O}\left( \frac{H}{\kappa} \sqrt{T \cdot \dim_{\mathrm{FE}}(\mathcal{F}, G, \sqrt{1/T}) \log\left(TH\mathcal{N}_{\mathcal{L}}(1/T)/\delta\right)} \right).
$$

This completes the whole proof of Theorem 11. □

## C.2 Sample Complexity of OPERA

**Corollary 25** (Sample Complexity of OPERA). For an MDP $M$ with hypothesis classes $\mathcal{F}, \mathcal{G}$ that satisfies Assumption 1 and a Decomosable Estimation Function $\ell$ satisfying Assumption 7. If there exists an Admissible Bellman Characterzation $G$ with low functional eluder dimension. For any $\epsilon \in (0, 1]$, we choose $\beta = c \left( \log(TH\mathcal{N}_{\mathcal{L}} \left( \frac{\kappa^2 \epsilon^2}{\dim_{\mathrm{FE}}(\mathcal{F}, G, \frac{\kappa\epsilon}{H})H^2} \right)/\delta) + T \frac{\kappa^2 \epsilon^2}{\dim_{\mathrm{FE}}(\mathcal{F}, G, \frac{\kappa\epsilon}{H})H^2} \right)$ for some large enough constant $c$. For the on-policy case when $\pi_{\mathrm{op}} = \pi_{\mathrm{est}} = \pi^t$, with probability at least $1 - \delta$ Algorithm 1 outputs a $\epsilon$-optimal policy $\pi_{\mathrm{out}}$ within $T$ trajectories where

$$
T = \frac{\dim_{\mathrm{FE}}(\mathcal{F}, G, \frac{\kappa\epsilon}{H}) \log\left(TH\mathcal{N}_{\mathcal{L}} \left( \frac{\kappa^2 \epsilon^2}{\dim_{\mathrm{FE}}(\mathcal{F}, G, \frac{\kappa\epsilon}{H})H^2} \right)/\delta\right) H^2}{\kappa^2 \epsilon^2}.
$$

*Proof of Corollary 25.* By the policy loss decomposition (C.2), (C.3) in Lemma 23 and Lemma 24, we have that

$$
\frac{1}{T} \sum_{t=1}^{T} V_1^*(s_1) - V_1^{\pi^t}(s_1) \leq \frac{1}{\kappa T} \sum_{t=1}^{T} \sum_{h=1}^{H} \left|G_{h,f^*}(f^t, f^t)\right|
$$

$$
\leq \mathcal{O}\left( \frac{H}{\kappa} \sqrt{\dim_{\mathrm{FE}}(\mathcal{F}, G, \omega) \left( \frac{\log\left(TH\mathcal{N}_{\mathcal{L}}(\rho)/\delta\right)}{T} + \rho \right)} + \frac{H\omega}{\kappa} \right).
$$
(C.4)

Taking $\omega = \frac{\kappa\epsilon}{H}$ and $\rho = \frac{\kappa^2 \epsilon^2}{\dim_{\mathrm{FE}}(\mathcal{F}, G, \frac{\kappa\epsilon}{H})H^2}$, the above Eq. (C.4) becomes

$$
\frac{1}{T} \sum_{t=1}^{T} V_1^*(s_1) - V_1^{\pi^t}(s_1) \leq \mathcal{O}\left( \frac{H}{\kappa} \sqrt{\frac{\dim_{\mathrm{FE}}(\mathcal{F}, G, \frac{\kappa\epsilon}{H}) \log\left(TH\mathcal{N}_{\mathcal{L}}(\rho)/\delta\right)}{T}} + \epsilon \right).
$$

Taking

$$
T = \frac{\dim_{\mathrm{FE}}(\mathcal{F}, G, \frac{\kappa\epsilon}{H}) \log\left(TH\mathcal{N}_{\mathcal{L}}(\rho)/\delta\right) H^2}{\kappa^2 \epsilon^2}
$$

yields the desired result. □

## D $Q$-type and $V$-type Sample Complexity Analysis

In Definition 5, we note that there are two ways to calculate the ABC of an MDP model depending on the different choices of the operating policy $\pi_{\mathrm{op}}$. Specifically, if $\pi_{\mathrm{op}} = \pi_g$, we call it the $Q$-type ABC. Otherwise, if $\pi_{\mathrm{op}} = \pi_f$, we call it the $V$-type ABC. For example, when taking

$$
G_{h,f^*}(f, g) = \mathbb{E}_{s_h \sim \pi_g, a_h \sim \pi_g} [Q_{h,f}(s_h, a_h) - r(s_h, a_h) - V_{h+1,f}(s_{h+1})]
$$

the FE dimension of $G_{h,f^*}(f, g)$ recovers the $Q$-type BE dimension (Definition 8 in Jin et al. (2021). When taking

$$
G_{h,f^*}(f, g) = \mathbb{E}_{s_h \sim \pi_g, a_h \sim \pi_f} [Q_{h,f}(s_h, a_h) - r(s_h, a_h) - V_{h+1,f}(s_{h+1})]
$$

the FE dimension of $G_{h,f^*}(f, g)$ recovers the $V$-type BE dimension (Definition 20 in Jin et al. (2021). The algorithm for solving $Q$-type or $V$-type models slightly differs in the executing policy $\pi_{\mathrm{est}}$. We use $\pi_{\mathrm{est}} = \pi^t$ for $Q$-type models in Algorithm 1, while $\pi_{\mathrm{est}} = U(\mathcal{A})$ is the uniform distribution on action set for $V$-type models.

The $Q$-type characterization and the $V$-type characterization have respective applicable zones. For example, the reactive POMDP model belongs to ABC with low FE dimension with respect to $V$-type ABC while inducing large FE dimension with respect to $Q$-type ABC. On the contrary, the low inherent bellman error problem in Zanette et al. (2020a) is more suitable for using a $Q$-type characterization rather than a $V$-type characterization. For general RL models, we often prefer $Q$-type ABC because the sample complexity of $V$-type algorithms scales with the dimension of the action space $|\mathcal{A}|$. Due to the uniform executing policy, we will only be able to derive regret bound for $Q$-type characterizations, as is explained in Jin et al. (2021).

In §4 and §C, we have illustrated regret bound and sample complexity results for the $Q$-type cases where we let $\pi_{\mathrm{op}} = \pi_{\mathrm{est}} = \pi^t$ through Algorithm 1. In the following Corollary 26, we prove sample complexity result for $V$-type ABC models.

**Corollary 26.** For an MDP $M$ with hypothesis classes $\mathcal{F}, \mathcal{G}$ that satisfies Assumption 1 and a Decomposable Estimation Function $\ell$ satisfying Assumption 7. If there exists an Admissible Bellman Characterization $G$ with low functional eluder dimension. For any $\epsilon \in (0,1]$, if we choose $\beta = \mathcal{O}\left(\log(TH\mathcal{N}_{\mathcal{L}}(\rho)/\delta) + T\rho\right)$. For $V$-type models when $\pi_{\mathrm{op}} = \pi_{\mathrm{est}} = \pi^t$, with probability at least $1-\delta$ Algorithm 1 outputs a $\epsilon$-optimal policy $\pi_{\mathrm{out}}$ within $T = \frac{|\mathcal{A}| \dim_{\mathrm{FE}}(\mathcal{F},G,\kappa\epsilon/H) \log(TH\mathcal{N}_{\mathcal{L}}(\rho)/\delta) H^2}{\kappa^2 \epsilon^2}$ trajectories where $\rho = \frac{\kappa^2 \epsilon^2}{\dim_{\mathrm{FE}}(\mathcal{F},G,\frac{\kappa\epsilon}{H}) H^2}$.

*Proof of Corollary 26.* The proof of Corollary 26 basically follows the proof of Theorem 11 and Corollary 25. We again have feasibility of $f^*$ and policy loss decomposition. However, due to different sampling policy, the proof of Lemma 23 differs at Eq. (F.5). Instead, we have

$$
\sum_{i=1}^{t-1} \max_{v \in \mathcal{V}} \mathbb{E}_{s_h \sim \pi^i, a_h \sim \pi^t} \mathbb{E}_{s_{h+1}} \left[ X_i(h, f^t, v) \mid s_h, a_h \right]
$$
$$
= \sum_{i=1}^{t-1} \max_{v \in \mathcal{V}} \mathbb{E}_{s_h \sim \pi^i, a_h \sim U(\mathcal{A})} \frac{\mathbb{1}(a_h^i = \pi_f(s_h^i))}{1/|\mathcal{A}|} \mathbb{E}_{s_{h+1}} \left[ X_i(h, f^t, v) \mid s_h, a_h \right]
$$
$$
= \sum_{i=1}^{t-1} \max_{v \in \mathcal{V}} \mathbb{E}_{s_h \sim \pi^i, a_h \sim U(\mathcal{A})} \frac{\mathbb{1}(a_h^i = \pi_f(s_h^i))}{1/|\mathcal{A}|} ||\mathbb{E}_{s_{h+1}} \left[ \ell_{h,f^i}(o_h, f_{h+1}^t, f_h^t, v) \mid s_h, a_h \right] ||^2
$$
$$
\leq \mathcal{O}(|\mathcal{A}| \left( \beta + Rt\rho + R^2\iota \right)). \tag{D.1}
$$

Thus, Eq. (C.3) in Lemma 23 becomes

$$
\sum_{i=1}^{t-1} \left( G_{h,f^*}(f^t, f^i) \right)^2 \leq \mathcal{O}(|\mathcal{A}|\beta).
$$

The rest of the proof follow the proof of Corollary 25 with an additional $|\mathcal{A}|$ factor. By the policy loss decomposition (C.2) and Lemma 24, we have that

$$
\frac{1}{T} \sum_{t=1}^{T} V_1^*(s_1) - V_1^{\pi^t}(s_1) \leq \frac{1}{\kappa T} \sum_{t=1}^{T} \sum_{h=1}^{H} |G_{h,f^*}(f^t, f^t)|
$$
$$
\leq \mathcal{O}\left( \frac{H}{\kappa} \sqrt{|\mathcal{A}| \dim_{\mathrm{FE}}(\mathcal{F},G,\omega) \left( \frac{\log(TH\mathcal{N}_{\mathcal{L}}(\rho)/\delta)}{T} + \rho \right)} + \frac{H\omega}{\kappa} \right). \tag{D.2}
$$

Taking $\omega = \frac{\kappa\epsilon}{H}$ and $\rho = \frac{\kappa^2 \epsilon^2}{\dim_{\mathrm{FE}}(\mathcal{F},G,\frac{\kappa\epsilon}{H}) H^2}$, the above Eq. (D.2) becomes

$$
\frac{1}{T} \sum_{t=1}^{T} V_1^*(s_1) - V_1^{\pi^t}(s_1) \leq \mathcal{O}\left( \frac{H}{\kappa} \sqrt{\frac{|\mathcal{A}| \dim_{\mathrm{FE}}(\mathcal{F},G,\frac{\kappa\epsilon}{H}) \log(TH\mathcal{N}_{\mathcal{L}}(\rho)/\delta)}{T}} + \epsilon \right).
$$

Taking

$$
T = \frac{|\mathcal{A}| \dim_{\mathrm{FE}}(\mathcal{F},G,\frac{\kappa\epsilon}{H}) \log(TH\mathcal{N}_{\mathcal{L}}(\rho)/\delta) H^2}{\kappa^2 \epsilon^2}
$$

yields the desired result. $\qquad\square$

## E    PROOF FOR SPECIFIC EXAMPLES

In this section, we consider three specific examples: linear mixture MDPs, low Witness rank MDPs, and KNRs. We explains how our framework exhibits superior properties than other general frameworks on these three instances of MDPs. For reader's convenience, we summarize the conditions introduced in Items (i), (ii) in Definition 6 and also Items (i), (ii) in Definition 5, that are essential for any RL models to fit in our framework:

- **Decomposability:**

$$\ell_{h,f'}(o_h, f_{h+1}, g_h, v) - \mathbb{E}_{s_{h+1}} \left[ \ell_{h,f'}(o_h, f_{h+1}, g_h, v) \mid s_h, a_h \right] = \ell_{h,f'}(o_h, f_{h+1}, \mathcal{T}(f)_h, v).$$

- **Global Discriminator Optimality:**

$$||\mathbb{E}_{s_{h+1}} \left[ \ell_{h,f'}(o_h, f_{h+1}, f_h, v_h^*(f)) \mid s_h, a_h \right] || \geq ||\mathbb{E}_{s_{h+1}} \left[ \ell_{h,f'}(o_h, f_{h+1}, f_h, v) \mid s_h, a_h \right] ||.$$

- **Dominating Average EF:**

$$\max_{v \in \mathcal{V}} \mathbb{E}_{s_h \sim \pi_g, a_h \sim \pi_{\text{op}}} ||\mathbb{E}_{s_{h+1}} \left[ \ell_{h,g}(o_h, f_{h+1}, f_h, v) \mid s_h, a_h \right] ||^2 \geq \left( G_{h,f^*}(f, g) \right)^2.$$

- **Bellman Dominance:**

$$\kappa \cdot \left| \mathbb{E}_{s_h, a_h \sim \pi_f} \left[ Q_{h,f}(s_h, a_h) - r(s_h, a_h) - V_{h+1,f}(s_{h+1}) \right] \right| \leq |G_{h,f^*}(f, f)|.$$

### E.1    LINEAR MIXTURE MDPS

In this case, we choose $\mathcal{F}_h = \mathcal{G}_h = \{\theta_h \in \mathcal{H}\}$. Thus, the hypothesis classes $\mathcal{F}$ and $\mathcal{G}$ consist of the set of parameters $\theta_1, \ldots, \theta_H \in \mathcal{H}$. Moreover, for each hypothesis class $f = (\theta_{1,f}, \ldots, \theta_{H,f}) \in \mathcal{F}$, the value function with respect to $f$ satiafies for any $h \in [H]$ that

$$Q_{h,f}(s, a) = \theta_{h,f}^\top \left( \psi(s, a) + \phi_{V_{h+1,f}}(s, a) \right),$$

where $\phi_{V_{h+1,f}}(s, a) := \sum_{s' \in \mathcal{S}} \phi(s, a, s') V_{h+1,f}(s')$. It is natural to define the DEF by

$$\ell_{h,f'}(o_h, f_{h+1}, g_h, v) = \theta_{h,g}^\top \left[ \psi(s_h, a_h) + \phi_{V_{h+1,f'}}(s_h, a_h) \right] - r_h - V_{h+1,f'}(s_{h+1}).$$

If we use $\Phi_h^{t-1}$ to denote the matrix $\left( (\psi + \phi_{V_{h+1,f^1}})(s_h^1, a_h^1), \ldots, (\psi + \phi_{V_{h+1,f^{t-1}}})(s_h^{t-1}, a_h^{t-1}) \right)$ and $\mathbf{y}_h^{t-1}$ to denote the vector $\left( r_h - V_{h+1,f^1}(s_{h+1}^i), \ldots, r_h - V_{h+1,f^{t-1}}(s_{h+1}^{t-1}) \right)$, Eq. (4.1) in Algorithm 1 under linear mixture setting can be written in a matrix form as:

$$||\theta_{h,f}^\top \Phi_h^{t-1} - \mathbf{y}_h^{t-1}||^2 - \inf_\theta ||\theta^\top \Phi_h^{t-1} - \mathbf{y}_h^{t-1}||^2 \leq \beta. \tag{E.1}$$

Taking $\widehat{\theta}_{h,t} = \arg\min_\theta ||\theta^\top \Phi_h^{t-1} - \mathbf{y}_h^{t-1}||^2 = \left( \Phi_h^{t-1} \left( \Phi_h^{t-1} \right)^\top \right)^{-1} \Phi_h^{t-1} \left( \mathbf{y}_h^{t-1} \right)^\top$ and $\Sigma_h^{t-1} := \Phi_h^{t-1} \left( \Phi_h^{t-1} \right)^\top$, simple algebra yields

$$||\theta_{h,f}^\top \Phi_h^{t-1} - \mathbf{y}_h^{t-1}||^2 - \inf_\theta ||\theta^\top \Phi_h^{t-1} - \mathbf{y}_h^{t-1}||^2 = || \left( \theta_{h,f} - \widehat{\theta}_{h,t} \right)^\top \Phi_h^{t-1}||^2 = \left\| \theta_{h,f} - \widehat{\theta}_{h,t} \right\|_{\Sigma_h^{t-1}}^2, \tag{E.2}$$

So Algorithm 1 reduces to Algorithm 2.

In particular, the confidence region defined by Eq. (E.2) in Algorithm 2 is the same as the confidence region used in the upper confidence RL with value-targeted regression (UCRL-VTR) algorithm (Jia et al., 2020; Ayoub et al., 2020). While in UCRL-VTR, they perform *step-by-step* local optimization within the confidence region, resulting in a confidence bonus added upon the $Q$ value function, our Algorithm 2 follows a *global optimization* scheme, where the objective is the total expected return by following the optimal policy under the current hypothesis. The design principle of the *global optimization* is the same as the ELEANOR algorithm (Zanette et al., 2020a). In fact, the difference between UCRL-VTR with Algorithm 2 is analogous to the difference between LSVI-UCB (Jin et al., 2020) with ELEANOR (Zanette et al., 2020a).

Algorithm 2 exhibits a $dH\sqrt{T}$ regret bound and $d^2 H^2 / \epsilon^2$ sample complexity result, as will be shown later in this subsection. Compared with the $d^3 H^4 / \epsilon^2$ sample complexity in Du et al. (2021), our

---
**Algorithm 2** OPERA (linear mixture MDPs)
---
1: **Initialize**: $\mathcal{D}_h = \varnothing$ for $h = 1, \ldots, H$
2: **for** iteration $t = 1, 2, \ldots, T$ **do**
3:     Set $\pi^t := \pi_{f^t}$ where $f^t$ is taken as $\operatorname{argmax}_{f \in \mathcal{F}} Q_{1,f}(s_1, \pi_f(s_1))$ subject to

$$\widehat{\theta}_{h,t} = \left(\Phi_h^{t-1} \left(\Phi_h^{t-1}\right)^\top\right)^{-1} \Phi_h^{t-1} \left(\mathbf{y}_h^{t-1}\right)^\top, \qquad \left\|\theta_{h,f} - \widehat{\theta}_{h,t}\right\|_{\Sigma_h^{t-1}}^2 \leq \beta \quad \text{for all } h \in [H] \quad \text{(E.3)}$$

4:     For any $h \in [H]$, collect tuple $(r_h, s_h, a_h, s_{h+1})$ by executing $s_h, a_h \sim \pi^t$
5:     Augment $\mathcal{D}_h = \mathcal{D}_h \cup \{(r_h, s_h, a_h, s_{h+1})\}$
6: **end for**
7: **Output**: $\pi_{\text{out}}$ uniformly sampled from $\{\pi^t\}_{t=1}^T$

---

algorithm improves over the best-known results on general frameworks that subsumes linear mixture MDPs. We provide more comparisons on the linear mixture model in §B.

In terms of computation, assume that there exists a planning oracle for the optimization problem in Line 3 of Algorithm 2 that requires $B$ time complexity to solve. Then for each $t \in [T], h \in [H]$, the computational complexity of the rest of the algorithm is dominated by the computation of $\left(\Sigma_h^{t-1}\right)^{-1}$, and the total computational complexity would be $\mathcal{O}(BT + d^2 HT)$.

Next, we proceed to prove that a linear mixture MDP belongs to ABC class with low FE dimension.

*Proof of Proposition 8.* In the linear mixture model, we choose hypothesis class $\mathcal{F}_h = \mathcal{G}_h = \{\theta_h \in \mathcal{H}\}$, and DEF function

$$\ell_{h,f'}(o_h, f_{h+1}, g_h, v) = \theta_{h,g}^\top \left[\psi(s_h, a_h) + \sum_{s'} \phi(s_h, a_h, s')V_{h+1,f'}(s')\right] - r_h - V_{h+1,f'}(s_{h+1}).$$

(a) **Decomposability.** Taking expectation over $s_{h+1}$ and we obtain that

$$\mathbb{E}_{s_{h+1}}\left[\ell_{h,f'}(o_h, f_{h+1}, g_h, v) \mid s_h, a_h\right] = (\theta_{h,g} - \theta_h^*)^\top \left[\psi(s_h, a_h) + \sum_{s'} \phi(s_h, a_h, s')V_{h+1,f'}(s')\right].$$

Thus, we have

$$\ell_{h,f'}(o_h, f_{h+1}, g_h, v) - \mathbb{E}_{s_{h+1}}\left[\ell_{h,f'}(o_h, f_{h+1}, g_h, v) \mid s_h, a_h\right]$$
$$= (\theta_h^*)^\top \left[\psi(s_h, a_h) + \sum_{s'} \phi(s_h, a_h, s')V_{h+1,f'}(s')\right] - r_h - V_{h+1,f'}(s_{h+1})$$
$$= \ell_{h,f'}(o_h, f_{h+1}, f_h^*, v).$$

(b) **Global Discriminator Optimality** holds automatically since $\ell$ is independent of $v$.

(c) **Dominating Average EF.** We have the following inequality for linear mixture models:

$$\mathbb{E}_{s_h, a_h \sim \pi_g} \|\mathbb{E}\left[\ell_{h,g}(o_h, f_{h+1}, f_h, v) \mid s_h, a_h\right]\|^2$$
$$= \mathbb{E}_{s_h, a_h \sim \pi_g} \left((\theta_{h,f} - \theta_h^*)^\top \left[\psi(s_h, a_h) + \sum_{s'} \phi(s_h, a_h, s')V_{h+1,g}(s')\right]\right)^2$$
$$\geq \left((\theta_{h,f} - \theta_h^*)^\top \mathbb{E}_{s_h, a_h \sim \pi_g} \left[\psi(s_h, a_h) + \sum_{s'} \phi(s_h, a_h, s')V_{h+1,g}(s')\right]\right)^2. \quad \text{(E.4)}$$

(d) **Bellman Dominance.** On the other hand, we know that

$$\mathbb{E}_{s_h, a_h \sim \pi_f}\left[Q_{h,f}(s_h, a_h) - r_h - V_{h+1,f}(s_{h+1})\right]$$
$$= \mathbb{E}_{s_h, a_h \sim \pi_f} (\theta_{h,f} - \theta_h^*)^\top \left[\psi(s_h, a_h) + \sum_{s'} \phi(s_h, a_h, s')V_{h+1,f}(s')\right]$$
$$= (\theta_{h,f} - \theta_h^*)^\top \mathbb{E}_{s_h, a_h \sim \pi_f} \left[\psi(s_h, a_h) + \sum_{s'} \phi(s_h, a_h, s')V_{h+1,f}(s')\right]. \quad \text{(E.5)}$$

---

**Algorithm 3** OPERA (Low Witness Rank MDPs)

---

1: **Initialize**: $\mathcal{D}_h = \varnothing$ for $h = 1, \ldots, H$
2: **for** iteration $t = 1, 2, \ldots, T$ **do**
3:     Set $\pi^t := \pi_{f^t}$ where $f^t$ is taken as $\text{argmax}_{f \in \mathcal{F}} Q_{1,f}(s_1, \pi_f(s_1))$ subject to

$$\max_{v \in \mathcal{V}} \left\{ \sum_{i=1}^{t-1} \left( \mathbb{E}_{\widetilde{s} \sim f_h} v(s_h^i, a_h^i, \widetilde{s}) - v(s_h^i, a_h^i, s_{h+1}^i) \right)^2 \right.$$

$$\left. - \inf_{g_h \in \mathcal{G}_h} \sum_{i=1}^{t-1} \left( \mathbb{E}_{\widetilde{s} \sim g_h} v(s_h^i, a_h^i, \widetilde{s}) - v(s_h^i, a_h^i, s_{h+1}^i) \right)^2 \right\} \leq \beta \quad \text{for all } h \in [H] \quad \text{(E.7)}$$

4:     For any $h \in [H]$, collect tuple $(r_h, s_h, a_h, s_{h+1})$ by rolling in $s_h \sim \pi^t$ and executing $a_h \sim U(\mathcal{A})$
5:     Augment $\mathcal{D}_h = \mathcal{D}_h \cup \{(r_h, s_h, a_h, s_{h+1})\}$
6: **end for**
7: **Output**: $\pi_{\text{out}}$ uniformly sampled from $\{\pi^t\}_{t=1}^T$

---

(e) **Low FE Dimension.** Observe from Eqs. (E.4) and (E.5) that we can choose ABC function of an linear mixture MDP as

$$G_{h,f^*}(f, g) := (\theta_{h,f} - \theta_h^*)^\top \mathbb{E}_{s_h, a_h \sim \pi_g} \left[ \psi(s_h, a_h) + \sum_{s'} \phi(s_h, a_h, s') V_{h+1,g}(s') \right]. \quad \text{(E.6)}$$

The next Lemma 27 proves that the FE dimension of $\mathcal{F}$ with respect to the coupling function $G_{h,f^*}(f, g)$ is less than the effective dimension $d$ of the parameter space $\mathcal{H}$.

**Lemma 27.** The linear mixture MDP model has FE dimension $\leq \widetilde{\mathcal{O}}(d)$ with respect to the ABC defined in (E.6).

We prove Lemma 27 in §G.

Thus, we conclude our proof of Proposition 8. $\qquad\qquad\qquad\qquad\qquad\qquad\qquad\qquad\qquad$ $\square$

From the above Proof of Proposition 8, we see that linear mixture MDPs perfectly fit our framework. We apply Theorem 11 and Corollary 25 to linear mixture MDPs and conclude directly that Algorithm 2 has a regret upper bound of $dH\sqrt{T}$ together with a sample complexity upper bound of $d^2 H^2/\epsilon^2$, matching the best-known results that uses a Hoeffding-type bonus for exploration.

## E.2   LOW WITNESS RANK MDPS

In this subsection, we provide a novel method for solving low Witness rank MDPs as a direct application of the OPERA algorithm. The witness rank is an important model-based assumption that covers several structural models including the factored MDPs (Kearns, 1998). Also, all models with low Bellman rank structure belong to the class of low Witness rank models while the opposite does not hold (Sun et al., 2019). Although the witness rank models can be solved in a model-free manner, model-free algorithms cannot find near-optimal solutions of general witness rank models in polynomial time. Meanwhile, existing frameworks (Sun et al., 2019; Du et al., 2021) with an efficient algorithm does not exhibit sharp sample complexity results. We recall that in low Witness rank settings, hypotheses on model-based parameters (transition kernel and reward function) are made. Based on this, there are two recent lines of related approaches. Sun et al. (2019) first proposed an algorithm that eliminates candidate models with high estimated witness model misfits. On the other hand, Du et al. (2021) proposed a general algorithmic framework that would imply an optimization-based algorithm on low Witness rank models.

The following definition is a generalized version of the witness rank in Sun et al. (2019), where we require the discriminator class $\mathcal{V}$ to be *complete*, meaning that the assemblage of functions by taking the value at $(s, a)$ from different functions also belongs to $\mathcal{V}$.

**Definition 28** (Witness Rank). For an MDP $M$, a given symmetric and complete discriminator class $\mathcal{V} = \{\mathcal{V}_h\}_{h \in [H]}, \mathcal{V}_h \subset \mathcal{S} \times \mathcal{A} \times \mathcal{S} \mapsto \mathbb{R}$ and a hypothesis class $\mathcal{F}$, we define the Witness rank of $M$ as the smallest $d$ such that for any two hypotheses $f, g \in \mathcal{F}$, there exist two mappings $X_h : \mathcal{F} \to \mathbb{R}^d$

and $W_h : \mathcal{F} \to \mathbb{R}^d$ and a constant $\kappa \in (0, 1]$, the following inequalities hold for all $h \in [H]$:

$$\max_{v \in \mathcal{V}_h} \mathbb{E}_{s_h \sim \pi_f, a_h \sim \pi_g} \left[ \mathbb{E}_{\widetilde{s} \sim g_h} v(s_h, a_h, \widetilde{s}) - \mathbb{E}_{\widetilde{s} \sim \mathbb{P}_h} v(s_h, a_h, \widetilde{s}) \right] \geq \langle W_h(g), X_h(f) \rangle, \qquad \text{(E.8)}$$

$$\kappa \cdot \mathbb{E}_{s_h \sim \pi_f, a_h \sim \pi_g} \left[ \mathbb{E}_{\widetilde{s} \sim g_h} V_{h+1,g}(\widetilde{s}) - \mathbb{E}_{\widetilde{s} \sim \mathbb{P}_h} V_{h+1,g}(\widetilde{s}) \right] \leq \langle W_h(g), X_h(f) \rangle. \qquad \text{(E.9)}$$

We prove an improved sample complexity result over existing literature and illustrate the differences in design scheme of our algorithm. We present the pseudocode in Algorithm 3. Note that in Eq. (E.7), we replace the DEF in Eq. (4.1) by (3.2). Next, we elaborate the design scheme of our algorithm in comparison with Sun et al. (2019) and Du et al. (2021). Note that the DEF $\mathbb{E}_{\widetilde{s} \sim g_h} v(s_h, a_h, \widetilde{s}) - v(s_h, a_h, s_{h+1})$ is similar with the discrepancy function used in Du et al. (2021) except for an importance sampling factor. Moreover, after taking sup over discriminator functions, the expected DEF equals the witnessed model misfit in Sun et al. (2019). Although Du et al. (2021) did not explicitly give an algorithm for witness rank, we observe some general differences between OPERA and BiLin-UCB (Du et al., 2021). The confidence region used in Algorithm 3 (simplified version for comparison) is $\sum_i [(\ell_f^i)^2 - \inf_g (\ell_g^i)^2] \leq \beta$ centered at the optimal hypothesis, while the confidence region used in BiLin-UCB is $\sum_i \left( \frac{1}{m} \sum_{j \leq m} \ell_f^{i,(j)} \right)^2 \leq \beta'$ that bound an estimate of $\ell$ centered at 0. Similarly as in BiLin-UCB, Sun et al. (2019) also attempts to bound a batched estimate of $\ell$. Their algorithm constantly eliminates out of range models, enforcing small witness model misfit on prior distributions. The analysis in Sun et al. (2019) and Du et al. (2021), however, does not enforce the additional assumption on the discriminator class; we obtain a sharper sample complexity as in Corollary 12.

If we assume that there exists a planning oracle for solving the optimization problem in Line 3 of Algorithm 3 with $B$ time complexity. The computation in the rest of the algorithm is dependent on the structure of discriminator class $\mathcal{V}$ and the hypothesis class $\mathcal{G}$. We omit the discussion here as the planning oracle with a total computational complexity of $\mathcal{O}(BT)$ is usually the dominating term.

In the forthcoming, we prove that low Witness rank MDPs belongs to ABC class with low FE dimension.

*Proof of Proposition 9.* In the low Witness rank model, we choose hypothesis class $\mathcal{F}_h = \mathcal{G}_h = \mathcal{M}$, and DEF function

$$\ell_h(o_h, f_{h+1}, g_h, v) = \mathbb{E}_{\widetilde{s} \sim g_h} v(s_h, a_h, \widetilde{s}) - v(s_h, a_h, s_{h+1}). \qquad \text{(E.10)}$$

Without loss of generality, we assume that the discriminator class $\mathcal{V}$ is rich enough in the sense that if $\forall s, a \in \mathcal{S} \times \mathcal{A}$, $v_{s,a}(\cdot, \cdot, \cdot) \in \mathcal{V}$, then $v(s, a, s') := v_{s,a}(s, a, s') \in \mathcal{V}$ (if not, we can use a rich enough $\mathcal{V}'$ induced by $\mathcal{V}$), an assumption generally satisfied by common discriminator classes. For example, Total variation, Exponential family, MMD, Factored MDP in Sun et al. (2019) all use a rich enough discriminator class. Also, if $\mathcal{V} = \{v : \|v\|_\infty \leq c\}$ for some absolute constant $c$, the function class is rich enough.

(a) **Decomposability.** Taking expectation over $s_{h+1}$ of Eq. (E.10) and we obtain that

$$\mathbb{E}_{s_{h+1}} \left[ \ell_h(o_h, f_{h+1}, g_h, v) \mid s_h, a_h \right] = \mathbb{E}_{\widetilde{s} \sim g_h} v(s_h, a_h, \widetilde{s}) - \mathbb{E}_{\widetilde{s} \sim \mathbb{P}_h} v(s_h, a_h, \widetilde{s}). \qquad \text{(E.11)}$$

Thus, we have

$$\ell_h(o_h, f_{h+1}, g_h, v) - \mathbb{E}_{s_{h+1}} \left[ \ell_h(o_h, f_{h+1}, g_h, v) \mid s_h, a_h \right] = \mathbb{E}_{\widetilde{s} \sim \mathbb{P}_h} v(s_h, a_h, \widetilde{s}) - v(s_h, a_h, s_{h+1})$$
$$= \ell_h(o_h, f_{h+1}, f_h^*, v).$$

(b) **Global Discriminator Optimality.** Eq. (E.11) implies that

$$\mathbb{E}_{s_{h+1}} \left[ \ell_h(o_h, f_{h+1}, f_h) \mid s_h, a_h \right] = \int v(s_h, a_h, s) \left( f_h(s \mid s_h, a_h) - \mathbb{P}_h(s \mid s_h, a_h) \right) ds.$$

We define $v_h^*(f)(s, a, s') = v_{s,a}(s, a, s')$ where

$$v_{s,a} := \arg\max_{v \in \mathcal{V}} \int v(s, a, \widetilde{s}) \left( f_h(\widetilde{s} \mid s, a) - \mathbb{P}_h(\widetilde{s} \mid s, a) \right) d\widetilde{s}.$$

It is easy to verify that $v_h^*(f)$ satisfies for all $h \in [H]$ and $(s_h, a_h) \in \mathcal{S} \times \mathcal{A}$,

$$\mathbb{E}_{s_{h+1}} \left[ \ell_h(o_h, f_{h+1}, f_h, v_h^*(f)) \mid s_h, a_h \right] \geq \mathbb{E}_{s_{h+1}} \left[ \ell_h(o_h, f_{h+1}, f_h, v) \mid s_h, a_h \right].$$

Finally, the symmetry of $\mathcal{V}$ concludes the global discriminator optimality.

(c) **Dominating Average EF.** We have the following inequality for low Witness rank model:

$$
\max_{v \in \mathcal{V}} \mathbb{E}_{s_h \sim \pi_g, a_h \sim \pi_f} || \mathbb{E} \left[ \ell_h(o_h, f_{h+1}, f_h, v) \mid s_h, a_h \right] ||^2
$$

$$
= \max_{v \in \mathcal{V}} \mathbb{E}_{s_h \sim \pi_g, a_h \sim \pi_f} \left( \mathbb{E}_{\widetilde{s} \sim f_h} v(s_h, a_h, \widetilde{s}) - \mathbb{E}_{\widetilde{s} \sim \mathbb{P}_h} v(s_h, a_h, \widetilde{s}) \right)^2
$$

$$
\geq \left( \max_{v \in \mathcal{V}} \mathbb{E}_{s_h \sim \pi_g, a_h \sim \pi_f} \left[ \mathbb{E}_{\widetilde{s} \sim f_h} v(s_h, a_h, \widetilde{s}) - \mathbb{E}_{\widetilde{s} \sim \mathbb{P}_h} v(s_h, a_h, \widetilde{s}) \right] \right)^2 \overset{(i)}{\geq} \langle W_h(f), X_h(g) \rangle^2 .
$$
(E.12)

where the last inequality (i) follows Definition 28 of witness rank.

(d) **Bellman Dominance.** On the other hand, by Definition 28 we know that

$$
\kappa \cdot \mathbb{E}_{s_h, a_h \sim \pi_f} \left[ Q_{h,f}(s_h, a_h) - r_h - V_{h+1,f}(s_{h+1}) \right] \leq \langle W_h(f), X_h(f) \rangle .
$$
(E.13)

(e) **Low FE Dimension.** We see from Eq. (E.12) and (E.13) that we can choose ABC function with low Witness rank RL model as

$$
G_{h,f^*}(f, g) := \langle W_h(f), X_h(g) \rangle .
$$
(E.14)

The next Lemma 29 proves that the FE dimension of $\mathcal{F}$ with respect to the coupling function $G_{h,f^*}(f, g)$ is less than the dimension $W_\kappa$ of the witness model.

**Lemma 29.** The low Witness rank MDP model has FE dimension $\leq \widetilde{\mathcal{O}}(W_\kappa)$ with respect to the ABC defined in (E.14).

We prove Lemma 29 in §G.

Thus, we conclude our proof of Proposition 9. $\qquad\square$

By Proposition 9 we can straightforwardly derive the sample complexity by applying Corollary 26. For better understanding of the context, we present a complete proof of the sample complexity result of witness rank model in §E.4.

### E.3 KERNELIZED NONLINEAR REGULATOR

In the KNR setting introduced in §3.2, the norm of $s_{h+1}$ might be arbitrarily large if the random vector $\epsilon_{h+1}$ is large in magnitude. On the contrary, our framework requires the boundedness of the DEF. To resolve this issue, we note the tail bound of one-dimensional Gaussian distribution indicates that for any given positive $x$:

$$
e^{x^2/2} \int_x^\infty e^{-t^2/2} dt \leq e^{x^2/2} \int_x^\infty \frac{t}{x} e^{-t^2/2} dt = \frac{1}{x}.
$$

Thus, for $TH$ i.i.d. $\mathbb{R}^{d_s}$-valued random vectors $\epsilon_h^t \sim \mathcal{N}(0, \sigma^2 I)$ and a fixed $\delta \in (0, 1)$, there exists an event $\mathcal{B}$ with $\mathbb{P}(\mathcal{B}) \geq 1 - \delta$ such that $\|\epsilon_h^t\|_\infty \leq \mathcal{O}\left( \sigma \sqrt{\log(THd_s/\delta)} \right)$ holds on event $\mathcal{B}$.

We first provide the application of OPERA on the KNR model, the algorithm is written in Algorithm 4. Note that by similar algebra as in Eq. (E.2), the confidence set (E.15) is equivalent to

$$
\|(U_{h,f} - \widehat{U}_{h,f})(\Sigma_h^{t-1})^{1/2}\|^2 \leq \beta,
$$

where $\Sigma_h^{t-1} := \Phi_h^{t-1}(\Phi_h^{t-1})^\top$ and $\widehat{U}_{h,f}$ is the optimal solution to the least square problem $\arg\min_U \sum_{i=1}^{t-1} \|U\phi(s_h^i, a_h^i) - s_{h+1}^i\|^2$. The OPERA algorithm reduces to the LC$^3$ algorithm in Kakade et al. (2020) except that LC$^3$ is under a homogeneous setting. The only difference between Algorithm 4 and LC$^3$ is that in Eq. (E.15), LC$^3$ sums over $t$ and $H$ and we can only sum over $t$ because of the inhomogeneous setting.

Bringing in the choice of $\beta$ in Corollary 13 yields a regret bound of $\widetilde{\mathcal{O}}\left( \sqrt{d_\phi^2 d_s H^4 T} \right)$. In comparison, LC$^3$ in Kakade et al. (2020) has a regret bound of $\widetilde{\mathcal{O}}\left( \sqrt{d_\phi(d_s + d_\phi)H^3 T} \right)$. The improved factor of $\sqrt{H}$ is due to the reduction from the time-inhomogeneous setting to the time-homogeneous setting. Thus, our regret bound matches the state-of-the-art result on KNR instances (Kakade et al., 2020) regarding the dependencies on $d_\phi, d_s, H$. However, $d_\phi^2 d_s$ in our result is slightly looser than

---

**Algorithm 4** OPERA (kernelized nonlinear regulator)

---

1: **Initialize**: $\mathcal{D}_h = \varnothing$ for $h = 1, \ldots, H$
2: **for** iteration $t = 1, 2, \ldots, T$ **do**
3:   Set $\pi^t := \pi_{f^t}$ where $f^t$ is taken as $\mathrm{argmax}_{f \in \mathcal{F}} Q_{1,f}(s_1, \pi_f(s_1))$ subject to

$$\sum_{i=1}^{t-1} ||U_{h,f}\phi(s_h^i, a_h^i) - s_{h+1}^i||^2 - \inf_{g_h \in \mathcal{G}_h} \sum_{i=1}^{t-1} ||U_{h,g}\phi(s_h^i, a_h^i) - s_{h+1}^i||^2 \le \beta \quad \text{for all } h \in [H]$$

(E.15)

4:   For any $h \in [H]$, collect tuple $(r_h, s_h, a_h, s_{h+1})$ by executing $s_h, a_h \sim \pi^t$
5:   Augment $\mathcal{D}_h = \mathcal{D}_h \cup \{(r_h, s_h, a_h, s_{h+1})\}$
6: **end for**
7: **Output**: $\pi_{\mathrm{out}}$ uniformly sampled from $\{\pi^t\}_{t=1}^T$

---

$d_\phi(d_s + d_\phi)$ in Kakade et al. (2020) and can be possibly improved by instance-specific analysis of KNR. Similar to the linear mixture MDP case, if we assume that there exists a planning oracle for the optimization problem in Line 3 of Algorithm 4 that requires $B$ time complexity to solve, the rest of the algorithm can be solved efficiently in $\mathcal{O}(d_\phi(d_\phi + d_s)HT)$ time complexity. So the total computational complexity of Algorithm 4 is $\mathcal{O}(BT + d_\phi(d_\phi + d_s)HT)$.

In addition, we would like to remark that we can adapt algorithms from the optimal control literature such as MPPI (Williams et al., 2015) and DMDMPC (Wagener et al., 2019) to solve the optimization problem in Line 3 of Algorithm 4. This approach has been used in Kakade et al. (2020), where they designed the LC$^3$ algorithm for solving KNRs. In particular, they leveraged the MPPI algorithm for the planning oracle and provided rich empirical results.

*Proof of Proposition 10.* In the KNR model, we choose hypothesis class $\mathcal{F}_h = \mathcal{G}_h = \{U \in \mathcal{H} \to \mathbb{R}^{d_s} : ||U|| \le R\}$, and DEF function

$$\ell_h(o_h, f_{h+1}, g_h, v) = U_{h,g}\phi(s_h, a_h) - s_{h+1}.$$

(a) **Decomposability.** Taking expectation over $s_{h+1}$ and we obtain that

$$\mathbb{E}_{s_{h+1}}[\ell_h(o_h, f_{h+1}, g_h, v) \mid s_h, a_h] = (U_{h,g} - U_h^*)\phi(s_h, a_h).$$

Thus, we have

$$\ell_h(o_h, f_{h+1}, g_h, v) - \mathbb{E}_{s_{h+1}}[\ell_h(o_h, f_{h+1}, g_h, v) \mid s_h, a_h] = U_h^*\phi(s_h, a_h) - s_{h+1} = \ell_h(o_h, f_{h+1}, f_h^*, v).$$

(b) **Global Discriminator Optimality** holds automatically since $\ell$ is independent of $v$.

(c) **Dominating Average EF.** We have the following inequality for the KNR model:

$$\mathbb{E}_{s_h, a_h \sim \pi_g}||\mathbb{E}[\ell_h(o_h, f_{h+1}, f_h, v) \mid s_h, a_h]||^2 = \mathbb{E}_{s_h, a_h \sim \pi_g}||(U_{h,f} - U_h^*)\phi(s_h, a_h)||^2. \quad \text{(E.16)}$$

(d) **Bellman Dominance.** On the other hand, we know that

$$\mathbb{E}_{s_h, a_h \sim \pi_f}[Q_{h,f}(s_h, a_h) - r_h - V_{h+1,f}(s_{h+1})] \le \frac{2H}{\sigma}\mathbb{E}_{s_h, a_h \sim \pi_f}||(U_{h,f} - U_h^*)\phi(s_h, a_h)||_2.$$

(E.17)

(e) **Low FE Dimension.** We see from Eqs. (E.16) and (E.17) that we can choose ABC function of an linear mixture MDP as

$$G_{h,f^*}(f, g) := \sqrt{\mathbb{E}_{s_h, a_h \sim \pi_g}||(U_{h,f} - U_h^*)\phi(s_h, a_h)||^2}, \quad \text{(E.18)}$$

and KNR has an ABC with $\kappa = \frac{\sigma}{2H}$. The next Lemma 30 proves that the FE dimension of $\mathcal{F}$ with respect to the coupling function $G_{h,f^*}(f, g)$ can be controlled by $d_\phi$:

**Lemma 30.** The KNR model has FE dimension $\le \widetilde{\mathcal{O}}(d_\phi)$ with respect to the ABC defined in (E.18).

We prove Lemma 30 in §G.

Thus, we conclude our proof of Proposition 10. $\qquad\qquad\square$

### E.4 PROOF OF COROLLARY 12

In this subsection, we provide sample complexity guarantee for models with low Witness rank. In the main text in §4.3 we presented our Corollary 12 for $\mathcal{M}$ and $\mathcal{V}$ with finite cardinality for convenience of comparison with previous works. Here, we prove general result for model class $\mathcal{M}$ and discriminator class $\mathcal{V}$ with finite $\rho$-covering.

*Proof of Corollary 12.* We start the proof by showing that $V^*(s_1) - V_1^{\pi^t}(s_1)$ can be upper bounded by a sum of Bellman errors, which is a simple deduction from the policy loss decomposition lemma in Jiang et al. (2017) and is the same as the equality in Eq. (C.2) in the proof of Theorem 11 in §C. Next, we verify that $f^*$ satisfies constraint (E.7) so that taking $f^t = \arg\max V_{1,f}(s_1)$ in the confidence region yields $V_1^*(s_1) \leq V_{1,f^t}(s_1)$.

**Lemma 31** (Feasibility of $f^*$). In Algorithm 3, given $\rho > 0$ and $\delta > 0$, we choose $\beta = c(\log(TH|\mathcal{M}_\rho||\mathcal{V}_\rho|/\delta) + T\rho)$ for some large enough constant $c$, then with probability at least $1 - \delta$, $f^*$ satisfies for any $t \in [T]$:

$$\max_{v \in \mathcal{V}} \left\{ \sum_{i=1}^{t-1} \left( \mathbb{E}_{\widetilde{s} \sim f_h^*} v(s_h^i, a_h^i, \widetilde{s}) - v(s_h^i, a_h^i, s_{h+1}^i) \right)^2 \right.$$
$$\left. - \inf_{g_h \in \mathcal{G}_h} \sum_{i=1}^{t-1} \left( \mathbb{E}_{\widetilde{s} \sim g_h} v(s_h^i, a_h^i, \widetilde{s}) - v(s_h^i, a_h^i, s_{h+1}^i) \right)^2 \right\} \leq \beta.$$

We prove Lemma 31 in §F.5. The next Lemma 32 is devoted to controlling the average squared DEF.

**Lemma 32.** In Algorithm 3, given $\rho > 0$ and $\delta > 0$, we choose $\beta = c(\log(TH|\mathcal{M}_\rho||\mathcal{V}_\rho|/\delta) + T\rho)$ for some large enough constant $c$, then with probability at least $1 - \delta$, for all $(t, h) \in [T] \times [H]$, we have

$$\sum_{i=1}^{t-1} \max_{v \in \mathcal{V}} \mathbb{E}_{s_h \sim \pi_i, a_h \sim \pi_f} \left( \mathbb{E}_{\widetilde{s} \sim f_h} v(s_h^i, a_h^i, \widetilde{s}) - \mathbb{E}_{\widetilde{s} \sim f^*} v(s_h^i, a_h^i, \widetilde{s}) \right)^2 \leq \mathcal{O}(|\mathcal{A}|\beta).$$

Proof is delayed to §F.4. By Lemma 32 and properties of the witness rank in Definition 28, we have

$$\sum_{i=1}^{t-1} \langle W_h(f), X_h(f_i) \rangle^2 \leq \sum_{i=1}^{t-1} \left\{ \max_{v \in \mathcal{V}} \mathbb{E}_{s_h \sim \pi_i, a_h \sim \pi_f} \left( \mathbb{E}_{\widetilde{s} \sim f_h} v(s_h^i, a_h^i, \widetilde{s}) - \mathbb{E}_{\widetilde{s} \sim f^*} v(s_h^i, a_h^i, \widetilde{s}) \right) \right\}^2$$
$$\leq \sum_{i=1}^{t-1} \max_{v \in \mathcal{V}} \mathbb{E}_{s_h \sim \pi_i, a_h \sim \pi_f} \left( \mathbb{E}_{\widetilde{s} \sim f_h} v(s_h^i, a_h^i, \widetilde{s}) - \mathbb{E}_{\widetilde{s} \sim f^*} v(s_h^i, a_h^i, \widetilde{s}) \right)^2 \leq \mathcal{O}(|\mathcal{A}|\beta).$$

Applying Lemma 24 with $G_{h,f^*}(f, g) := \langle W_h(f), X_h(g) \rangle$ and $g_i = f^i, f_t = f^t$, we have

$$\sum_{i=1}^{t} |\langle W_h(f), X_h(f_i) \rangle| \leq \mathcal{O}\left( \sqrt{|\mathcal{A}| \dim_{\text{FE}}(\mathcal{F}, G_{h,f^*}, \omega) \beta t} + t\omega \right).$$

Policy loss decomposition (C.2) yields

$$\frac{1}{T} \sum_{t=1}^{T} V_1^*(s_1) - V_1^{\pi^t}(s_1) \leq \mathcal{O}\left( \frac{H}{\kappa} \sqrt{|\mathcal{A}| \dim_{\text{FE}}(\mathcal{F}, G_{h,f^*}, \omega) \left( \frac{\log(TH|\mathcal{M}_\rho||\mathcal{V}_\rho|/\delta)}{T} + \rho \right)} + \frac{H\omega}{\kappa} \right).$$

Taking $\omega = \frac{\kappa\epsilon}{H}$ and $\rho = \frac{\epsilon^2}{\dim_{\text{FE}}(\mathcal{F}, G, \frac{\epsilon}{H}) H^2}$, the above Eq. (C.4) becomes

$$\frac{1}{T} \sum_{t=1}^{T} V_1^*(s_1) - V_1^{\pi^t}(s_1) \leq \mathcal{O}\left( \frac{H}{\kappa} \sqrt{|\mathcal{A}| \frac{\dim_{\text{FE}}(\mathcal{F}, G, \frac{\epsilon}{H}) \log(TH|\mathcal{M}_\rho||\mathcal{V}_\rho|/\delta)}{T}} + \epsilon \right).$$

Taking

$$T = \frac{|\mathcal{A}| \dim_{\text{FE}}(\mathcal{F}, G, \frac{\epsilon}{H}) \log(TH|\mathcal{M}_\rho||\mathcal{V}_\rho|/\delta) H^2}{\kappa^2 \epsilon_{h+1}^2}$$

yields the desired result. □

### E.5 PROOF OF COROLLARY 13

We can directly apply Theorem 11 to the KNR model based on Proposition 10 to obtain the regret bound result. For better understanding of our framework, we illustrate the main features in the proof of Corollary 13 that are different from the proof of Theorem 11.

*Proof of Corollary 13.* To resolve the unboundedness issue, we unfold the analysis of KNR case and conclude a high-probability event $\mathcal{B}$ analogous to the argument in §E.3. However, doing so would impose an additional $\sqrt{d_s}$ factor induced by estimating the $\ell_2$-norm of multivariate Gaussians. In lieu to this, we present a sharper convergence analysis that incorporates KNR instance-specific structures. We recall the DEF of the KNR model:

$$\ell_h(o_h, f_{h+1}, g_h, v) = U_{h,g}\phi(s_h, a_h) - s_{h+1}.$$

We first define an auxilliary random variable

$$\begin{aligned}
X_t(h, f, v) &:= \left(\ell_h(o_h^t, f_{h+1}, g_h, v)\right)^2 - \left(\ell_h(o_h^t, f_{h+1}, \mathcal{T}(f)_h, v)\right)^2 \\
&= \left\|U_{h,f}\phi(s_h^t, a_h^t) - s_{h+1}^t\right\|_2^2 - \left\|U_h^*\phi(s_h^t, a_h^t) - s_{h+1}^t\right\|_2^2 \\
&= \left\langle (U_{h,f} - U_h^*)\phi(s_h^t, a_h^t), (U_{h,f} - U_h^*)\phi(s_h^t, a_h^t) - 2\epsilon_{h+1}^t \right\rangle \\
&= \|(U_{h,f} - U_h^*)\phi(s_h^t, a_h^t)\|^2 - 2\left\langle (U_{h,f} - U_h^*)\phi(s_h^t, a_h^t), \epsilon_{h+1}^t \right\rangle.
\end{aligned}$$

By the boundedness of operator $U_{h,f}$, $U_h^*$ and uniform boundedness of $\phi(s, a)$, we obtain that $\|(U_{h,f} - U_h^*)\phi(s_h^t, a_h^t)\|^2 \le 4B_U^2 B^2$. The conditional distribution of $\left\langle (U_{h,f} - U_h^*)\phi(s_h^t, a_h^t), \epsilon_{h+1}^t \right\rangle$ is a zero-mean Gaussian with variance $\sigma^2\|(U_{h,f} - U_h^*)\phi(s_h^t, a_h^t)\|^2 \le 4B_U^2 B^2\sigma^2$. By the tail bound of Gaussian distributions along with standard union bound, we know that with probability at least $1 - \delta$,

$$\left|\left\langle (U_{h,f} - U_h^*)\phi(s_h^t, a_h^t), \epsilon_{h+1}^t \right\rangle\right| \le \mathcal{O}\left(\sigma\sqrt{\log(TH/\delta)}\right)$$

holds uniformly for all $t \in [T]$ and $h \in [H]$. Thus, we bound the absolute value of the auxillary variable $X_t$ by $|X_t| \le R\sigma$ where $R$ is positive and of order $\mathcal{O}\left(\sqrt{\log(TH/\delta)}\right)$. Taking expectation with respect to $s_{h+1}$, we have

$$\mathbb{E}_{s_{h+1}}\left[X_t(h, f, v) \mid s_h, a_h\right] = \left\|(U_{h,f} - U_h^*)\phi(s_h^t, a_h^t)\right\|^2.$$

On the other hand,

$$\begin{aligned}
&\mathbb{E}_{s_{h+1}}\left[\left(X_t(h, f, v)\right)^2 \mid s_h, a_h\right] \\
&= \mathbb{E}_{s_{h+1}}\left[\left(\|(U_{h,f} - U_h^*)\phi(s_h^t, a_h^t)\|^2 - 2\left\langle (U_{h,f} - U_h^*)\phi(s_h^t, a_h^t), \epsilon_{h+1}^t \right\rangle\right)^2 \mid s_h, a_h\right] \\
&= \mathbb{E}_{s_{h+1}}\left[\|(U_{h,f} - U_h^*)\phi(s_h^t, a_h^t)\|^4 + 4\left\langle (U_{h,f} - U_h^*)\phi(s_h^t, a_h^t), \epsilon_{h+1}^t \right\rangle^2 \mid s_h, a_h\right] \\
&= \mathbb{E}_{s_{h+1}}\left[\|(U_{h,f} - U_h^*)\phi(s_h^t, a_h^t)\|^4 + 4\|(U_{h,f} - U_h^*)\phi(s_h^t, a_h^t)\|^2\sigma^2 \mid s_h, a_h\right] \\
&\le \mathcal{O}\left(\sigma^2 R^2 \mathbb{E}\left[X_t(h, f, v) \mid s_h, a_h\right]\right).
\end{aligned}$$

By taking $Z_t = X_t(h, f, v) - \mathbb{E}_{s_{h+1}}\left[X_t(h, f, v) \mid s_h, a_h\right]$ with $|Z_t| \le 2R\sigma$ in Freedman's inequality (F.1) in Lemma 33, we have for any $\eta$ satisfying $0 < \eta < \frac{1}{2R^2\sigma^2}$ almost surely, with probability at least $1 - \delta$:

$$\sum_{i=1}^t Z_i \le \mathcal{O}\left(R^2\sigma^2\eta\sum_{i=1}^t \mathbb{E}_{s_{h+1}}\left[X_i(h, f, v) \mid s_h, a_h\right] + \frac{\log(\delta^{-1})}{\eta}\right).$$

Optimizing over $\eta$, we have

$$\sum_{i=1}^t Z_i \le \mathcal{O}\left(R\sigma\sqrt{\sum_{i=1}^t \mathbb{E}_{s_{h+1}}\left[X_i(h, f, v) \mid s_h, a_h\right]\log(\delta^{-1})} + R^2\sigma^2\log(\delta^{-1})\right). \tag{E.19}$$

Following the same Freedman's inequality (Lemma 33) and $\rho$-covering argument as as in the proof of Theorem 11 with derivations detailed in §F.1, we have with probability $\geq 1 - \delta$ and $\beta = \mathcal{O}\left(\sigma^2 \log(TH\mathcal{N}_{\mathcal{L}}(\rho)/\delta) + \sigma\rho T\right)$:

$$\sum_{i=1}^{t} \left(\mathbb{E}_{s_h, a_h \sim \pi^i} ||(U_{h,f^t} - U_h^*)\phi(s_h, a_h)||\right)^2 \leq \sum_{i=1}^{t} \mathbb{E}_{s_h, a_h \sim \pi^i} ||(U_{h,f^t} - U_h^*)\phi(s_h, a_h)||^2 \leq \mathcal{O}(\beta).$$

Feasibility of $f^*$ can be derived by taking the same auxilliary random variable and analyze on $-\sum_{i=1}^{t} X_i(h, f, v)$ as in the proof of Lemma 31.

As explained in §E.3 , we can apply Lemma 24 with $\omega = \sqrt{\frac{1}{T}}$, $\rho = \frac{1}{T}$,

$$G_{h,f^*}(f, g) = \sqrt{\mathbb{E}_{s_h, a_h \sim \pi_g} ||(U_{h,f} - U_h^*)\phi(s_h, a_h)||^2},$$

and have

$$\sum_{i=1}^{t} \mathbb{E}_{s_h, a_h \sim \pi^i} \sqrt{||(U_{h,f^t} - U_h^*)\phi(s_h, a_h)||^2} \leq \sigma \sqrt{\dim_{\text{FE}}\left(\mathcal{F}, G, \sqrt{1/T}\right) \log\left(TH\mathcal{N}_{\mathcal{L}}(1/T)\right) \cdot t}.$$

The rest of the proof follows by applying Bellman dominance, policy loss decomposition and calculating the FE dimension based on $G_{h,f^*}(f, g)$, which is shown in Lemma 30. We therefore obtain that

$$\sum_{t=1}^{T} V_1^*(s_1) - V_1^{\pi^t}(s_1) \leq \frac{1}{\kappa} \sum_{t=1}^{T} \sum_{h=1}^{H} |G_{h,f^*}(f^t, f^t)|$$

$$\leq \mathcal{O}\left(\frac{H}{\kappa} \sigma \sqrt{T \cdot \dim_{\text{FE}}(\mathcal{F}, G, \sqrt{1/T}) \log\left(TH\mathcal{N}_{\mathcal{L}}(1/T)/\delta\right)}\right)$$

$$= \tilde{\mathcal{O}}\left(H^2 \sqrt{d_\phi^2 d_s T}\right).$$

$\square$

# F    Proof of Technical Lemmas

We start with introducing the Freedman's inequality that are crucial in proving concentration properties in our main results.

**Lemma 33** (Freedman-Style Inequality, Agarwal et al. 2014)**.** Consider an adapted sequence $\{Z_t, \mathcal{J}_t\}_{t=1,2,\ldots,T}$ that satisfies $\mathbb{E}\left[Z_t \mid \mathcal{J}_{t-1}\right] = 0$ and $Z_t \leq R$ for any $t = 1, 2, \ldots T$. Then for any $\delta > 0$ and $\eta \in [0, \frac{1}{R}]$, it holds with probability at least $1 - \delta$ that

$$\sum_{t=1}^{T} Z_t \leq (e-2)\eta \sum_{t=1}^{T} \mathbb{E}\left[Z_t^2 \mid \mathcal{J}_{t-1}\right] + \frac{\log(\delta^{-1})}{\eta}. \tag{F.1}$$

Before proving our technical lemmas, we note that for notational simplicity we use the expectation $\mathbb{E}_{s_{h+1}}[\cdot \mid s_h, a_h]$ to denote the conditional expectation with respect to the transition probability of the true model at $h$. The value of $s_h, a_h$ is data dependent (might be $s_h^i, a_h^i$ or $s_h^t, a_h^t$ depending on the function inside the expectation).

## F.1    Proof of Lemma 23

*Proof of Lemma 23.* We recall that $\ell$ has a bounded $\ell_2$-norm in Definition 6 and assume that $||\ell_{h,f'}(\cdot, f_{h+1}, g_h, v)|| \leq R$ for $\forall h \in [H], f', f \in \mathcal{F}, g \in \mathcal{G}, v \in \mathcal{V}$ throughout the paper. For a sequence of data $\mathcal{D}_h = \{r_h^t, s_h^t, a_h^t, s_{h+1}^t\}_{t=1,2,\ldots,T}$, we first build an auxiliary random variable defined for every $(t, h, f, v) \in [T] \times [H] \times \mathcal{F} \times \mathcal{V}$ and consider

$$X_i(h, f, v) := ||\ell_{h,f^i}(o_h^i, f_{h+1}, f_h, v)||^2 - ||\ell_{h,f^i}(o_h^i, f_{h+1}, \mathcal{T}(f)_h, v)||^2,$$

where the randomness is due to uniformly sampling the data sequence $\mathcal{D}_h$. We know that $|X_t(h, f)| \leq R^2$. Take conditional expectation of $X_i$ with respect to $s_h, a_h$, we have by definition that

$$\mathbb{E}_{s_{h+1}}\left[X_i(h, f, v) \mid s_h, a_h\right]$$
$$= \mathbb{E}_{s_{h+1}}\left[||\ell_{h,f^i}(o_h^i, f_{h+1}, f_h, v)||^2 - ||\ell_{h,f^i}(o_h^i, f_{h+1}, \mathcal{T}(f)_h, v)||^2 \mid s_h, a_h\right]$$

Using the fact that $\|a\|^2 - \|b\|^2 = \langle a - b, a + b \rangle$ for arbitrary vectors $a, b$ and property (i) in Definition 6 we have

$$\mathbb{E}_{s_{h+1}}\left[X_i(h, f, v) \mid s_h, a_h\right] = \big\langle \ell_{h,f^i}(o_h^i, f_{h+1}, f_h, v) - \ell_{h,f'}(o_h^i, f_{h+1}, \mathcal{T}(f)_h, v),$$
$$\mathbb{E}_{s_{h+1}}\left[\ell_{h,f^i}(o_h^i, f_{h+1}, f_h, v) + \ell_{h,f'}(o_h^i, f_{h+1}, \mathcal{T}(f)_h, v) \mid s_h, a_h\right]\big\rangle$$
$$= \|\mathbb{E}_{s_{h+1}}\left[\ell_{h,f^i}(o_h^i, f_{h+1}, f_h, v) \mid s_h, a_h\right]\|^2.$$

On the other hand,

$$\mathbb{E}_{s_{h+1}}\left[\left(X_i(h, f, v)\right)^2 \mid s_h, a_h\right] \leq \mathbb{E}_{s_{h+1}}\big[\|\ell_{h,f^i}(o_h^i, f_{h+1}, f_h, v) - \ell_{h,f^i}(o_h^i, f_{h+1}, \mathcal{T}(f)_h, v)\|^2$$
$$\cdot \|\ell_{h,f'}(o_h^i, f_{h+1}, f_h, v) + \ell_{h,f'}(o_h^i, f_{h+1}, \mathcal{T}(f)_h, v)\|^2 \mid s_h, a_h\big]$$
$$\leq 4\|\mathbb{E}_{s_{h+1}}\left[\ell_{h,f^i}(o_h^i, f_{h+1}, f_h, v) \mid s_h, a_h\right]\|^2 R^2$$
$$\leq 4R^2 \mathbb{E}_{s_{h+1}}\left[X_i(h, f, v) \mid s_h, a_h\right].$$

By taking $Z_t = X_t(h, f, v) - \mathbb{E}_{s_{h+1}}\left[X_t(h, f, v) \mid s_h, a_h\right]$ with $|Z_t| \leq 2R^2$ in Freedman's inequality (F.1) in Lemma 33, we have for any $\eta$ satisfying $0 < \eta < \frac{1}{2R^2}$, with probability at least $1 - \delta$:

$$\sum_{i=1}^{t} Z_i \leq \mathcal{O}\left(\eta \sum_{i=1}^{t} \mathrm{Var}\left[X_i(h, f, v) \mid s_h, a_h\right] + \frac{\log(\delta^{-1})}{\eta}\right)$$
$$\leq \mathcal{O}\left(\eta \sum_{i=1}^{t} \mathbb{E}_{s_{h+1}}\left[X_i^2(h, f, v) \mid s_h, a_h\right] + \frac{\log(\delta^{-1})}{\eta}\right)$$
$$\leq \mathcal{O}\left(4R^2 \eta \sum_{i=1}^{t} \mathbb{E}_{s_{h+1}}\left[X_i(h, f, v) \mid s_h, a_h\right] + \frac{\log(\delta^{-1})}{\eta}\right).$$

Taking $\eta = \frac{\sqrt{\log(\delta^{-1})}}{2R\sqrt{\sum_{i=1}^{t} \mathbb{E}[X_i(h,f,v)|s_h,a_h]}} \vee \frac{1}{2R^2}$, we have

$$\sum_{i=1}^{t} Z_i \leq \mathcal{O}\left(2R\sqrt{\sum_{i=1}^{t} \mathbb{E}_{s_{h+1}}\left[X_i(h, f, v) \mid s_h, a_h\right]\log(\delta^{-1})} + 2R^2 \log(\delta^{-1})\right). \tag{F.2}$$

Similarly by applying Freedman's inequality to $\sum_{i=1}^{t} -Z_t$ and combining with Eq. (F.2), we have that for any three-tuple $(t, h, f)$, the following holds with probability at least $1 - 2\delta$:

$$\left|\sum_{i=1}^{t} Z_i\right| \leq \mathcal{O}\left(2R\sqrt{\sum_{i=1}^{t} \mathbb{E}_{s_{h+1}}\left[X_i(h, f, v) \mid s_h, a_h\right]\log(\delta^{-1})} + 2R^2 \log(\delta^{-1})\right).$$

We note that in §3 we have that $\mathcal{L}$ admits a $\rho$-covering of $\mathcal{F}, \mathcal{G}, \mathcal{V}$, meaning that for any $\ell_{h,f'}(\cdot, f, g, v)$ and a $\rho > 0$ there exists a $\widetilde{\rho}$ and a four-tuple $(\widetilde{f'}, \widetilde{f}, \widetilde{g}, \widetilde{v}) \in \mathcal{F}_{\widetilde{\rho}} \times \mathcal{F}_{\widetilde{\rho}} \times \mathcal{G}_{\widetilde{\rho}} \times \mathcal{V}_{\widetilde{\rho}}$ such that $\left\|\ell_{h,\widetilde{f'}}(\cdot, \widetilde{f}, \widetilde{g}, \widetilde{v}) - \ell_{h,f'}(\cdot, f, g, v)\right\|_{\infty} \leq \rho$, where $\mathcal{F}_{\widetilde{\rho}}, \mathcal{G}_{\widetilde{\rho}}, \mathcal{V}_{\widetilde{\rho}}$ are $\widetilde{\rho}$-covers of $\mathcal{F}, \mathcal{G}, \mathcal{V}$ respectively. This is denoted by $(\widetilde{f'}, \widetilde{f}, \widetilde{g}, \widetilde{v}) \in \mathcal{L}_{\rho}$. In definition of $X_t$, $\widetilde{g}$ is always taken as $\widetilde{f}$ or a function of $\mathcal{T}(\widetilde{f})$. Then if $\mathcal{T}$ is Lipschitz, as it is mostly the expectation operator, we omit the $\widetilde{g}$ in the tuple and use $(\widetilde{f'}, \widetilde{f}, \widetilde{v}) \in \mathcal{L}_{\rho}$ to denote an element in the $\rho$-covering. By taking a union bound over $\mathcal{L}_{\rho}$, we have with probability at least $1 - 2\delta$ that the following holds for any $(\widetilde{f^i}, \widetilde{f}, \widetilde{v}) \in \mathcal{L}_{\rho}$,

$$\left|\sum_{i=1}^{t} \widetilde{X}_i(h, \widetilde{f}, \widetilde{v}) - \sum_{i=1}^{t} \mathbb{E}_{s_{h+1}}\left[\widetilde{X}_i(h, \widetilde{f}, \widetilde{v}) \mid s_h, a_h\right]\right|$$
$$\leq \mathcal{O}\left(2R\sqrt{\sum_{i=1}^{t} \mathbb{E}_{s_{h+1}}\left[\widetilde{X}_i(h, \widetilde{f}, \widetilde{v}) \mid s_h, a_h\right]\iota} + 2R^2 \iota\right), \tag{F.3}$$

where $\widetilde{X}_i(h, \widetilde{f}, \widetilde{v}) := ||\ell_{h,\widetilde{f}^i}(o_h^i, \widetilde{f}_{h+1}, \widetilde{f}_h, \widetilde{v})||^2 - ||\ell_{h,\widetilde{f}^i}(o_h^i, \widetilde{f}_{h+1}, \mathcal{T}(\widetilde{f})_h, \widetilde{v})||^2$ and $\iota = \log\left(\frac{HT\mathcal{N}_{\mathcal{L}}(\rho)}{\delta}\right)$. Further for any $X_i(h, f^t, v)$, we choose the three-tuple $(\widetilde{f}^i, \widetilde{f}^t, \widetilde{v}) := \arg\min_{(\widetilde{f}^i, \widetilde{f}^t, \widetilde{v}) \in \mathcal{L}_\rho} \left| X_i(h, f^t, v) - \widetilde{X}_i(h, \widetilde{f}^t, \widetilde{v}) \right| \le \rho$ and by the $\rho$-covering argument, we arrive at

$$\sum_{i=1}^{t-1} \widetilde{X}_i(h, \widetilde{f}^t, \widetilde{v}) = \sum_{i=1}^{t-1} \left[ ||\ell_{h,\widetilde{f}^i}(o_h^i, \widetilde{f}_{h+1}^t, \widetilde{f}_h^t, \widetilde{v})||^2 - ||\ell_{h,\widetilde{f}^i}(o_h^i, \widetilde{f}_{h+1}^t, \mathcal{T}(\widetilde{f})_h^t, \widetilde{v})||^2 \right]$$

$$\le \sum_{i=1}^{t-1} \left[ ||\ell_{h,f^i}(o_h^i, f_{h+1}^t, f_h^t, v)||^2 - ||\ell_{h,f^i}(o_h^i, f_{h+1}^t, \mathcal{T}(f^t)_h, v)||^2 \right] + \mathcal{O}(Rt\rho) \stackrel{(i)}{\le} \mathcal{O}(\beta + Rt\rho),$$
(F.4)

where $(i)$ comes from the constraint (4.1) of Algorithm 1.
Combining (F.3) with (F.4), we derive the following

$$\sum_{i=1}^{t-1} \mathbb{E}_{s_{h+1}} \left[ \widetilde{X}_i(h, \widetilde{f}^t, \widetilde{v}) \mid s_h, a_h \right] \le \mathcal{O}(\beta + Rt\rho + R^2\iota).$$

Applying the $\rho$-covering argument as in before, we conclude

$$\max_{v \in \mathcal{V}} \sum_{i=1}^{t-1} \mathbb{E}_{s_{h+1}} \left[ X_i(h, f^t, v) \mid s_h, a_h \right] \le \mathcal{O}(\beta + Rt\rho + R^2\iota).$$

Global optimality of the discriminator in (ii) of Definition 6 implies that $v_h^*$ is the optimal discriminator under any distribution or summation of $s_h, a_h$ (and thus max is interchangeable with summation):

$$\sum_{i=1}^{t-1} \mathbb{E}_{s_h, a_h \sim \pi^i} ||\mathbb{E}_{s_{h+1}} \left[ \ell_{h,f^i}(o_h, f_{h+1}^t, f_h^t, v_h^*(f^t)) \mid s_h, a_h \right]||^2$$

$$\ge \sum_{i=1}^{t-1} \mathbb{E}_{s_h, a_h \sim \pi^i} ||\mathbb{E}_{s_{h+1}} \left[ \ell_{h,f^i}(o_h, f_{h+1}^t, f_h^t, v) \mid s_h, a_h \right]||^2, \quad \forall v \in \mathcal{V}.$$

Thus, we have

$$\sum_{i=1}^{t-1} \max_{v \in \mathcal{V}} \mathbb{E}_{s_h, a_h \sim \pi^i} ||\mathbb{E}_{s_{h+1}} \left[ \ell_{h,f^i}(o_h, f_{h+1}^t, f_h^t, v) \mid s_h, a_h \right]||^2$$

$$= \sum_{i=1}^{t-1} \mathbb{E}_{s_h, a_h \sim \pi^i} ||\mathbb{E}_{s_{h+1}} \left[ \ell_{h,f^i}(o_h, f_{h+1}^t, f_h^t, v_h^*(f^t)) \mid s_h, a_h \right]||^2,$$

and also

$$\sum_{i=1}^{t-1} \mathbb{E}_{s_h, a_h \sim \pi^i} ||\mathbb{E}_{s_{h+1}} \left[ \ell_{h,f^i}(o_h, f_{h+1}^t, f_h^t, v_h^*(f^t)) \mid s_h, a_h \right]||^2$$

$$= \max_{v \in \mathcal{V}} \sum_{i=1}^{t-1} \mathbb{E}_{s_h, a_h \sim \pi^i} ||\mathbb{E}_{s_{h+1}} \left[ \ell_{h,f^i}(o_h, f_{h+1}^t, f_h^t, v) \mid s_h, a_h \right]||^2$$

$$= \max_{v \in \mathcal{V}} \sum_{i=1}^{t-1} \mathbb{E}_{s_h, a_h \sim \pi^i} \mathbb{E}_{s_{h+1}} \left[ X_i(h, f^t, v) \mid s_h, a_h \right] \le \mathcal{O}(\beta + Rt\rho + R^2\iota). \qquad \text{(F.5)}$$

We apply property (i) in Definition 5 and conclude that

$$\sum_{i=1}^{t-1} \left( G_{h,f^*}(f^t, f^i) \right)^2 \le \mathcal{O}(\beta),$$

which finishes the proof of Lemma 23. $\qquad\qquad\square$

## F.2 PROOF OF LEMMA 21

*Proof of Lemma 21.* For a data set $\mathcal{D}_h = \{r_h^t, s_h^t, a_h^t, s_{h+1}^t\}_{t=1,2,\ldots T}$, we first build an auxillary random variable defined for every $(t, h, f, v) \in [T] \times [H] \times \mathcal{F} \times \mathcal{V}$

$$X_i(h, f, v) := ||\ell_{h,f^i}(o_h^i, f_h^*, f_h, v)||^2 - ||\ell_{h,f^i}(o_h^i, f_h^*, f_h^*, v)||^2.$$

By similar derivations as in the proof of Lemma 23, we have

$$\mathbb{E}_{s_{h+1}}[X_i(h, f, v) \mid s_h, a_h] = \left(\mathbb{E}_{s_{h+1}}\left[\ell_{h,f^i}(o_h^i, f_h^*, f_h, v) \mid s_h, a_h\right]\right)^2,$$

$$\mathbb{E}_{s_{h+1}}\left[(X_i(h, f, v))^2 \mid s_h, a_h\right] \le 4R^2 \mathbb{E}_{s_{h+1}}[X_i(h, f, v) \mid s_h, a_h].$$

Take $Z_t = X_t(h, f, v) - \mathbb{E}_{s_{h+1}}[X_t(h, f, v) \mid s_h, a_h]$ with $|Z_t| \le 2R^2$ in Freedman's inequality (F.1) in Lemma 33. Then via the same procedure as in the proof of Lemma 23 we have that for any four-tuple $(t, h, f, v)$, the following holds with probability at least $1 - 2\delta$:

$$\left|\sum_{i=1}^t X_i(h, f, v) - \sum_{i=1}^t \mathbb{E}_{s_{h+1}}[X_i(h, f, v) \mid s_h, a_h]\right|$$

$$\le \mathcal{O}\left(2R\sqrt{\sum_{i=1}^t \mathbb{E}_{s_{h+1}}[X_i(h, f, v) \mid s_h, a_h]\log(\delta^{-1})} + 2R^2 \log(\delta^{-1})\right).$$

Thus, we have

$$-\sum_{i=1}^t X_i(h, f, v) \le \mathcal{O}(R^2 \log(\delta^{-1})).$$

By the same $\rho$-covering argument as in the proof of Lemma 23, there exists a $\rho$-covering of $\mathcal{L}$ such that we can take a union bound over $\mathcal{L}_\rho$ and have $-\sum_{i=1}^{t-1} \widetilde{X}_i(h, \widetilde{f}, \widetilde{v}) \le \mathcal{O}\left(R^2\iota + Rt\rho\right)$ where $\iota = \log\left(\frac{HT\mathcal{N}_\mathcal{L}(\rho)}{\delta}\right)$. Then for $f^*$, any $f \in \mathcal{F}$ and any $v \in \mathcal{V}$, we can use the nearest three-tuple $(\widetilde{f^i}, \widetilde{f}, \widetilde{v})$ in the $\rho$-covering and conclude that

$$\max_{v \in \mathcal{V}} \sum_{i=1}^{t-1} \left[||\ell_{h,f^i}(o_h^i, f_h^*, f_h^*, v)||^2 - ||\ell_{h,f^i}(o_h^i, f_h^*, f_h, v)||^2\right] = \max_{v \in \mathcal{V}} \sum_{i=1}^{t-1} -X_i(h, f, v) \le \mathcal{O}(\beta).$$

This in sum finishes our proof of Lemma 21 with $\beta = \mathcal{O}\left(R^2\iota + R\rho t\right)$. $\qquad\square$

## F.3 PROOF OF LEMMA 24

*Proof of Lemma 24.* The proof basically follows Appendix §C of Russo & Van Roy (2013) and Appendix §D of Jin et al. (2021). We first prove that for all $t \in [T]$,

$$\sum_{k=1}^t \mathbb{1}(|G(f_k, g_k)| > \epsilon) \le (\beta/\epsilon^2 + 1)\dim_{FE}(\mathcal{F}, G, \epsilon). \tag{F.6}$$

Let $m := \sum_{k=1}^t \mathbb{1}(|G(f_k, g_k)| > \epsilon)$, then there exists $\{s_1, \ldots, s_m\}$ which is a subsequence of $[t]$ such that $G(f_{s_1}, g_{s_1}), \ldots, G(f_{s_m}, g_{s_m}) > \epsilon$.

We first show that for the sequence $\{f_{s_1}, \ldots, f_{s_m}\} \subseteq \mathcal{F}$, there exists $j \in [m]$ such that $f_{s_j}$ is $\epsilon$-independent on at least $L = \lceil(m-1)/\dim_{FE}(\mathcal{F}, G, \epsilon)\rceil$ disjoint sequences in $\{f_{s_1}, \ldots, f_{s_{j-1}}\}$ (Russo & Van Roy, 2013). We will prove this by following procedure. Starting with singleton sequences $B_1 = \{f_{s_1}\}, \ldots, B_L = \{f_{s_L}\}$ and $j = L + 1$. For each $j$, if $f_{s_j}$ is $\epsilon$-dependent on $B_1, \ldots, B_L$ we already achieved our goal and the process stops. Otherwise, there exist $i \in [L]$ such that $f_{s_j}$ is $\epsilon$-dependent of $B_i$ and update $B_i = B_i \cup \{f_{s_j}\}$. Then we add increment $j$ by 1 and continue the process. By the definition of FE dimension, the cardinally of each set $B_1, \ldots, B_L$ cannot larger than $\dim_{FE}(\mathcal{F}, G, \epsilon)$ at any point in this process. Therefore, by pigeonhole principle the process stops by step $j = L\dim_{FE}(\mathcal{F}, G, \epsilon) + 1 \le m$.

Therefore, we have proved that there exists $j$ such that $|G(f_{s_j}, g_{s_j})| > \epsilon$ and $f_{s_j}$ is $\epsilon$-independent with at least $L = \lceil (m-1)/\dim_{FE}(\mathcal{F}, G, \epsilon) \rceil$ disjoint sequences in $\{f_{s_1}, \ldots, f_{s_{j-1}}\}$. For each of the sequences $\{\widehat{f}_1, \ldots, \widehat{f}_l\}$, by definition of the FE dimension in Definition 3 we have that

$$\sum_{k=1}^{l} \left(G(\widehat{f}_k, g_{s_j})\right)^2 \geq \epsilon^2. \tag{F.7}$$

Summing all of bounds (F.7) for $L$ disjoint sequences together we have that

$$\sum_{k=1}^{s_j-1} \left(G(f_t, g_{s_j})\right)^2 \geq L\epsilon^2 = \lceil (m-1)/\dim_{FE}(\mathcal{F}, G, \epsilon) \rceil \cdot \epsilon^2. \tag{F.8}$$

The left hand side of (F.8) can be upper bounded by $\beta^2$ due to the condition of lemma. Therefore, we have proved that $\beta^2 \geq \lceil (m-1)/\dim_{FE}(\mathcal{F}, G, \epsilon) \rceil \cdot \epsilon^2$ which completes the proof of (F.6).

Now let $d = \dim_{FE}(\mathcal{F}, G, \omega)$ and sort $|G(f_1, g_1)|, \ldots, |G(f_t, g_t)|$ in a nonincreasing order, denoted by $e_1, \ldots, e_t$. Then we have that

$$\sum_{k=1}^{t} |G(f_k, g_k)| = \sum_{k=1}^{t} e_k = \sum_{k=1}^{t} e_k \, \mathbb{1}(e_k \leq \omega) + \sum_{i=1}^{t} e_k \, \mathbb{1}(e_k > \omega) \leq t\omega + \sum_{i=1}^{t} e_k \, \mathbb{1}(e_k > \omega). \tag{F.9}$$

For $k \in [t]$, we want to give an upper bound for those $e_k \, \mathbb{1}(e_k > \omega)$. Assume $e_k > \omega$, then for any $\alpha$ such that $e_k > \alpha \geq \omega$, by (F.6), we have that

$$k \leq \sum_{i=1}^{t} \mathbb{1}(e_i > \omega) \leq (\beta/\alpha^2 + 1) \dim_{FE}(\mathcal{F}, G, \alpha) \leq (\beta/\alpha^2 + 1)d,$$

which implies that $\alpha \leq \sqrt{d\beta/(k-d)}$. Taking the limit $\alpha \to e_k^-$, we have that $e_k \leq \min\{\sqrt{d\beta/(k-d)}, C\}$. Finally, we have that

$$\sum_{k=1}^{t} e_i \, \mathbb{1}(e_k > \omega) \leq \min\{d, t\} \cdot C + \sum_{i=d+1}^{t} \sqrt{\frac{d\beta}{k-d}}$$

$$\leq \min\{d, t\} \cdot C + \sqrt{d\beta} \int_0^t z^{-1/2} dz \leq \min\{d, t\} \cdot C + 2\sqrt{d\beta t}. \tag{F.10}$$

Plugging (F.10) into (F.9) completes the proof. $\qquad\square$

### F.4 PROOF OF LEMMA 32

*Proof of Lemma 32.* We assume that $\|v\|_\infty \leq B$ and treat $B$ as an absolute constant ($B = 2$ in Sun et al. (2019)) in the following derivations. For a dataset $\mathcal{D}_h = \{r_h^t, s_h^t, a_h^t, s_{h+1}^t\}_{t=1,2,\ldots T}$, we first build an auxillary random variable defined for every $(t, h, f, v) \in [T] \times [H] \times \mathcal{F} \times \mathcal{V}$

$$X_t(h, f, v) := \left[ \left(\mathbb{E}_{\widetilde{s}\sim f} v(s_h^t, a_h^t, \widetilde{s}) - v(s_h^t, a_h^t, s_{h+1}^t)\right)^2 - \left(\mathbb{E}_{\widetilde{s}\sim f^*} v(s_h^t, a_h^t, \widetilde{s}) - v(s_h^t, a_h^t, s_{h+1}^t)\right)^2 \right],$$

where the randomness lies in the sampling of the dataset $\mathcal{D}_h$. We know that $|X_t(h, f)| \leq 4B^2$ almost surely. Take conditional expectation of $X_i$ with respect to $s_h, a_h$, we have by definition that

$$\mathbb{E}_{s_{h+1}}\left[X_i(h, f, v) \mid s_h, a_h\right] = \mathbb{E}_{s_{h+1}}\left[\left(\mathbb{E}_{\widetilde{s}\sim f} v(s_h^i, a_h^i, \widetilde{s}) - v(s_h^i, a_h^i, s_{h+1}^i)\right)^2\right.$$
$$\left. - \left(\mathbb{E}_{\widetilde{s}\sim f^*} v(s_h^i, a_h^i, \widetilde{s}) - v(s_h^i, a_h^i, s_{h+1}^i)\right)^2 \mid s_h, a_h\right].$$

Using the fact that $a^2 - b^2 = (a-b)(a+b)$ and $\mathbb{E}_{\widetilde{s}\sim f} v(s_h^i, a_h^i, \widetilde{s}) - \mathbb{E}_{\widetilde{s}\sim f^*} v(s_h^i, a_h^i, \widetilde{s})$ is nonrandom given $s_h, a_h$, we have

$$\mathbb{E}_{s_{h+1}}\left[X_i(h, f, v) \mid s_h, a_h\right]$$
$$= \left(\mathbb{E}_{\widetilde{s}\sim f} v(s_h^i, a_h^i, \widetilde{s}) - \mathbb{E}_{\widetilde{s}\sim f^*} v(s_h^i, a_h^i, \widetilde{s})\right)$$
$$\cdot \mathbb{E}_{s_{h+1}}\left[\mathbb{E}_{\widetilde{s}\sim f} v(s_h^i, a_h^i, \widetilde{s}) + \mathbb{E}_{\widetilde{s}\sim f^*} v(s_h^i, a_h^i, \widetilde{s}) - 2v(s_h^i, a_h^i, s_{h+1}^i) \mid s_h, a_h\right]$$
$$= \left(\mathbb{E}_{\widetilde{s}\sim f} v(s_h^i, a_h^i, \widetilde{s}) - \mathbb{E}_{\widetilde{s}\sim f^*} v(s_h^i, a_h^i, \widetilde{s})\right)^2.$$

On the other hand,

$$\mathbb{E}_{s_{h+1}}\left[X_i(h,f,v)^2 \mid s_h, a_h\right] \leq \mathbb{E}_{s_{h+1}}\left[\left[\left(\mathbb{E}_{\widetilde{s}\sim f}v(s_h^i, a_h^i, \widetilde{s}) - \mathbb{E}_{\widetilde{s}\sim f^*}v(s_h^i, a_h^i, \widetilde{s})\right)4B\right]^2 \mid s_h, a_h\right]$$

$$= 16B^2\left(\mathbb{E}_{\widetilde{s}\sim f}v(s_h^i, a_h^i, \widetilde{s}) - \mathbb{E}_{\widetilde{s}\sim f^*}v(s_h^i, a_h^i, \widetilde{s})\right)^2$$

$$\leq 16B^2 \mathbb{E}_{s_{h+1}}\left[X_i(h,f,v) \mid s_h, a_h\right].$$

By taking $Z_t = X_t(h,f,v) - \mathbb{E}_{s_{h+1}}\left[X_t(h,f,v) \mid s_h, a_h\right]$ with $|Z_t| \leq 8B^2$ a.s. in Freedman's inequality (F.1) in Lemma 33, by the same procedure as in the proof of Lemma 23, we have that for any four-tuple $(t,h,f,v)$, the following holds with probability at least $1 - 2\delta$:

$$\left|\sum_{i=1}^t X_i(h,f,v) - \sum_{i=1}^t \mathbb{E}_{s_{h+1}}\left[X_i(h,f,v) \mid s_h, a_h\right]\right|$$

$$\leq \mathcal{O}\left(4B\sqrt{\sum_{i=1}^t \mathbb{E}_{s_{h+1}}\left[X_i(h,f,v) \mid s_h, a_h\right]\log(\delta^{-1}) + 8B^2\log(\delta^{-1})}\right). \tag{F.11}$$

Let $\mathcal{M}_\rho$ be a $\rho$-cover of $\mathcal{M}$ and $\mathcal{V}_\rho$ a $\rho$-cover of $\mathcal{V}$. By taking a union bound over all $(t,h,f',v') \in [T]\times[H]\times\mathcal{M}_\rho\times\mathcal{V}_\rho$, we have with probability at least $1 - 2\delta$ that the following holds for any $f'\in\mathcal{M}_\rho, v'\in\mathcal{V}_\rho$,

$$\left|\sum_{i=1}^t X_i(h,f',v') - \sum_{i=1}^t \mathbb{E}_{s_{h+1}}\left[X_i(h,f',v') \mid s_h, a_h\right]\right|$$

$$\leq \mathcal{O}\left(4B\sqrt{\sum_{i=1}^t \mathbb{E}_{s_{h+1}}\left[X_i(h,f',v') \mid s_h, a_h\right]\iota + 8B^2\iota}\right), \tag{F.12}$$

where $\iota = \log(\frac{HT|\mathcal{M}_\rho||\mathcal{V}_\rho|}{\delta})$. Further for any $f^t$ calculated at $t\in[T]$ and any $v\in\mathcal{V}$, we choose $f' = \arg\min_{\widetilde{f}\in\mathcal{M}_\rho}\text{dist}(\widetilde{f}, f^t)$ where dist is the distance measure on $\mathcal{M}$, $v' = \min_{v'\in\mathcal{V}_\rho}(v',v)$ and conclude

$$\sum_{i=1}^{t-1}X_i(h,f',v')$$

$$= \sum_{i=1}^{t-1}\left[\left(\mathbb{E}_{\widetilde{s}\sim f'}v'(s_h^i, a_h^i, \widetilde{s}) - v'(s_h^i, a_h^i, s_{h+1}^i)\right)^2 - \left(\mathbb{E}_{\widetilde{s}\sim f^*}v'(s_h^i, a_h^i, \widetilde{s}) - v'(s_h^i, a_h^i, s_{h+1}^i)\right)^2\right]$$

$$\leq \sum_{i=1}^{t-1}\left[\left(\mathbb{E}_{\widetilde{s}\sim f^t}v'(s_h^i, a_h^i, \widetilde{s}) - v'(s_h^i, a_h^i, s_{h+1}^i)\right)^2 - \left(\mathbb{E}_{\widetilde{s}\sim f^*}v'(s_h^i, a_h^i, \widetilde{s}) - v'(s_h^i, a_h^i, s_{h+1}^i)\right)^2\right] + \mathcal{O}(Bt\rho)$$

$$\overset{(i)}{\leq} \mathcal{O}(\beta + Bt\rho), \tag{F.13}$$

where $(i)$ is due to the constraint of Algorithm 3. Combining (F.12) with (F.13), we derive the following

$$\sum_{i=1}^{t-1}\mathbb{E}_{s_{h+1}}\left[X_i(h,f',v') \mid s_h, a_h\right] \leq \mathcal{O}(\beta + Bt\rho + B^2\iota).$$

Note that $f'$ is chosen as the nearest model to $f^t$ in the $\rho$-covering of $\mathcal{M}$ and for any $v$ there exists a nearest $v'$ in the $\rho$-covering of $\mathcal{V}$, we conclude

$$\max_{v\in\mathcal{V}}\sum_{i=1}^{t-1}\mathbb{E}_{s_{h+1}}\left[X_i(h,f^t,v) \mid s_h, a_h\right] \leq \mathcal{O}(\beta + Bt\rho + B^2\iota).$$

Note we also have proved property (ii) in Definition 6 in §E.2, and we apply the global optimality of the discriminator as in the proof of Lemma 23 and obtains

$$\sum_{i=1}^{t-1}\max_{v\in\mathcal{V}}\mathbb{E}_{s_{h+1}}\left[X_i(h,f^t,v) \mid s_h, a_h\right] \leq \mathcal{O}(\beta + Bt\rho + B^2\iota).$$

Multiplying $\left[\mathbb{E}_{\widetilde{s}\sim f}v(s_h^i, a_h^i, \widetilde{s}) - \mathbb{E}_{\widetilde{s}\sim f^*}v(s_h^i, a_h^i, \widetilde{s})\right]^2$ by $\frac{\mathbb{1}(a_h^i = \pi_f(s_h^i))}{1/|\mathcal{A}|}$, taking expectation on $s_h^i \sim \pi^i, a_h^i \sim \pi_f$ and again using the global discriminator optimality, we arrive at

$$\sum_{i=1}^{t-1} \max_{v \in \mathcal{V}} \mathbb{E}_{s_h^i \sim \pi_i, a_h^i \sim \pi_f} \left[\left(\mathbb{E}_{\widetilde{s}\sim f_h}v(s_h^i, a_h^i, \widetilde{s}) - \mathbb{E}_{\widetilde{s}\sim f^*}v(s_h^i, a_h^i, \widetilde{s})\right)^2\right]$$

$$= \sum_{i=1}^{t-1} \max_{v \in \mathcal{V}} \mathbb{E}_{s_h^i \sim \pi^i, a_h^i \sim U(A)} \frac{\mathbb{1}(a_h^i = \pi_f(s_h^i))}{1/|\mathcal{A}|} \left[\left(\mathbb{E}_{\widetilde{s}\sim f_h}v(s_h^i, a_h^i, \widetilde{s}) - \mathbb{E}_{\widetilde{s}\sim f^*}v(s_h^i, a_h^i, \widetilde{s})\right)^2\right]$$

$$\leq \mathcal{O}(|\mathcal{A}|\left(\beta + Bt\rho + B^2\iota\right)),$$

which concludes the proof. $\qquad\square$

## F.5 Proof of Lemma 31

*Proof of Lemma 31.* For a dataset $\mathcal{D}_h = \{r_h^t, s_h^t, a_h^t, s_{h+1}^t\}_{t=1,2,...T}$, we first build an auxillary random variable defined for every $(t, h, f, v) \in [T] \times [H] \times \mathcal{F} \times \mathcal{V}$

$$X_t(h, f, v) := \left[\left(\mathbb{E}_{\widetilde{s}\sim f}v(s_h^t, a_h^t, \widetilde{s}) - v(s_h^t, a_h^t, s_{h+1}^t)\right)^2 - \left(\mathbb{E}_{\widetilde{s}\sim f^*}v(s_h^t, a_h^t, \widetilde{s}) - v(s_h^t, a_h^t, s_{h+1}^t)\right)^2\right].$$

By Eq. (F.11), with probability at least $1 - 2\delta$,

$$\left|\sum_{i=1}^t X_i(h, f, v) - \sum_{i=1}^t \mathbb{E}_{s_{h+1}}\left[X_i(h, f, v) \mid s_h, a_h\right]\right|$$

$$\leq \mathcal{O}\left(4B\sqrt{\sum_{i=1}^t \mathbb{E}_{s_{h+1}}\left[X_i(h, f, v) \mid s_h, a_h\right]\log(\delta^{-1}) + 8B^2\log(\delta^{-1})}\right).$$

Let $\mathcal{M}_\rho$ be a $\rho$-cover of $\mathcal{M}$ and $\mathcal{V}_\rho$ a $\rho$-cover of $\mathcal{V}$. By taking a union bound over all $(t, h, f', v`) \in [T] \times [H] \times \mathcal{M}_\rho \times \mathcal{V}_\rho$, we have with probability at least $1 - 2\delta$ that the following holds for any $f' \in \mathcal{Z}_\rho$,

$$\left|\sum_{i=1}^t X_i(h, f', v') - \sum_{i=1}^t \mathbb{E}_{s_{h+1}}\left[X_i(h, f', v') \mid s_h, a_h\right]\right|$$

$$\leq \mathcal{O}\left(4B\sqrt{\sum_{i=1}^t \mathbb{E}_{s_{h+1}}\left[X_i(h, f', v') \mid s_h, a_h\right]\iota + 8B^2\iota}\right),$$

where $\iota = \log\left(\frac{HT|\mathcal{M}_\rho||\mathcal{V}_\rho|}{\delta}\right)$. Thus, we have

$$-\sum_{i=1}^t X_i(h, f', v') \leq \mathcal{O}\left(B^2\iota\right).$$

Further for any $f \in \mathcal{F}$ and any $v \in \mathcal{V}$, we choose $f' = \arg\min_{\widetilde{f} \in \mathcal{M}_\rho} \text{dist}(\widetilde{f}, f)$ where dist is the distance measure on $\mathcal{M}$, $v' = \min_{v' \in \mathcal{V}_\rho}(v', v)$ and have

$$-\sum_{i=1}^{t-1} X_i(h, f, v) = \sum_{i=1}^{t-1}\left(\mathbb{E}_{\widetilde{s}\sim f^*}v(s_h^t, a_h^t, \widetilde{s}) - v(s_h^t, a_h^t, s_{h+1}^t)\right)^2 - \sum_{i=1}^{t-1}\left(\mathbb{E}_{\widetilde{s}\sim f}v(s_h^t, a_h^t, \widetilde{s}) - v(s_h^t, a_h^t, s_{h+1}^t)\right)^2$$

$$\leq \mathcal{O}\left(B^2\iota + B\rho t\right).$$

Thus,

$$\max_{v \in \mathcal{V}}\left[\sum_{i=1}^{t-1}\left(\mathbb{E}_{\widetilde{s}\sim f^*}v(s_h^i, a_h^i, \widetilde{s}) - v(s_h^i, a_h^i, s_{h+1}^i)\right)^2 - \inf_{g \in \mathcal{Q}}\sum_{i=1}^{t-1}\left(\mathbb{E}_{\widetilde{s}\sim g}v(s_h^i, a_h^i, \widetilde{s}) - v(s_h^i, a_h^i, s_{h+1}^i)\right)^2\right] \leq \beta,$$

which concludes the proof. $\qquad\square$

# G  PROOF FOR FUNCTIONAL ELUDER DIMENSION

In the following proposition, we prove that the Bellman eluder (BE) dimension (Jin et al., 2021) is a special case of the FE dimension when $G_h(g, f) := \mathbb{E}_{\pi_{h,f}}(g_h - \mathcal{T}_h g_{h+1})$.

**Proposition 34.** For any hypothesis class $\mathcal{F}$, taking coupling function $G$ to be the union of $\{G_h : \mathcal{F}_h \times \mathcal{F}_h \to \mathbb{R}\}_{h=1,\dots,H}$ with each $G_h(g, f) := \mathbb{E}_{\pi_{h,f}}(g_h - \mathcal{T}_h g_{h+1})$.

$$\dim_{\mathrm{FE}}(\mathcal{F}, G, \epsilon) \leq \dim_{\mathrm{BE}}(\mathcal{F}, \Pi, \epsilon).$$

*Proof of Proposition 34.* By definition of the functional eluder dimension,

$$\dim_{\mathrm{FE}}(\mathcal{F}, G, \epsilon) = \max_{h \in [H]} \dim_{\mathrm{FE}}(\mathcal{F}, G_h, \epsilon),$$

where $\dim_{\mathrm{FE}}(\mathcal{F}, G_h, \epsilon)$ is the length $n$ of the longest sequence satisfying for every $t \in [n]$, $\sqrt{\sum_{i=1}^{t-1} (G_h(g_t, f_i))^2} \leq \epsilon'$ and $|G_h(g_t, f_t)| > \epsilon'$. Bringing in $G_h(g, f) := \mathbb{E}_{\pi_{h,f}}(g_h - \mathcal{T}_h g_{h+1})$, we have $f_1, \dots, f_n$ is also the longest sequence that satisfies for some $g_1, \dots, g_n$ that

$$\sqrt{\sum_{i=1}^{t-1} \left(\mathbb{E}_{\pi_{h,f_i}}(g_{t,h} - \mathcal{T}_h g_{t,h+1})\right)^2} \leq \epsilon', \qquad \text{and} \qquad \left|\mathbb{E}_{\pi_{h,f_t}}(g_{t,h} - \mathcal{T}_h g_{t,h+1})\right| > \epsilon'.$$

Thus, $\dim_{\mathrm{DE}}((I - \mathcal{T}_h)\mathcal{F}, \Pi_h, \epsilon) \geq n$. Taking maximum over $h \in [H]$, we have

$$\dim_{\mathrm{FE}}(\mathcal{F}, G, \epsilon) = \max_{h \in [H]} \dim_{\mathrm{FE}}(\mathcal{F}, G_h, \epsilon) \leq \max_{h \in [H]} \dim_{\mathrm{DE}}((I - \mathcal{T}_h)\mathcal{F}, \Pi_h, \epsilon) = \dim_{\mathrm{BE}}(\mathcal{F}, \Pi, \epsilon),$$

which concludes our proof. $\qquad \qquad \square$

Combining Proposition 34 with Proposition 29 in Jin et al. (2021), it is straightforward to conclude that FE dimension is smaller than the effective dimension. In particular, Proposition 33 says $\dim_{\mathrm{FE}}$ is controlled by $\dim_{\mathrm{BE}}$, Proposition 29 in Jin et al. (2021) says $\dim_{\mathrm{BE}}$ is controlled by the effective dimension $\dim_{\mathrm{eff}}$, therefore low effective dimension would imply ABC with low FE dimension.

In the following paragraphs and Proposition 35 we prove this conclusion from sketch to grant a better understanding of the FE dimension.

The effective dimension (Jin et al., 2021) (or equivalently, critical information gain (Du et al., 2021)) $d_{\mathrm{eff}}(\mathcal{X}, \epsilon)$ of a set $\mathcal{X}$ is defined as the smallest interger $n > 0$ such that

$$n > e \cdot \sup_{x_1, \dots, x_n \in \mathcal{X}} \log \det \left( I + \frac{1}{\epsilon^2} \sum_{i=1}^{n} x_i x_i^\top \right).$$

Remark 5.2 in Du et al. (2021) showed that for finite dimensional setting with $\mathcal{X} \subseteq \mathbb{R}^d$ and $||x|| \leq B$, $d_{\mathrm{eff}}(\mathcal{X}, \epsilon) = \widetilde{\mathcal{O}}(d)$. Moreover, the effective dimension can be small even for infinite dimensional RKHS case.

In the next proposition, we prove that when the coupling function exhibits a bilinear structure $G(f, g) = \langle W(f), X(g) \rangle_{\mathcal{H}}$ with feature space $\mathcal{X} := \{X(g) \in \mathcal{H} : g \in \mathcal{F}\}$ and $||X(g)||_{\mathcal{H}} \leq \sqrt{B}$, the functional eluder dimension in Definition 4 is always less than the effective dimesion of $\mathcal{X}$.

**Proposition 35.** For any hypothesis class $\mathcal{F}$ and coupling function $G(\cdot, \cdot) : \mathcal{F} \times \mathcal{F} \to \mathbb{R}$ that can be expressed in bilinear form $\langle W(f), X(g) \rangle_{\mathcal{H}}$, we have

$$\dim_{\mathrm{FE}}(\mathcal{F}, G, \epsilon) \leq d_{\mathrm{eff}}\left(\mathcal{X}, \epsilon/\sqrt{B}\right).$$

*Proof of Proposition 35.* The proof basically follows the proof of Proposition 29 in Jin et al. (2021) with modifications specified for the functional eluder dimension. Given a hypothesis class $\mathcal{F}$ and a coupling function $G(\cdot, \cdot) : \mathcal{F} \times \mathcal{F} \to \mathbb{R}$. Suppose there exists an $\epsilon$'-independent sequence $f_1, \dots, f_n \in \mathcal{F}$ such that there exist $g_1, \dots, g_n \in \mathcal{F}$,

$$\begin{cases} \sqrt{\sum_{i=1}^{t-1} (G(g_t, f_i))^2} \leq \epsilon', & t \in [n], \\[2mm] |G(g_t, f_t)| > \epsilon', & t \in [n]. \end{cases} \tag{G.1}$$

When $G(f, g) := \langle W(f), X(g) \rangle_{\mathcal{H}}$, the above becomes

$$\begin{cases} \sqrt{\sum_{i=1}^{t-1} \langle W(g_t), X(f_i) \rangle_{\mathcal{H}}^2} \leq \epsilon', & t \in [n], \\ |\langle W(g_t), X(f_t) \rangle_{\mathcal{H}}| > \epsilon', & t \in [n]. \end{cases} \tag{G.2}$$

Defining $\Sigma_t = \sum_{i=1}^{t-1} X(f_i) X(f_i)^\top + \frac{\epsilon'^2}{B} \cdot I$, we have by Eq. (G.2) that $\|W(g_t)\|_{\Sigma_t} \leq \sqrt{2}\epsilon'$. Furthermore,

$$\epsilon' \leq |\langle W(g_t), X(f_t) \rangle_{\mathcal{H}}| \leq \|W(g_t)\|_{\Sigma_t} \cdot \|X(f_t)\|_{\Sigma_t^{-1}} \leq \sqrt{2}\epsilon' \|X(f_t)\|_{\Sigma_t^{-1}}.$$

Thus, we have $\|X(f_t)\|_{\Sigma_t^{-1}}^2 \geq \frac{1}{2}$ for any $t \in [n]$. By applying the log-determinant argument, we have

$$\sum_{t=1}^n \log \left(1 + \|x_t\|_{\Sigma_t^{-1}}^2\right) = \log \left(\frac{\det(\Sigma_{n+1})}{\det(\Sigma_t)}\right) = \log \det \left(1 + \frac{B}{\epsilon'^2} \sum_{i=1}^n x_i x_i^\top\right).$$

The above equality implies

$$\frac{1}{2} \leq \min_{t \in [n]} \|x_t\|_{\Sigma_t^{-1}}^2 \leq \exp \left(\frac{1}{n} \log \det \left(1 + \frac{B}{\epsilon'^2} \sum_{i=1}^n x_i x_i^\top\right)\right) - 1. \tag{G.3}$$

Taking $n = d_{\text{eff}}(\mathcal{X}, \epsilon/\sqrt{B})$ yields

$$\exp \left(\frac{1}{n} \log \det \left(1 + \frac{B}{\epsilon'^2} \sum_{i=1}^n x_i x_i^\top\right)\right) \leq \frac{1}{n} \sup_{x_1, \ldots, x_n \in \mathcal{X}} \log \det \left(I + \frac{B}{\epsilon^2} \sum_{i=1}^n x_i x_i^\top\right) \leq e^{-1},$$

which contradicts with the inequality (G.3) and concludes our proof. $\qquad\square$

We now provide the detailed proofs of Lemmas 27, 29 and 30.

### G.1 PROOF OF LEMMA 27

*Proof of Lemma 27.* Taking

$$G_{h, f^*}(f, g) := (\theta_{h, g} - \theta_h^*)^\top \mathbb{E}_{s_h, a_h \sim \pi_g} \left[\psi(s_h, a_h) + \sum_{s'} \phi(s_h, a_h, s') V_{h+1, g}(s')\right] = \langle W_h(f), X_h(g) \rangle,$$

where $W_h(f) := \theta_{h, f} - \theta_h^*, X_h(g) := \mathbb{E}_{s_h, a_h \sim \pi_g}[\psi(s_h, a_h) + \sum_{s'} \phi(s_h, a_h, s') V_{h+1, g}(s')]$ in Proposition 35. Properties of the effective dimension yield that the FE dimension of the linear mixture MDP model is $\leq \widetilde{\mathcal{O}}(d)$. $\qquad\square$

### G.2 PROOF OF LEMMA 29

*Proof of Lemma 29.* Taking $G_{h, f^*}(f, g) := \langle W_h(f), X_h(g) \rangle$ in Proposition 35, and properties of the effective dimension yields the conclusion that the FE dimension of low Witness rank MDP model is $\leq \widetilde{\mathcal{O}}(W_\kappa)$.

$\qquad\square$

### G.3 PROOF OF LEMMA 30

We first introduce two auxillary lemmas:

**Lemma 36.** Let random variable $x_i \in \mathbb{R}^d$ and $\mathbb{E}\|x_i\|_2^2 \leq B^2$. Then we have that

$$\frac{1}{n} \log \det \left(I + \frac{1}{\lambda} \sum_{t=0}^{n-1} \mathbb{E}[x_t x_t^\top]\right) \leq \frac{d \log \left(1 + \frac{nB^2}{d\lambda}\right)}{n}.$$

*Proof.* We first have

$$\text{trace}\left(I + \frac{1}{\lambda} \sum_{t=0}^{n-1} \mathbb{E}[x_t x_t^\top]\right) = d + \frac{1}{\lambda} \sum_{t=0}^{n-1} \mathbb{E}[\|x_t\|_2^2] \leq d + \frac{nB^2}{\lambda}.$$

Therefore, using the Determinant-Trace inequality, we get the first result,

$$\log \det \left( I + \frac{1}{\lambda} \sum_{t=0}^{n-1} \mathbb{E}[x_t x_t^\top] \right) \leq d \log \frac{\operatorname{trace}\left( I + \frac{1}{\lambda} \sum_{t=0}^{n-1} \mathbb{E}[x_t x_t^\top] \right)}{d} \leq d \log \left( 1 + \frac{nB^2}{d\lambda} \right).$$

Dividing $n$ from the both side of the inequality completes the proof. $\qquad\square$

The following lemma is a variant of the well-known Elliptical Potential Lemma (Dani et al., 2008; Srinivas et al., 2009; Abbasi-Yadkori et al., 2011; Agarwal et al., 2020a).

**Lemma 37** (Randomized elliptical potential). Consider a sequence of random vectors $\{x_0, \ldots, x_{T-1}\}$. Let $\lambda > 0$ and $\Sigma_0 = \lambda I$ and $\Sigma_t = \Sigma_0 + \sum_{i=0}^{t-1} \mathbb{E}[x_i x_i^\top]$, we have that

$$\min_{t \in [T]} \log \left( 1 + \mathbb{E}\|x_t\|_{\Sigma_t^{-1}}^2 \right) \leq \frac{1}{T} \log \left( \frac{\det(\Sigma_T)}{\det(\lambda I)} \right).$$

*Proof.* By definition of $\Sigma_t$ we have that

$$\log \det(\Sigma_{t+1}) = \log \det(\Sigma_t) + \log \det(I + \Sigma_t^{-1/2} \mathbb{E}[x_t x_t^\top](\Sigma_t)^{-1/2}). \tag{G.4}$$

Denote $\Lambda_t = \Sigma_t^{-1/2} \mathbb{E}[x_t x_t^\top](\Sigma_t)^{-1/2}$ with eigenvalue $\lambda_1, \ldots, \lambda_d \geq 0$, we have that

$$\det(I + \Lambda_t) = \Pi_{i=1}^d (\lambda_i + 1) \geq 1 + \sum_{i=1}^d \lambda_i = \operatorname{trace}(1 + \Lambda_t) = 1 + \mathbb{E}\|x_t\|_{\Sigma_t^{-1}}^2. \tag{G.5}$$

Plugging (G.5) into (G.4) gives that

$$\log \det(\Sigma_{t+1}) \geq \log \det(\Sigma_t) + \log(1 + \mathbb{E}\|x_t\|_{\Sigma_t^{-1}}^2) \geq \log \det(\Sigma_t) + \min_{t \in [T]} \log \left( 1 + \mathbb{E}\|x_t\|_{\Sigma_t^{-1}}^2 \right).$$

Taking telescope sum from $t = 0$ to $t = T - 1$ completes the proof. $\qquad\square$

*Proof of Lemma 30.* Given a hypothesis class $\mathcal{F}$ and a coupling function $G(\cdot, \cdot) : \mathcal{F} \times \mathcal{F} \to \mathbb{R}$. Let $n$ to be defined as follows,

$$n := \min \left\{ n \in \mathbb{N} : n \geq e d_\phi \log(1 + 4n d_s R^4/(d_\phi \epsilon'^2)) \right\}.$$

Then we have that $n = \widetilde{O}(d_\phi)$. We will prove $\dim_{FE}(\mathcal{F}, G, \epsilon) \leq n$ by contradiction. Suppose that $\dim_{FE}(\mathcal{F}, G, \epsilon) > n$, there exists an $\epsilon'$-independent (where $\epsilon' \geq \epsilon$) sequence $f_1, \ldots, f_n \in \mathcal{F}$ such that there exist $g_1, \ldots, g_n \in \mathcal{F}$,

$$\begin{cases} \sqrt{\sum_{i=1}^{t-1} (G(g_t, f_i))^2} \leq \epsilon', & t \in [n], \\ |G(g_t, f_t)| > \epsilon', & t \in [n]. \end{cases} \tag{G.6}$$

Recall that the ABC function of KNR model is defiend as,

$$G_{h,f^*}(f, g) = \sqrt{\left\langle \operatorname{vec}\left( (U_{h,f} - U_h^*)^\top (U_{h,f} - U_h^*) \right), \operatorname{vec}\left( \mathbb{E}_{s_h, a_h \sim \pi_g} \phi(s_h, a_h) \phi(s_h, a_h)^\top \right) \right\rangle}$$

$$= \sqrt{\mathbb{E}_{s_h, a_h \sim \pi_g} \|(U_{h,f} - U_h^*) \phi(s_h, a_h)\|^2}.$$

Therefore, condition (G.6) can be reduced to

$$\begin{cases} \sqrt{\sum_{i=1}^{t-1} \mathbb{E}_{s_h, a_h \sim \pi_{f_i}} \|(U_{h,g_t} - U_h^*) \phi(s_h, a_h)\|^2} \leq \epsilon', & t \in [n], \\ \sqrt{\mathbb{E}_{s_h, a_h \sim \pi_{f_t}} \|(U_{h,g_t} - U_h^*) \phi(s_h, a_h)\|^2} > \epsilon', & t \in [n]. \end{cases} \tag{G.7}$$

Denote $U_{h,g_t,j}, j \in [d_s]$ and $U_{h,j}^*, j \in [d_s]$ to be the rows of $U_{h,g_t}$ and $U_h^*$. Taking square over both side of the inequalities in (G.7) gives that

$$
\begin{cases}
\displaystyle\sum_{i=1}^{t-1}\sum_{j=1}^{d_s} \mathbb{E}_{s_h,a_h\sim\pi_{f_i}}[(U_{h,g_t,j} - U_{h,j}^*)\phi(s_h,a_h)]^2 \le \epsilon'^2, & t \in [n], \\[2ex]
\displaystyle\sum_{j=1}^{d_s} \mathbb{E}_{s_h,a_h\sim\pi_{f_t}}[(U_{h,g_t,j} - U_{h,j}^*)\phi(s_h,a_h)]^2 > \epsilon'^2, & t \in [n].
\end{cases}
\tag{G.8}
$$

Define $\Sigma_t = \sum_{i=1}^{t-1} \mathbb{E}_{s_h,a_h\sim\pi_{f_i}}[\phi(s_h,a_h)\phi(s_h,a_h)^\top] + (\epsilon'^2/4d_sR^2)\cdot I$. Then by (G.8), we have that

$$
\sum_{j=1}^{d_s} \|(U_{h,g_t,j} - U_{h,j}^*)\|_{\Sigma_t}^2
$$

$$
= \sum_{i=1}^{t-1}\sum_{j=1}^{d_s} \mathbb{E}_{s_h,a_h\sim\pi_{f_i}}[(U_{h,g_t,j} - U_{h,j}^*)\phi(s_h,a_h)]^2 + (\epsilon'^2/4d_sR^2)\cdot \sum_{j=1}^{d_s}\mathbb{E}_{s_h,a_h\sim\pi_{f_t}}\|U_{h,g_t,j} - U_{h,j}^*\|_2^2
$$

$$
\le \sum_{i=1}^{t-1}\sum_{j=1}^{d_s} \mathbb{E}_{s_h,a_h\sim\pi_{f_i}}[(U_{h,g_t,j} - U_{h,j}^*)\phi(s_h,a_h)]^2 + (\epsilon'^2/4d_sR^2)\cdot \left[2d_s\max_j\|U_{h,g_t,j}\|_2^2 + 2d_s\max_j\|U_{h,j}^*\|_2^2\right]
$$

$$
\le 2\epsilon'^2,
$$

where the first equality is by the Cauchy-Schwartz inequality and the last inequality is by $\|U_{h,g_t,j}\|_2 \le \|U_{h,g_t}\|_2 \le R$, $\|U_{h,j}^*\|_2 \le \|U_h^*\|_2 \le R$. Furthermore we have that

$$
\epsilon'^2 \le \sum_{j=1}^{d_s} \mathbb{E}_{s_h,a_h\sim\pi_{f_t}}[(U_{h,g_t,j} - U_{h,j}^*)\phi(s_h,a_h)]^2
$$

$$
= \sum_{j=1}^{d_s} \mathbb{E}_{s_h,a_h\sim\pi_{f_t}}[(U_{h,g_t,j} - U_{h,j}^*)\Sigma_t^{1/2}\Sigma_t^{-1/2}\phi(s_h,a_h)]^2
$$

$$
\le \sum_{j=1}^{d_s} \mathbb{E}_{s_h,a_h\sim\pi_{f_t}}\|(U_{h,g_t,j} - U_{h,j}^*)\Sigma_t^{1/2}\|_2^2 \cdot \mathbb{E}_{s_h,a_h\sim\pi_{f_t}}\|\Sigma_t^{-1/2}\phi(s_h,a_h)\|_2^2
$$

$$
= \mathbb{E}_{s_h,a_h\sim\pi_{f_t}}\|\Sigma_t^{-1/2}\phi(s_h,a_h)\|_2^2 \cdot \sum_{j=1}^{d_s}\|(U_{h,g_t,j} - U_{h,j}^*)\|_{\Sigma_t}^2,
$$

where the last inequality is by the Cauchy-Schwarz inequality for random variables. Thus, we have that $\mathbb{E}_{s_h,a_h\sim\pi_{f_t}}\|\Sigma_t^{-1/2}\phi(s_h,a_h)\|_2^2 \ge 1/2$ for all $t \in [n]$. By applying Lemma 37, we have that

$$
\min_{t\in[n]} \log(1 + \mathbb{E}_{s_h,a_h\sim\pi_{f_t}}\|\Sigma_t^{-1/2}\phi(s_h,a_h)\|_2^2)
$$

$$
\le \frac{1}{n}\log\left(\frac{\det(\Sigma_{n+1})}{\det(\Sigma_1)}\right) = \frac{1}{n}\log\det\left(1 + \frac{4d_sR^2}{\epsilon'^2}\sum_{i=1}^n \mathbb{E}_{s_h,a_h\sim\pi_{f_i}}[\phi(s_h,a_h)\phi(s_h,a_h)^\top]\right).
$$

The above equation further implies that

$$
\frac{1}{n}\log\det\left(1 + \frac{4d_sR^2}{\epsilon'^2}\sum_{i=1}^n \mathbb{E}_{s_h,a_h\sim\pi_{f_i}}[\phi(s_h,a_h)\phi(s_h,a_h)^\top])\right)
$$

$$
\ge \min_{t\in[n]} \log\left(1 + \mathbb{E}_{s_h,a_h\sim\pi_{f_t}}\|\Sigma_t^{-1/2}\phi(s_h,a_h)\|_2^2\right) \ge \log(3/2).
$$

On the other hand, Lemma 36 implies that

$$
\frac{1}{n}\log\det\left(1 + \frac{4d_sR^2}{\epsilon^2}\sum_{i=1}^n \mathbb{E}_{s_h,a_h\sim\pi_{f_i}}[\phi(s_h,a_h)\phi(s_h,a_h)^\top]\right) \le \frac{d_\phi\log\left(1 + \frac{4nd_sR^4}{d_\phi\epsilon'^2}\right)}{n} \le e^{-1}.
$$

This leads to a contradiction because $\epsilon' \ge \epsilon$ and $\log(3/2) > e^{-1}$. We complete the proof of $\dim_{FE}(\mathcal{F},G,\epsilon) = \widetilde{O}(d_\phi)$. $\qquad\square$

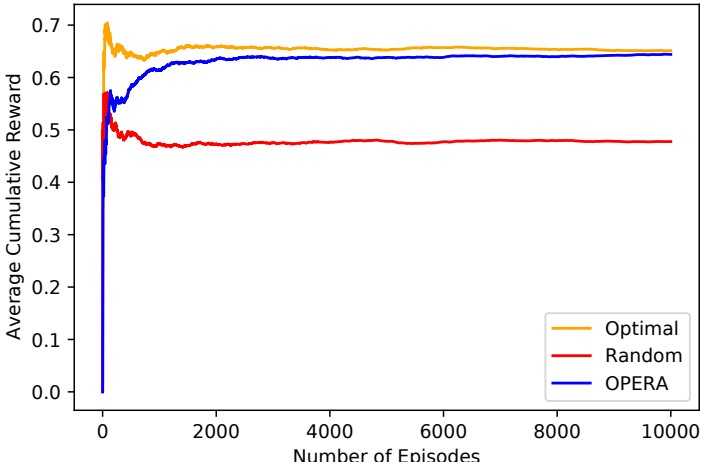

Figure 2: Cumulative regret comparison in the first 10000 episodes of different RL algorithms (i.e., OPERA, Optimal Policy, Random Policy). Results are averaged over 10 runs.

## H  EXPERIMENT

In this section, we carry out experiments to evaluate the empirical performance of our algorithm OPERA for linear mixture MDPs (Algorithm 2). In this experiment, we construct an MDP $M$ with dimension $d = 3$ and episode length $H = 5$. The state space $\mathcal{S}$ consists of $H + 2$ different states $x_1, \ldots, x_{H+2}$ and the action space $\mathcal{A} = \{-1, 1\}^{d-1}$ consists of $2^{d-1}$ different actions. For each step $h \in [H]$ and episode $k \in [K]$, we assume that the reward function $r(s, a)$ is known (so no need to introduce $\psi$ as in Section E.1). In particular, for all $1 \le h \le H$, the reward function $r_h(s, a) = 1$ if and only if $s = x_{H+2}$ and $r_h(s, a) = 0$ otherwise. For each step $h \in [H]$ and corresponding transition probability function $\mathbb{P}_h$, $x_{H+1}$ and $x_{H+2}$ are absorbing states. For other states $x_h(1 \le h \le H)$, the transition probability satisfies that

$$\mathbb{P}_h(x_{h+1}|x_h, \mathbf{a}) = 0.95 - \langle 0.01 \cdot \mathbf{1}_{d-1}, \mathbf{a} \rangle,$$
$$\mathbb{P}_h(x_{H+2}|x_h, \mathbf{a}) = 0.05 + \langle 0.01 \cdot \mathbf{1}_{d-1}, \mathbf{a} \rangle,$$

where each $\mathbf{1}_{d-1}$ is a $(d-1)$-dimensional vector of all ones, $\mathbf{a} \in \mathcal{A}$ is also a $(d-1)$-dimensional vector. Then we have that $\mathbb{P}_h(x_{h'} \mid x_h, \mathbf{a}) = \langle \theta_h^*, \phi(x_{h'}, \mathbf{a}, x_h) \rangle$ where $\theta_h^* = [0.01 \cdot \mathbf{1}_{d-1}, 1]$, and $\phi(x_{h'}, \mathbf{a}, x_h)$ are as follows,

- $\phi(x_{h'}, \mathbf{a}, x_h) = [-\mathbf{a}, 0.95]$ if $1 \le h \le H$ and $h' = h + 1$,
- $\phi(x_{h'}, \mathbf{a}, x_h) = [\mathbf{a}, 0.05]$ if $1 \le h \le H$ and $h' = H + 2$,
- $\phi(x_{h'}, \mathbf{a}, x_h) = [\mathbf{0}_{d-1}, 1]$ if $h' = h = H + 1$,
- $\phi(x_{h'}, \mathbf{a}, x_h) = [\mathbf{0}_{d-1}, 1]$ if $h' = h = H + 2$,
- $\phi(x_{h'}, \mathbf{a}, x_h) = \mathbf{0}_d$, otherwise.

Here $\mathbf{0}_{d-1}$ is a $(d-1)$-dimensional vector of all zeros. We compare our algorithm OPERA with the following two baselines: *Optimal* (optimal policy) and *Random* (uniformly random policy which chooses actions uniformly from $\mathcal{A}$). For numerical stability, we add $\lambda I$ to $\Sigma_h^{(t)}$ with $\lambda = 1$ and use CVX (Diamond & Boyd, 2016; Agrawal et al., 2018) to approximately solve (E.3) when we implement Algorithm 2. The cumulative rewards of different algorithms averaged over 10 runs for the first 10000 episodes are plotted in Figure 2. We can see that OPERA performs much better than the random policy and can converge to the optimal policy after 10000 episodes.

