# OpenReview forum: "A General Framework for Sample-Efficient Function Approximation in Reinforcement Learning"
_ICLR.cc/2023/Conference — ICLR 2023 notable top 25%_

### Official Review · Reviewer_2LkA · 2022-10-21

**Confidence:** 2
**Correctness:** 4
**Technical Novelty And Significance:** 3
**Empirical Novelty And Significance:** Not applicable
**Recommendation:** 8

**Clarity, Quality, Novelty And Reproducibility:**

## Clarity

The paper is extremely well written and easy to follow, although a bit on the technical side. I listed some small comments on writing earlier up in the review. The two most useful clarifications which I think could help:
- Clarify the roll of $\pi_{est}$ on page 8
- Simplify section 3.2 (delaying some examples which readers could be familiar with in the literature) to allow for more discussion around ABC in Section 3.2

## Quality + Novelty

The paper provides novel theoretical contributions for developing an encompassing framework to understand the complexity of model-free and model-based reinforcement learning with function approximation.  The theoretical results then highlight the fact that the definition encompasses many well-studied notions of complexity in the literature.  They complement the new measure with a simple algorithm based on confidence sets which achieve strong regret guarantees (although no matching lower bounds are included).

**Strength And Weaknesses:**

## Strengths

1. The authors provide a novel problem complexity measure which subsumes several others which are studied in the literature.
2. The authors provide a simple algorithmic framework based on optimistic principle over uncertainty sets which achieves strong regret performance.

## Weaknesses

1. The authors include no discussion on the computational complexity of the algorithm.
2. There are no theoretical matching lower bounds (or discussion along this point)
3. The authors provide no empirical results of their algorithm's performance
4. The theoretical contributions and algorithm design seems like a straightforward extension of prior mechanisms with general function approximation techniques.  The novel algorithmic contributions could be highlighted more in the writing.


**Summary Of The Paper:**

The paper proposes a general framework which unifies problem complexity measures with function approximation in the model-based and model-free lens for reinforcement learning.  In particular, the authors propose Admissible Bellman Characterization (ABC) class that subsumes many prior models for problem complexity in the prior literature.  The authors also provide a new algorithm, OPERA, achieving regret bounds that match or improve over the prior literature.

More concretely, the authors consider an MDP $(S,A,r,P,H)$ where $S$ is state space, $A$ the action space, $r$ the reward function and $P$ the global transition distribution, and $H$ the horizon.  Given access to a function class (representing either the transition kernel in model-based or the $Q$ function for model-free) typical algorithms come up with complexity measures of the function class and design algorithms with sublinear regret with respect to that complexity measure.  The main point of this paper is to present a unified complexity measure framework that includes all model-free and model-based RL classes, while simultaneously allowing a simple algorithm with sample efficiency guarantees.  To this lens, the authors make two several frameworks:
1. The authors propose a new framework called ABC that covers structural assumptions previously studied in model-free and model-based RL.  To some extent, the framework can be thought of as extending the Eluder Dimension to include (a) arbitrary coupling functions for measuring the error, (b) a discriminator function which characterizes the interaction between different hypotheses
2. Under the ABC framework the authors design an algorithm OPERA based on maximizing value function in a confidence region around model estimation - showing that the framework has sublinear regret guarantees.

## Questions
- What is the role of $\pi_{est}$ on page 8?

## Minor Comments
- FLAMBE is used to refer to both the model and the algorithm
- Value terms are swapped in regret definition on page 3
- Definition of $\pi_f(s)$ on bottom of page 3 has typo in it

**Summary Of The Review:**

The paper provides strong theoretical contributions for developing a framework which unifies model-based and model-free RL via their Admissible Bellman Characterization formulation - which subsumes most models in the literature for tractable RL with function approximation.  The paper is extremely well written, although very technical, but highlights the main differences and contributions between the paper and the related work. However, the paper offers no empirical results (or discussion on computational tractability of their proposed algorithm), although standard in the RL with function approximation literature.

---

> ### Author Response · Authors · 2022-11-13
> **Response to Reviewer 2LkA**
>
> Thank you for your supportive comments!
>
> We address your questions and concerns as follows. We have also revised the paper accordingly and the revision is highlighted in violet color.
>
> **Q1**: Role of $\pi_{est}$
>
> **A1**: $\pi_{est}$ is designed for instances where the $V$-type complexity is better than the $Q$-type complexity, e.g., the reactive POMDP. In this case, we should take $\pi_{est} = U(A)$ (a uniform policy) and use importance sampling to get an expectation over $s_h \sim \pi_f, a_h \sim \pi_g$.
>
> **Q2**: Computational efficiency of the algorithm
>
> **A2**: In general, our algorithm OPERA is not computationally efficient because the hypothesis class $\mathcal{F}$ can be arbitrarily complex. However, OPERA is oracle efficient given the planning oracle for solving the optimization problem in Line 3 of Algorithm 1.
>
> In addition, for special MDP instances such as linear mixture MDPs and KNR, our algorithm reduces to existing algorithms and therefore can be implemented in practice. We have added a case-by-case discussion of the computational complexity in Section E.1, E.2, E.3, assuming a known complexity of the planning oracle.
>
> **Q3**: Discussion on lower bound
>
> **A3**: We add a discussion in Section B that when specialized to linear mixture MDPs, our result matches the lower bound up to $\mathcal{O}(H^{1/2})$. For the KNR and low witness rank cases, we are not aware of existing lower bound results. We leave the discussion on lower bounds for other instances, and possible lower bound results on the general framework to future work.
>
> **Q4** The authors provide no empirical results of their algorithm's performance
>
> **A4** While the key contributions of our work are on the theoretical side, our algorithm can be implemented if we specify the hypothesis class $\mathcal{F}$. We added a related discussion in Section E.3. In detail, for the linear mixture MDP (Modi et al., 2020),  OPERA will reduce to Algorithms 2, which can be solved by off-the-shelf optimizers such as CVX. For the KNR case, our algorithm will reduce to Algorithm 4, which is the same as LC3 algorithm (Kakade et al., 2020) in the time-homogenous case. However, it is hard to compare the empirical performance with other frameworks with general function approximation (e.g., Du et al., 2021, Jin et al. 2021, Foster et al. 2021), since all these works do not provide empirical experiments.
>
>
> **Q5**: Simplify section 3.2
>
> **A5**: As suggested, we postpone the definition of the linear mixture MDPs originally in Section 3.2 to the appendix.
>
> **Q6**:  The novel algorithmic contributions could be highlighted more in the writing
>
> **A6**: In Section 4.3, we have highlighted in our revision that our ABC framework and OPERA algorithm not only provide a wider coverage of RL problems than previous general RL frameworks, but also match or even outperform previous RL algorithms for specific examples in terms of sample complexity.
>
> **Q7**: Minor Comments
>
> **A7**: We have addressed all your minor comments accordingly.

---

> ### Author Response · Authors · 2022-11-17
> **Empirical Results on a Synthetic Data**
>
> We have added an experiment of OPERA on a synthetic dataset for learning linear mixture MDPs in Section H. In detail, we have implemented Algorithm 2 using CVX to approximately solve Eq. (E.3). We compare OPERA with two baselines: (1) Optimal (optimal policy) and (2) Random (uniformly random policy which chooses actions uniformly from the action set). We plot the cumulative rewards of different algorithms averaged over 10 runs for the first $10000$ episodes in Figure 2. We can see that OPERA performs much better than the random policy and can converge to the optimal policy after $10000$ episodes. For more details about the experiment, please refer to Section H. We hope this can further address your concern about no empirical results.

---

### Official Review · Reviewer_TL1B · 2022-10-24

**Confidence:** 3
**Correctness:** 4
**Technical Novelty And Significance:** 2
**Empirical Novelty And Significance:** Not applicable
**Recommendation:** 8

**Clarity, Quality, Novelty And Reproducibility:**

Overall, I think the paper scores very highly in clarity.
The quality, and reproducibility also appear to be solid... although I have not checked all steps in detail.

In terms of *novelty* I think the main contribution is the calm and coherent way of combining lots of existing perspectives, and rationalizing this treatment in one place.
This is definitely valuable, but I would not rate *novelty* as one of the main strengths.

**Strength And Weaknesses:**

There are several things to like about the paper:
- The writing and paper structure are good, and give a good overview of the many lines of research in this area.
- The general analysis and flavour of the results are clear and concise. Take, for example, Theorem 12 as an example. This clearly highlights the main dependencies in the regret bound.
- The ABC class supersedes the many different notions of complexity, as outlined well in Figure 1.

There may be a few places where the paper falls down in terms of top-level impact:
- First, it feels like this work is really mostly consolidating/summarising threads of research that have been going on for a while in the eluder-dimension++ pieces of RL with generalization. This is valuable, but feels more "cleaning up" than forging new course.
- The OPERA algorithm is really more of a high-level principle than a practical algorithm. Thinking about how we can make scalable/practical algorithms is probably not just an "unnecessary complication" but I agree it is OK to leave this to future work in this conference paper.
- There is a *lot* of material to go through in reviewing this paper... so many theorems and definitions. This is great for clarity, since it allows a motivated reader to go through piece by piece, but there is a slight concern with *so many* pieces that it takes away from the "main thing".

**Summary Of The Paper:**

This paper presents a unified perspective and analysis for almost all of the existing sample efficient RL algorithms.
This notion is based on Admissible Bellman Characterisation, and provides an a bound in terms of the functional eluder dimension.
Together with this analysis, the authors present an algorithm OPERA, which really is a general form of optimism in the face of uncertainty.
While this algorithm is not necessarily practical (it may not be computable) it provides some general flavour of the types of results that might be possible.

**Summary Of The Review:**

This paper collates many existing results in the OFU-RL literature with function approximation.
By revisiting the functional eluder dimension under the ABC condition, they show that OPERA algorithm provides a more unified perspective on statistically efficient reinforcement learning.

---

> ### Author Response · Authors · 2022-11-13
> **Response to Reviewer Reviewer TL1B**
>
> Thank you for your positive comments! We will address your questions and concerns as follows.
>
> **Q1**: Mostly consolidating/summarising threads of research that have been going on for a while
>
> **A1**: While our algorithm design and proofs are very different from either bilinear classes, DEC or witness rank, they indeed share some similarity with the GOLF algorithm and the BE dimension (Jin et al. (2021)). We have mentioned in several parts of our paper that our algorithm is a generalization of GOLF that applies to a much wider range of MDPs while maintaining the statistical rates. In particular, our framework can deal with model-based RL while BE dimension cannot.
>
> Compared with the Bellman representability in the DEC framework, the adaptation to witness rank and KNR is nontrivial, as there are nonlinear max operators and monotone functions involved. In addition to the max operator and the vector-valued estimation function, (1) in Definition 5 is more relaxed than the related existing frameworks, as $\mathbb{E}(X^2) \geq (\mathbb{E}X)^2$. Moreover, the DEC framework in Foster et al. (2021) relies on complicated algorithms to explicitly calculate the bound and uses a posterior sampling based approach for exploration. We compare the sample complexity results in Table 1 and show that our algorithm achieves state-of-the-art results in almost all model-based and model-free MDP classes.
>
> For the comparison with the witness rank model and the bilinear classes, in addition to the relaxed form in (i) of Definition 5 and vector-valued estimator, we allow a generalized estimation function and a more flexible coupling function beyond the bilinear form. For example, our framework uses a coupling function in the form of $\sqrt{\cdot}$ for both KNR and general Bellman complete, which is crucial to achieve the $\sqrt{T}$ regret under our framework.
>
> **Q2**: High-level principle rather than a practical algorithm
>
> **A2**: OPERA is not only a high-level principle, but also a practical algorithm when specialized to specific MDP instances.
>
> For example, for linear MDPs, OPERA reduces to the ELEANOR algorithm (Zanette et al. 2020), and for linear mixture MDPs, OPERA reduces to the global optimization version of UCRL-VTR.
>
> Thank you for your suggestion. We have made the following revision to discuss how OPERA can be implemented in practice and its scalability.
>
> **(a.)** We added a discussion on the computational complexity of OPERA and pointed out that the algorithm is oracle efficient (Section E.1, E.2, E.3)
>
> **(b.)** We added discussions on how to compute the planning oracle in practice (Section E.3).
>
> **Q3**: a slight concern with so many pieces that it takes away from the "main thing".
>
> **A3**: We have tried our best to make the definitions and theorems succinct and concise while not affecting the understanding of our framework.
>
> To avoid distracting the audience, we further moved the definition of linear mixture MDPs from the main paper to the appendix. Instead, we have added a discussion in Section 4.3 on the algorithmic innovation and advantage of OPERA under specific instances.
>
> The revision is marked in violet color.

---

### Official Review · Reviewer_JjVG · 2022-10-24

**Confidence:** 3
**Clarity, Quality, Novelty And Reproducibility:** Good
**Correctness:** 4
**Technical Novelty And Significance:** 3
**Empirical Novelty And Significance:** Not applicable
**Recommendation:** 8

**Strength And Weaknesses:**

Strength:
1. The authors proposed a unified framework which covers a large section of MDPs, both model-free and model-based
2. The framework maintains the best-known sample complexity results for special instances.
3. They also propose a novel algorithm, OPtimization-based ExploRation with Approximation (OPERA), which maximizes the value function constrained in a small confidence region around the model minimizing the estimation function.

Weakness:
Although this framework improves the sample bounds, it is as generic as the existing framework, meaning the algorithms which were not covered in previous framework (such as those with state-action aggregation) are also not covered here.

**Summary Of The Paper:**

This paper proposes a unified framework which encompasses both model-based and model-free MDP algorithms. The framework, Admissible Bellman Characterization, covers a wide range of MDPs. The key idea is to use decomposable estimation function which has decomposable sturctural properties for optimization-based exploration (which is also their algorithm). To measure the complexity of model class, they use the functional eluder dimension.

**Summary Of The Review:**

It is good paper, written quite well. It improves on the existing frameworks which could not preserve the sample complexity bounds.

---

> ### Author Response · Authors · 2022-11-13
> **Response to Reviewer JjVG**
>
> Thank you for your positive comments! We will address your concerns as follows.
>
> **Q1**: the algorithms which were not covered in previous framework (such as those with state-action aggregation) are also not covered here.
>
> **A1**: In fact, our framework can cover the state-action aggregation model [1] you mentioned, and we have added a detailed discussion in Section B.7 (marked in violet color). In detail, we can choose the estimation function to be the Bellman residual, and take $G(f, g) = \left\langle w_f, E_{\pi_g}\phi(s_h, a_h)\right\rangle$ $- E_{\pi_g} \max_{a\in\mathcal{A}}\left\langle w_f, \phi(s_{h+1}, a)\right\rangle$ (for notational simplicity, we set $w^* = 0$ here), where $w_f$ is the value vector of all aggregated state-action pairs and $\phi$ is a $0-1$ matrix. The caveat is that it is unclear if the FE dimension based on the aforementioned $G(f, g)$ is bounded in a nontrivial way. We will study its FE dimension in our future work. In contrast, the bilinear classes cannot cover the above state-action aggregation model due to its restrictive bilinear form.
>
> We would also like to emphasize that our framework extends the coverage of existing frameworks to a large extent, and provides new insights and new techniques (e.g., vector-valued estimation function, nonlinear coupling function, FE dimension, etc.). We leave it to future work to explore new MDP models within our framework that are not covered by previous frameworks.
>
> [1] Dong et al. Provably Efficient Reinforcement Learning with Aggregated States, arXiv 1912.06366

---

### Decision · Program_Chairs · 2023-01-20

**Decision:**

Accept: notable-top-25%

**Justification For Why Not Higher Score:**

The results are quite technical, and are mostly of interest to a subset of RL theory researchers. Moreover, it does not seem that the results are breaking new grounds that everyone in the community should be immediately aware of. They are important, but as far as I can say, are mostly technical refinements of existing concepts.

**Justification For Why Not Lower Score:**

This is a solid paper with three 8s.

**Metareview: Summary, Strengths And Weaknesses:**

The paper considers the problem of sample efficient RL with large state space that requires function approximation. It defines a new class of Admissible Bellman Characterization (ABC) that includes existing structural characterization, such as Bellman Eluder and Bilinear Classes, as special cases. The paper also introduces OPERA, an optimistic under certain constraint set algorithm, for efficient exploration. It provides a regret bound that is proportional to $\sqrt{T dim_{FE} }$, where $dim_{FE}$ is the introduced functional Eluder dimension.

While reading this paper, I was wondering whether there is any interesting class of MDPs that is within ABC but is not already captured with one of the existing characterizations? The paper might have already answered it and I just missed it. But if not, it would be good to discuss it.

Overall, all reviewers agree that this is a good paper. The ABC class encompasses many other previous results. The paper provides an algorithm with a good regret guarantee and is written clearly. I recommend the acceptance of this paper.

**Note From Pc:**

if the above contains the word "oral" or "spotlight" please see: "oral" presentation means -> notable-top-5% and "spotlight" means -> notable-top-25%. As stated in our emails, we are disassociating presentation type from AC recommendations